# Deep linear networks for regression are implicitly regularized towards flat minima

**Pierre Marion**
Institute of Mathematics
EPFL
Station Z, CH-1015 Lausanne, Switzerland
`pierre.marion@epfl.ch`

**Lénaïc Chizat**
Institute of Mathematics
EPFL
Station Z, CH-1015 Lausanne, Switzerland
`lenaic.chizat@epfl.ch`

## Abstract

The largest eigenvalue of the Hessian, or sharpness, of neural networks is a key quantity to understand their optimization dynamics. In this paper, we study the sharpness of deep linear networks for univariate regression. Minimizers can have arbitrarily large sharpness, but not an arbitrarily small one. Indeed, we show a lower bound on the sharpness of minimizers, which grows linearly with depth. We then study the properties of the minimizer found by gradient flow, which is the limit of gradient descent with vanishing learning rate. We show an *implicit regularization towards flat minima*: the sharpness of the minimizer is no more than a constant times the lower bound. The constant depends on the condition number of the data covariance matrix, but not on width or depth. This result is proven both for a *small-scale initialization* and a *residual initialization*. Results of independent interest are shown in both cases. For small-scale initialization, we show that the learned weight matrices are approximately rank-one and that their singular vectors align. For residual initialization, convergence of the gradient flow for a Gaussian initialization of the residual network is proven. Numerical experiments illustrate our results and connect them to gradient descent with non-vanishing learning rate.

## 1 Introduction

Neural networks have intricate optimization dynamics due to the non-convexity of their objective. A key quantity to understand these dynamics is the largest eigenvalue of the Hessian or *sharpness* $S(\mathcal{W})$ (see Section 3 for a formal definition), in particular because of its connection with the choice of learning rate $\eta$. Classical theory from convex optimization indicates that the sharpness should remain lower than $2/\eta$ to avoid divergence (Nesterov, 2018). The relevance of this point of view for deep learning has recently been questioned since neural networks have been shown to often operate at the *edge of stability* (Cohen et al., 2021), where the sharpness oscillates around $2/\eta$, while the loss still steadily decreases, albeit non-monotonically. Damian et al. (2023) explained the stability of gradient descent slightly above the $2/\eta$ threshold to be a general phenomenon for non-quadratic objectives, where the third-order derivatives of the loss induce a self-stabilization effect. They also show that, under appropriate assumptions, gradient descent on a risk $R^L$ implicitly solves the constrained minimization problem

$$\min_{\mathcal{W}} R^L(\mathcal{W}) \quad \text{such that} \quad S(\mathcal{W}) \leqslant \frac{2}{\eta}. \tag{1}$$

Trainability of neural networks initialized with a sharpness larger than $2/\eta$ is also studied by Lewkowycz et al. (2020), which describes a transient catapult regime, which lasts until the sharpness goes below $2/\eta$. These results beg the question of quantifying the largest learning rate that enables successful training of neural networks. For classification with linearly separable data and logistic loss,

38th Conference on Neural Information Processing Systems (NeurIPS 2024).

Wu et al. (2024) show that gradient descent converges for any learning rate. In this work, we address the case of deep linear networks for regression. As illustrated by Figure 1a, the picture then differs from the classification case: **when the learning rate exceeds some critical value, the network fails to learn**. We further remark that this critical value does not seem to be related to the initial scale: when the learning rate is under the critical value, learning is successful for a wide range of initial scales. In Figure 1b, we see that the large initialization scales correspond to initial sharpnesses well over the $2/\eta$ threshold, confirming that training is possible while initializing beyond the $2/\eta$ threshold.

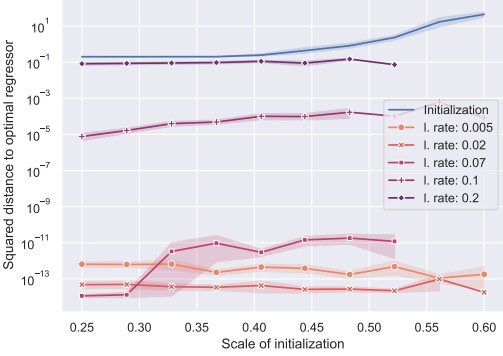
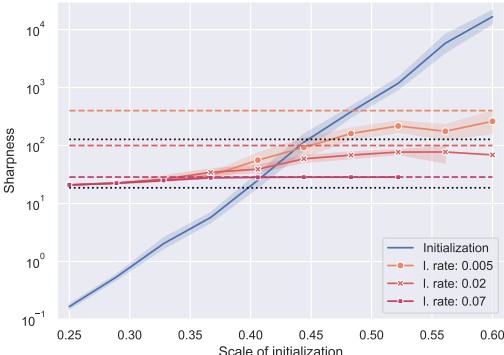

(a) Squared distance of the trained network to the empirical risk minimizer, for various learning rates and initialization scales. Training succeeds when the learning rate is lower than a critical value independent of the initialization scale.

(b) Sharpness at initialization and after training, for various learning rates and initialization scales. For a given learning rate $\eta$, the dashed lines represent the $2/\eta$ threshold. The dotted black lines represent the lower and upper bounds given in Theorem 1 and Corollary 2.

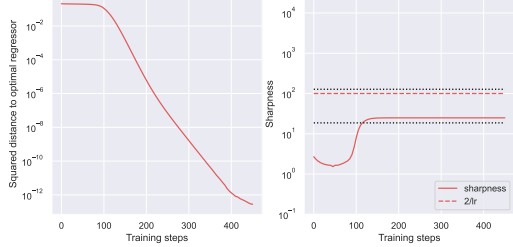
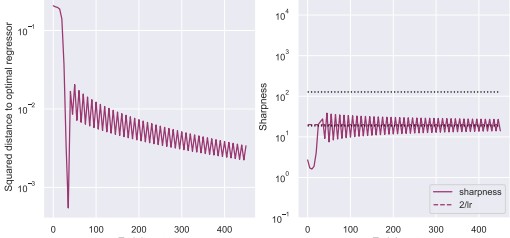

(c) Evolution during training of the squared distance to the empirical risk minimizer and of sharpness, for $\eta = 0.02$ and an initialization scale of 0.35. The network does not enter edge of stability.

(d) Evolution during training of the squared distance to the empirical risk minimizer and of sharpness, for $\eta = 0.1$ and an initialization scale of 0.35. The network enters edge of stability.

Figure 1: Training a deep linear network on a univariate regression task with quadratic loss. The weight matrices are initialized as Gaussian random variables, whose standard deviation is the x-axis of plots 1a and 1b. Experimental details are given in Appendix C.

To understand where the critical value for the learning rate comes from, **we characterize the sharpness of minimizers** of the empirical risk. We show that there exist minimizers with arbitrarily large sharpness, but *not* with arbitrarily small sharpness. Indeed, the sharpness of any minimizer grows linearly with depth, as made precise next.

**Theorem 1.** *Let $X \in \mathbb{R}^{n \times d}$ be a design matrix and $y \in \mathbb{R}^n$ a target. Then the minimal sharpness $S_{\min}$ of any linear network $x \mapsto W_L \dots W_1 x$ of depth $L$ that implements the optimal linear regressor $w^\star \in \mathbb{R}^d$ satisfies*

$$2\|w^\star\|_2^{2-\frac{2}{L}} La \leqslant S_{\min} \leqslant 2\|w^\star\|_2^{2-\frac{2}{L}} \sqrt{(2L-1)\Lambda^2 + (L-1)^2 a^2} \,,$$

*where $\Lambda$ is the largest eigenvalue of the empirical covariance matrix $\hat{\Sigma} := \frac{1}{n} X^\top X$, and $a := (w^\star/\|w^\star\|)^\top \hat{\Sigma} (w^\star/\|w^\star\|)$.*

We note that this bound is similar to a result of Mulayoff and Michaeli (2020), although we alleviate their assumption of data whiteness ($\hat{\Sigma} = I$). In particular, we do not assume that data covariance

matrix is full rank. Furthermore, our proof technique differs, since we do not require tensor algebra, and instead exhibit a direction in which the second derivative of the loss is large.

This result shows that it is not possible to find a minimizer of the empirical risk in regions of low sharpness. Combined with (1), this suggests an interpretation of the critical value for the learning rate: gradient descent should not be able to converge to a global minimizer as soon as

$$S_{\min} > \frac{2}{\eta} \quad \Leftrightarrow \quad \eta > \frac{2}{S_{\min}} \simeq (\|w^\star\|_2^{2-\frac{2}{L}} La)^{-1}.$$

This is confirmed experimentally by Figure 2, which shows that the critical value of the learning rate matches our theoretical prediction. We note that this gives a quantitative answer to the observations of Lewkowycz et al. (2020), which shows the existence of a maximal architecture-dependent learning rate beyond which training fails. The dependence of learning rate on depth (namely, constant over depth) also matches other papers that study scaling of neural networks (Chizat and Netrapalli, 2024; Yang et al., 2024).

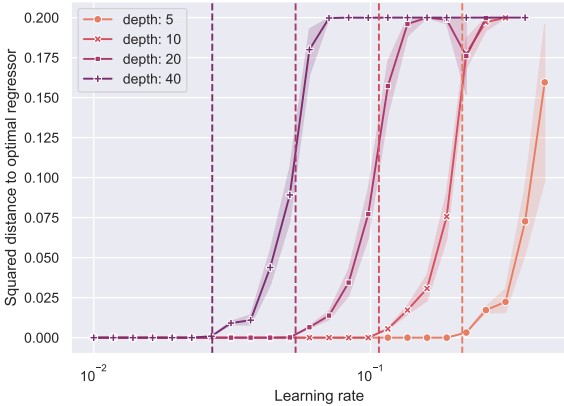

Figure 2: Squared distance of the trained network to the empirical risk minimizer, for various learning rates and depth. For each depth, learning succeeds if the learning rate is below a threshold, which corresponds to the theoretical value $\frac{2}{S_{\min}} \simeq (\|w^\star\|_2^{2-\frac{2}{L}} La)^{-1}$ of Theorem 1 (dashed vertical line).

To deepen our understanding of the training dynamics, we aim at quantifying the sharpness of the minimizer found by gradient descent (when it succeeds): we know that it has to be larger than $S_{\min}$, but is it close to $S_{\min}$ or is it much larger? Inspecting Figure 1b, we see that the answer empirically depends on the interplay between initialization scale and learning rate. For small initialization scale, the sharpness after training is equal to a value relatively close to $S_{\min}$ and independent of the learning rate. As the initialization scale increases, the sharpness of the trained network also increases, and plateaus at the value $2/\eta$. The plateauing for large initialization scales can be explained by the edge of stability analysis (1), which upper bounds the sharpness of the minimizer by $2/\eta$ (see Figure 1d). However, this gives no insight on the value of the sharpness when the learning rate is sufficiently small so that the network does not enter the edge of stability regime (see Figure 1c).

In this paper, we study the limiting case for vanishing $\eta$, i.e., training with gradient flow. **As our main finding, we bound the sharpness of the minimizer found by gradient flow** in the case of overdetermined regression, meaning that the sample size $n$ is larger that the data dimension $d$ and the data empirical covariance matrix is nonsingular. In this case, we prove that the ratio between the sharpness after training and $S_{\min}$ is less than a constant depending mainly on the condition number of the empirical covariance matrix $\hat\Sigma$. In particular, the ratio does not depend on the width or depth of the network. This shows an **implicit regularization towards flat minima**. Note that the phenomenon we exhibit is different from the well-studied implicit regularization towards flat minima caused by stochasticity in SGD (Keskar et al., 2017; Smith and Le, 2018; Blanc et al., 2020; Damian et al., 2021; Li et al., 2022; Liu et al., 2023). In the present study, the dynamics are purely deterministic, and the low sharpness is due to the fact that the weight matrices found by gradient flow have (approximately) the same norm across layers, and that this norm is (approximately) the smallest possible one so that the network can minimize the risk.

**Link with generalization.** Flatter minima have been found to generalize better (Hochreiter and Schmidhuber, 1997; Jastrzębski et al., 2017; Keskar et al., 2017; Jiang et al., 2020), although the picture is subtle (Dinh et al., 2017; Neyshabur et al., 2017; Andriushchenko et al., 2023). However, in this paper, we focus on the link of sharpness with (non-convex) optimization dynamics, rather than with generalization abilities. Indeed, our implicit regularization result holds in the overdetermined setting, where all minimizers of the empirical risk implement the same function thus have the same generalization error, although they differ in parameter space and in particular have different sharpnesses. We leave to future work extensions to more complex settings where our approach may link sharpness and generalization, beginning with deep linear networks for underdetermined regression. We refer to Appendix C for more comments on the link with generalization.

We investigate two initialization schemes, quite different in nature: small-scale initialization and residual initialization. Let us explain both settings, by presenting our approach, contributions of independent interest, and related works.

**Small-scale initialization.** In this setting, we consider an initialization of the weight matrices $W_k$ with i.i.d. Gaussian entries of small variance. Initialization scale is known to play a key role in training of neural networks: small-scale initialization corresponds to the "feature learning" regime where the weights change significantly during training, by opposition to the "lazy" regime (see, e.g., Chizat et al., 2019). We show convergence of the empirical risk to zero, then **characterize the structure of the minimizer found by gradient flow**, a novel result of interest independently of its connection with sharpness. At convergence, **the weight matrices are close to being rank-one**, in the sense that all their singular values but the largest one are small. Furthermore, the first left singular vector of any weight matrix aligns with the first right singular vector of the next weight matrix. From this specific structure, we deduce our bound on the sharpness of the trained network. The bound is illustrated on Figure 1b, where our lower and upper theoretical bounds on the sharpness are plotted as dotted black lines. We observe that the sharpness after training, when starting from a small-scale initialization, is indeed situated between the black lines.

The result and proof extend the study by Ji and Telgarsky (2020) for classification, although the parameters do not diverge to infinity contrarily to the classification case, thus requiring a finer control of the distance to the rank-one aligned solution. In regression, implicit regularization towards low-rank structure in parameter space was also studied by Saxe et al. (2014); Lampinen and Ganguli (2019); Gidel et al. (2019); Saxe et al. (2019); Varre et al. (2023) for two-layer neural networks and in Timor et al. (2023) for deep ReLU networks. This latter paper assumes convergence of the optimization algorithm and show that a solution with minimal $\ell_2$-norm has to be low-rank. In our linear setting, we instead show convergence. As detailed below, we impose mild requirements on the structure on the initialization beyond its scale; they are satisfied for instance by initializing one weight matrix to zero and the others with i.i.d. Gaussian entries. In particular, we do not require the so-called "zero-balanced initialization" as is common in the literature on deep linear networks (see, e.g., Arora et al., 2018; Advani et al., 2020; Li et al., 2021) or a deficient-margin initialization as in Arora et al. (2019a). Finally, the limit when initialization scale tends to zero has been described for deep linear networks in Jacot et al. (2021) for multivariate regression. It consists in a saddle-to-saddle dynamics, where the rank of the weight matrices increases after each saddle. The present study considers instead a non-asymptotic setting where the initialization scale is small but nonzero, and shows convergence to a rank-one limit because univariate and not multivariate regression is considered.

We note that sharpness at initialization can be made arbitrarily small, since it is controlled by the initialization scale, while sharpness after training scales as $\Theta(L)$. This therefore showcases an **example of sharpening during training** (although we make no statement on monotonicity).

**Residual initialization.** Architectures of deep neural networks used in practice often present residual connections, which stabilize training (He et al., 2015). A simple non-linear residual architecture writes $h_{k+1} = h_k + \sigma(N_k h_k)$. Removing the non-linearity, we get $h_{k+1} = (I + N_k)h_k$, which prompts us to consider deep linear networks with square weight matrices $W_k \in \mathbb{R}^{d \times d}$ that are initialized as

$$W_k(0) = I + \frac{s}{\sqrt{Ld}} N_k \,, \tag{2}$$

where the $N_k \in \mathbb{R}^{d \times d}$ are filled with i.i.d. standard Gaussian entries and $s \geqslant 0$ is a hyperparameter tuning the initialization scale. The scaling of the residual branch in $1/\sqrt{d}$ is common and corresponds

for instance to the so-called Glorot and He initializations respectively from Glorot and Bengio (2010) and He et al. (2015). It ensures that the variance of the residual branch is independent of the width $d$. Similarly, as studied in Arpit et al. (2019); Marion et al. (2022), the scaling in $1/\sqrt{L}$ is the right one so that the initialization noise neither blows up nor decays when $L \to \infty$. Note that, in practice, this scaling factor is often replaced by batch normalization (Ioffe and Szegedy, 2015), which has been shown empirically to have a similar effect to $1/\sqrt{L}$ scaling (De and Smith, 2020).

In this setting which we refer to as *residual initialization*, we show global convergence of the empirical risk. To our knowledge, **it is the first time that convergence is proven for a standard Gaussian initialization of the residual network outside the large width $d \geqslant n$ regime**. Previous works considered either an identity initialization (Bartlett et al., 2018; Arora et al., 2019a; Zou et al., 2020) or a smooth initialization such that $\|W_{k+1}(0) - W_k(0)\|_F = \mathcal{O}(1/L)$ (Sander et al., 2022; Marion et al., 2024). The extension to standard Gaussian initialization leverages sharp bounds for the singular values of the product of $W_k(0)$. Our main assumption, in alignment with the literature, is that the risk at initialization should not be larger than a constant (depending on $\hat{\Sigma}$ and $s$).

We then show that the weights after training can be written

$$W_k(\infty) = I + \frac{s}{\sqrt{Ld}}N_k + \frac{1}{L}\theta_k \,,$$

where the Frobenius norm of the $\theta_k$ is bounded by a constant (depending only on $s$). This structure finally enables us to bound the sharpness of the trained network. Remark that, to connect this analysis with our discussion of sharpness in univariate regression, we add to the residual network a final fixed projection vector $p \in \mathbb{R}^d$, so that our neural network writes $x \mapsto p^\top W_L \dots W_1 x$, but the proof of convergence also holds without this projection.

Experimentally, we give in Appendix C plots in the residual case that are qualitatively similar to Figure 1. The main difference is that the initial sharpness is less sensitive to the initialization scale $s$.

**Organization of the paper.** Section 2 details our setting and notations. Section 3 studies the sharpness of minimizers of the empirical risk and proves Theorem 1. Dynamics of gradient flow starting from small-scale initialization and residual initialization are respectively presented in Sections 4 and 5. The Appendix contains proofs, additional plots, experimental details, and related works.

## 2 Setting

**Model.** We consider linear networks of depth $L$ from $\mathbb{R}^d$ to $\mathbb{R}$, which are linear maps

$$x \mapsto p^\top W_L \dots W_1 x \tag{3}$$

parameterized by weight matrices $W_1, \dots, W_L$, where $W_k \in \mathbb{R}^{d_k \times d_{k-1}}$, $d_0 = d$ and $p \in \mathbb{R}^{d_L}$ is a fixed vector. This definition includes both fully-connected networks by setting $d_L = 1$ and $p = 1$, and residual networks by setting $d_1 = \dots = d_L = d$, the $W_k$ close to the identity, and $p$ to some fixed (potentially random) vector in $\mathbb{R}^d$. We let $\mathcal{W} = (W_1, \dots, W_L)$ and $w_{\text{prod}} = W_1^\top \dots W_L^\top p$. Given $X \in \mathbb{R}^{n \times d}$ a design matrix and $y \in \mathbb{R}^n$ a target, we consider the empirical risk for regression

$$R^L(\mathcal{W}) := \frac{1}{n}\|y - XW_1^\top \dots W_L^\top p\|_2^2 = \frac{1}{n}\|y - Xw_{\text{prod}}\|_2^2 =: R^1(w_{\text{prod}}) \,.$$

The notations $R^L(\mathcal{W})$ and $R^1(w_{\text{prod}})$ may seem redundant, but are actually practical to define gradients of the risk both with respect to a single matrix $W_k$ and to the product $w_{\text{prod}}$. Let $\hat{\Sigma} := \frac{1}{n}X^\top X$ the empirical covariance matrix, and $\lambda$ and $\Lambda$ respectively its smallest *nonzero* and largest eigenvalue. For now, we do not assume that $\hat{\Sigma}$ is full rank, so there is more than one solution to the regression problem $\min_{w_{\text{prod}} \in \mathbb{R}^d} R^1(w_{\text{prod}})$. We denote by $w^\star \in \mathbb{R}^d$ the smallest norm solution, and we let $R_{\min} = R^1(w^\star)$ be the minimum of $R^1$ (and $R^L$). In all the following, we assume that $w^\star \neq 0$. Note that, due to the overparameterization induced by the neural network, there exists an infinity of parameterizations of the mapping $x \mapsto w^{\star\top}x$.

**Gradient flow.** We consider that the neural network is trained by gradient flow on the empirical risk $R^L$, that is, the parameters evolve according to the ordinary differential equation

$$\frac{dW_k}{dt} = -\frac{\partial R^L}{\partial W_k} \,. \tag{4}$$

An application of the chain rule gives

$$\nabla_k R^L(\mathcal{W}) := \frac{\partial R^L}{\partial W_k} = W_{k+1}^\top \ldots W_L^\top p \nabla R^1(w_{\text{prod}})^\top W_1^\top \ldots W_{k-1}^\top , \tag{5}$$

with

$$\nabla R^1(w_{\text{prod}}) = -\frac{2}{n} X^\top (y - X w_{\text{prod}}) = -\frac{2}{n} X^\top X (w^\star - w_{\text{prod}}) . \tag{6}$$

where the second equality is a consequence of $\nabla R^1(w^\star) = 0$.

**Notations.** For $k \in \{1, \ldots L\}$, we denote respectively by $\sigma_k$, $u_k$, and $v_k$ the first singular value (which equals the $\ell_2$ operator norm), the first left singular vector and the first right singular vector of $W_k$. The $\ell_2$ operator norm of a matrix $M$ is denoted by $\|M\|_2$ and its Frobenius norm by $\|M\|_F$. Its smallest singular value is denoted by $\sigma_{\min}(M)$. For a vector $v$, we let $\|v\|_2$ its Euclidean norm. Finally, for quantities that depend on the gradient flow time $t$, we omit for concision their explicit dependence on $t$ when it is dispensable.

## 3 Estimates of the minimal sharpness of minimizers

To define the sharpness of the model, we let $D = \sum_{k=1}^L d_k d_{k-1}$ and identify the space of parameters with $\mathbb{R}^D$, which amounts to stacking all the entries of the weight matrices in a large $D$-dimensional vector. Then the norm of the parameters seen as a $D$-dimensional vector can be related to the Frobenius norm of the matrices by

$$\|\mathcal{W}\|_2^2 = \sum_{k=1}^L \|W_k\|_F^2 .$$

This allows us to define the Hessian of the risk $H : \mathbb{R}^D \to \mathbb{R}^{D \times D}$, and we denote by $S(\mathcal{W})$ its largest eigenvalue for some parameters $\mathcal{W}$, or sharpness. We note that there exists alternative definitions of the sharpness, but this one is the most relevant to study optimization dynamics. Our results are specific to this definition. The following result gives estimates on the minimal sharpness of minimizers of the empirical risk (and is a strictly stronger statement than Theorem 1).

**Theorem 2.** *Let $S_{\min} = \inf_{\mathcal{W} \in \arg\min R^L(\mathcal{W})} S(\mathcal{W})$ and $a := (w^\star/\|w^\star\|)^\top \hat{\Sigma} (w^\star/\|w^\star\|)$. We have*

$$S_{\min} \geqslant 2a \|w^\star\|_2^{2 - \frac{1}{L}} \|p\|^{\frac{1}{L}} \sum_{k=1}^L \frac{1}{\|W_k\|_F} ,$$

*and*

$$2 \|w^\star\|_2^{2 - \frac{2}{L}} \|p\|^{\frac{2}{L}} La \leqslant S_{\min} \leqslant 2 \|w^\star\|_2^{2 - \frac{2}{L}} \|p\|^{\frac{2}{L}} \sqrt{(2L-1)\Lambda^2 + (L-1)^2 a^2} .$$

The proof of the result relies on the following variational characterization of the sharpness as the direction of the highest change of the gradient

$$S(\mathcal{W})^2 = \lim_{\xi \to 0} \sup_{\|W_k - \tilde{W}_k\|_F \leqslant \xi} \frac{\sum_{k=1}^L \|\nabla_k R^L(\mathcal{W}) - \nabla_k R^L(\tilde{\mathcal{W}})\|_F^2}{\sum_{k=1}^L \|W_k - \tilde{W}_k\|_F^2} . \tag{7}$$

Lower bounds are proven by considering well-chosen directions $\tilde{W}_k$, for instance $\tilde{W}_k = (1 + \xi\beta_k)W_k$ for the first lower bound. The upper bound is proven by constructing a specific minimizer and bounding its sharpness. The first lower bound shows that the sharpness of minimizers can be arbitrarily high if one of the matrices has a low-enough norm. More precisely, take any minimizer $\mathcal{W} = (W_1, \ldots W_L)$ and consider $\mathcal{W}^C = (CW_1, W_2/C, W_3, \ldots, W_L)$, for some $C > 0$. Then $\mathcal{W}^C$ is still a minimizer, and

$$S(\mathcal{W}^C) \geqslant \frac{2a \|w^\star\|_2^{2 - \frac{1}{L}} \|p\|^{\frac{1}{L}}}{\|W_2/C\|_F} = \frac{2a \|w^\star\|_2^{2 - \frac{1}{L}} \|p\|^{\frac{1}{L}} C}{\|W_2\|_F} \xrightarrow{C \to \infty} \infty.$$

The fact that a reparameterization of the network can lead to arbitrarily high sharpness is consistent with a similar result by Dinh et al. (2017) for two-layer ReLU networks.

Note that the first lower bound is arbitrarily small for minimizers such that the norms $\|W_k\|_F$ are large. On the contrary, the second lower bound is uniform and asymptotically matches the upper bound when $L \to \infty$: we have $S_{\min} \sim 2\|w^\star\|_2^2 La$.

As already noted by Mulayoff and Michaeli (2020), the intuition behind the linear scaling of the bound with depth can be seen from a one-dimensional example: take $f(x_1, \ldots, x_L) = \prod_{k=1}^{L} x_i$. Then an easy computation shows that the sharpness of $f$ at $(1, \ldots, 1)$ is equal to $L - 1$. This showcases a simple example where the output of $f$ is constant with $L$ while its sharpness grows linearly with $L$.

## 4 Analysis of gradient flow from small initialization

In this section, we characterize the structure of the minimizer found by gradient flow starting from a small-scale initialization. The proof is inspired by the one of Ji and Telgarsky (2020) for linearly-separable classification, with a finer analysis due to the finiteness of minimizers in our setting.

We consider the model (3) with $d_L = 1$ and $p = 1$. Denoting by $R_0$ the empirical risk when the weight matrices are equal to zero, we can state our assumption on the initialization.

$(A_1)$ The initialization satisfies that $R^L(\mathcal{W}(0)) \leqslant R_0$ and $\nabla R^L(\mathcal{W}(0)) \neq 0$.

It is satisfied for instance if one of the weight matrices $W_k$ is equal to zero at initialization while the others have i.i.d. Gaussian entries, so that $R^L(\mathcal{W}(0)) = R_0$ and $\nabla_k R^L(\mathcal{W}(0)) \neq 0$ (almost surely).

Linear networks trained by gradient flow possess the following remarkable property (Arora et al., 2018) that shall be useful in the remainder.

**Lemma 1.** *For any time $t \geqslant 0$ and any $k \in \{1, \ldots, L-1\}$,*

$$W_{k+1}^\top(t)W_{k+1}(t) - W_{k+1}^\top(0)W_{k+1}(0) = W_k(t)W_k^\top(t) - W_k(0)W_k^\top(0).$$

Define now

$$\varepsilon := 3 \max_{1 \leqslant k \leqslant L} \|W_k(0)\|_F^2 + 2 \sum_{k=1}^{L-1} \|W_k(0)W_k^\top(0) - W_{k+1}^\top(0)W_{k+1}(0)\|_2.$$

Note that $\varepsilon$ only depends on the initialization, and can be made arbitrarily small by scaling down the initialization. The following key lemma connects throughout training three key quantities to $\varepsilon$.

**Lemma 2.** *The parameters following gradient flow satisfy for any $t \geqslant 0$ that*

- *for $k \in \{1, \ldots, L\}$,* $\quad \|W_k(t)\|_F^2 - \|W_k(t)\|_2^2 \leqslant \varepsilon,$

- *for $j, k \in \{1, \ldots, L\}$,* $\quad |\sigma_k^2(t) - \sigma_j^2(t)| \leqslant \varepsilon,$

- *for $k \in \{1, \ldots, L-1\}$,* $\quad \langle v_{k+1}(t), u_k(t)\rangle^2 \geqslant 1 - \dfrac{\varepsilon}{\sigma_{k+1}^2(t)}.$

The first identity of the Lemma bounds the sum of the squared singular values of $W_k(t)$, except the largest one. In other words, it quantifies how close $W_k(t)$ is to the rank-one approximation given by the first term in its singular value decomposition. The second statement bounds the distance between the spectral norms of any two weight matrices. The last bound quantifies the alignment between the first left singular vector of $W_k(t)$ and the first right singular vector of $W_{k+1}(t)$. In particular, if $\varepsilon$ is small and $\sigma_{k+1}^2(t)$ is of order 1, then $v_{k+1}(t)$ and $u_k(t)$ are nearly aligned.

We next use this Lemma to show that the neural network satisfies a Polyak-Łojasiewicz (PL) condition, which is one of the main tools to study non-convex optimization dynamics (Rebjock and Boumal, 2023). A well-known result, recalled in Appendix A for completeness, shows that this implies exponential convergence of the gradient flow to a minimizer $\mathcal{W}^{\mathrm{sI}}$ of the empirical risk.

**Theorem 3.** *Under Assumption $(A_1)$, the network satisfies the PL condition for $t \geqslant 1$, in the sense that there exists some $\mu > 0$ such that, for $t \geqslant 1$,*

$$\sum_{k=1}^{L} \left\|\nabla_k R^L(\mathcal{W}(t))\right\|_F^2 \geqslant \mu(R^L(\mathcal{W}(t)) - R_{\min}).$$

The proof leverages the structure of the gradient of the risk with respect to the first weight matrix, which relies on the linearity of the neural network and the fact that we consider a univariate output. More precisely, recall that, by (5),

$$\nabla_1 R^L(\mathcal{W}(t)) = \underbrace{(W_L(t)\ldots W_2(t))^\top}_{d_1 \times 1} \underbrace{\nabla R^1(w_{\mathrm{prod}}(t))^\top}_{1 \times d_0}.$$

Therefore the Frobenius norm of the gradient decomposes as the product of two vector norms

$$\left\|\nabla_1 R^L(\mathcal{W}(t))\right\|_F^2 = \|W_L(t)\ldots W_2(t)\|_2^2 \|\nabla R^1(w_{\mathrm{prod}}(t))\|_2^2$$
$$\geqslant 4\lambda \|W_L(t)\ldots W_2(t)\|_2^2 (R^L(\mathcal{W}(t)) - R_{\min}),$$

where the lower bound unfolds from a straightforward computation. The delicate step is to lower bound $\|W_L(t)\ldots W_2(t)\|_2$, which we approach by distinguishing depending on the magnitude of $\sigma_1(t) = \|W_1(t)\|_2$. If $\sigma_1(t)$ is large, we use Lemma 2 to deduce both that all $\sigma_k(t)$ are large and then that the first singular vectors of successive weight matrices are aligned. This implies that the product of weight matrices has a large norm. To analyze the case where $\sigma_1(t)$ is small, we use Assumption $(A_1)$ to bound away $R^L(\mathcal{W}(t))$ from $R_0$ for $t \geqslant 1$, and therefore $w_{\mathrm{prod}}(t)$ from 0. The fact that $w_{\mathrm{prod}}(t)$ cannot be too close to 0 while $\sigma_1(t)$ is small implies that $\|W_L(t)\ldots W_2(t)\|_2$ is large. All in all, this allows us to lower bound $\|W_L(t)\ldots W_2(t)\|_2$, and the PL condition follows.

To characterize the weights at the end of the training, we make the following assumption.

$(A_2)$ The data covariance matrix $\hat{\Sigma}$ is full rank, and we have $\varepsilon \leqslant 1$, $\|w^\star\|_2 \geqslant 1$, and $32L\sqrt{\varepsilon} \leqslant 1$.

The first statement ensures unicity of the minimizer $w^\star$ of $R^1$, and thus, given that the risk goes to 0, we have $w_{\mathrm{prod}} \to w^\star$. The last condition means that the initialization has to be scaled down as the depth increases, so that $\varepsilon = \mathcal{O}(1/L^2)$. Intermediates conditions are technical. We can then show the following corollary.

**Corollary 1.** *Under Assumptions $(A_1)$–$(A_2)$, there exists $T \geqslant 1$, such that, for all $t \geqslant T$ and $k \in \{1,\ldots,L\}$,*

$$\left(\frac{\|w^\star\|_2}{2}\right)^{1/L} \leqslant \sigma_k(t) \leqslant \left(2\|w^\star\|_2\right)^{1/L}.$$

Together with Lemma 2, this result gives a precise description of the structure of the weights at the end of the gradient flow trajectory. Up to the small factor $\varepsilon$, the weights are rank-one matrices, with equal norms and aligned singular vectors. Since the product of weights aligns with $w^\star$, this means that the first right singular vector of $W_1$ has to align with $w^\star$, and then the weight matrices align with their neighbors in order to propagate the signal in the network.

Combining this specific structure of the weights with the variational characterization (7) of the sharpness and the explicit formulas (5)–(6) for the gradients, we derive the following upper bound on the sharpness of the found minimizer.

**Corollary 2.** *Under Assumptions $(A_1)$–$(A_2)$, the following bounds on the sharpness of the minimizer $\mathcal{W}^{\mathrm{SI}}$ hold:*

$$1 \leqslant \frac{S(\mathcal{W}^{\mathrm{SI}})}{S_{\min}} \leqslant 4\frac{\Lambda}{\lambda}.$$

This result shows that the sharpness of the minimizer is close to $S_{\min}$ in the sense that their ratio is bounded by a constant times the condition number of $\hat{\Sigma}$. For example, in the case of white data ($\hat{\Sigma} = I$), we obtain that $S(\mathcal{W}^{\mathrm{SI}}) \leqslant 4S_{\min}$.

## 5 Analysis of gradient flow from residual initialization

We now study the case of residual initialization. We consider a linear network of the form (3) with

$$W_k(t) = I + \frac{s}{\sqrt{Ld}}N_k + \frac{1}{L}\theta_k(t),$$

where each matrix is a $d \times d$ matrix and the $N_k$ are filled with i.i.d. Gaussian entries $\mathcal{N}(0,1)$. We refer to Section 1 for a discussion of the scaling factor in front of $N_k$. Following the standard initialization of residual networks, we assume that the $\theta_k$ are initialized to zero. Note that the scaling factor $1/L$ in front of the $\theta_k$ has no impact on the dynamics; it is convenient for exposition and computations, since we show that, with this scaling factor, the $\theta_k$ are of order $\mathcal{O}(1)$ after training.

Before stating the main result of this section on the convergence of the gradient flow, recall that $p \in \mathbb{R}^d$ is a fixed vector appended at the end of the residual network to project its output back in $\mathbb{R}$.

**Theorem 4.** *There exist $C_1, \ldots, C_5 > 0$ depending only on $s$ such that, if $L \geqslant C_1$ and $d \geqslant C_2$, then, with probability at least*

$$1 - 16 \exp(-C_3 d),$$

*if*

$$R^L(\mathcal{W}(0)) - R_{\min} \leqslant \frac{C_4 \lambda^2 \|p\|_2^2}{\Lambda},$$

*the gradient flow dynamics* (4) *converge to a global minimizer $\mathcal{W}^{\mathrm{RI}}$ of the risk. Furthermore, the minimizer $\mathcal{W}^{\mathrm{RI}}$ satisfies*

$$W_k^{\mathrm{RI}} = I + \frac{s}{\sqrt{Ld}} N_k + \frac{1}{L} \theta_k^{\mathrm{RI}} \quad \text{with} \quad \|\theta_k^{\mathrm{RI}}\|_F \leqslant C_5, \quad 1 \leqslant k \leqslant L. \tag{8}$$

To our knowledge, Theorem 4 is the first result showing convergence of gradient flow for standard Gaussian initialization of residual networks without assuming overparameterization. The main requirement is that the loss at initialization be not too large, as is standard in the literature analyzing gradient flow for deep linear residual networks (Bartlett et al., 2018; Arora et al., 2019a; Zou et al., 2020; Sander et al., 2022). Note that our bound on the loss at initialization does not depend on the width $d$, depth $L$, or sample size $n$. We emphasize that the same proof holds for multivariate regression, in the absence of the projection vector $p$. We focus here on univariate regression to connect the result with the analysis of sharpness for univariate regression in Section 3. Details on adaptation to multivariate regression are given in Appendix B.7. Finally, the precise dependence of $C_1$ to $C_5$ on $s$ can be found in the proof.

The proof is a refinement of the analysis for identity initialization of residual networks (Zou et al., 2020; Sander et al., 2022). From the expression of the gradients (5), we can show that

$$4\Lambda \|p\|_2^2 \|\Pi_{L:k+1}\|_2^2 \|\Pi_{k-1:1}\|_2^2 (R(\mathcal{W}) - R_{\min})$$
$$\geqslant \|\nabla_k R^L(\mathcal{W})\|_F^2 \geqslant 4\lambda \|p\|_2^2 \sigma_{\min}^2(\Pi_{L:k+1}) \sigma_{\min}^2(\Pi_{k-1:1})(R(\mathcal{W}) - R_{\min}),$$

with $\Pi_{L:k} := W_L \ldots W_k$ and $\Pi_{k:1} := W_k \ldots W_1$. Letting

$$t^* = \inf \left\{ t \in \mathbb{R}_+, \exists k \in \{1, \ldots, L\}, \|\theta_k(t)\|_F > C_5 \right\},$$

the crucial step is to lower bound $\sigma_{\min}^2(\Pi_{L:k+1})$ and $\sigma_{\min}^2(\Pi_{k-1:1})$ uniformly for $t \in [0, t^\star]$, in order to get a PL condition valid for $t \in [0, t^\star]$. Then, the condition on the loss at initialization is used to prove that $t^\star = \infty$, thereby the PL condition holds for all $t \geqslant 0$. We deduce both convergence and the bound on the norm of $\theta_k^{\mathrm{RI}}$. The lower bound on $\sigma_{\min}^2(\Pi_{L:k+1})$ and $\sigma_{\min}^2(\Pi_{k-1:1})$ is straightforward in the case of an identity initialization. In the case of Gaussian initialization, the proof is more intricate, and leverages the following high probability bounds on the singular values of residual networks.

**Lemma 3.** *There exist $C_1, \ldots, C_4 > 0$ depending only on $s$ such that, if*

$$L \geqslant C_1, \quad d \geqslant C_2, \quad u \in [C_3, C_4 L^{1/4}],$$

*then, with probability at least*

$$1 - 8 \exp\left(-\frac{du^2}{32s^2}\right),$$

*it holds for all $\theta$ such that $\max_{1 \leqslant k \leqslant L} \|\theta_k\|_2 \leqslant \frac{1}{64} \exp(-2s^2 - 4u)$ and all $k \in \{1, \ldots, L\}$ that*

$$\left\| \left( I + \frac{s}{\sqrt{Ld}} N_k + \frac{1}{L} \theta_k \right) \ldots \left( I + \frac{s}{\sqrt{Ld}} N_1 + \frac{1}{L} \theta_1 \right) \right\|_2 \leqslant 4 \exp\left(\frac{s^2}{2} + u\right),$$

*and*

$$\sigma_{\min}\left( \left( I + \frac{s}{\sqrt{Ld}} N_k + \frac{1}{L} \theta_k \right) \ldots \left( I + \frac{s}{\sqrt{Ld}} N_1 + \frac{1}{L} \theta_1 \right) \right) \geqslant \frac{1}{4} \exp\left(-\frac{2s^2}{d} - u\right).$$

The proof of this result goes in three steps. We first study the evolution of the norm of the activations across the layers of the residual network when $\theta = 0$, and prove a high-probability bound by leveraging concentration inequalities for Gaussian and $\chi^2$ distributions. Then an $\varepsilon$-net argument allows to bound the singular values. Finally, the extension to $\theta$ in a ball around 0 is done via a perturbation analysis. The proof technique is related to previous works (Marion et al., 2022; Zhang et al., 2022), but provides a crisper and sounder bound. More precisely, Marion et al. (2022) show a bound on the norm of the activations with a probability of failure that decays polynomially with the width $d$, which is not sufficient to apply the $\varepsilon$-net argument that requires an exponentially decreasing probability of failure. As for Zhang et al. (2022), they provide a similar bound with the purpose of showing convergence of wide residual networks, however with a less sharp probability of failure that increases polynomially with depth.[1] Finally, as previously, the dependence of $C_1$ to $C_4$ on $s$ can be found in the proof.

The characterization (8) of the minimizer in Theorem 4 allows to bound its sharpness, as made precise by the following corollary. It holds under the same assumptions and high-probability bound as the conclusion of Theorem 4.

**Corollary 3.** *Under the assumptions of Theorem 4, and if the data covariance matrix $\hat{\Sigma}$ is full rank, there exists $C > 0$ depending only on $s$ such that the following bounds on the sharpness of the minimizer $\mathcal{W}^{\mathrm{RI}}$ hold:*

$$1 \leqslant \frac{S(\mathcal{W}^{\mathrm{RI}})}{S_{\min}} \leqslant C \frac{\Lambda}{\lambda}\,.$$

As for Corollary 2, the proof relies on the fact that the norms of the weight matrices are close to each other and to the smallest possible norm to minimize the risk. This result shows again an implicit regularization towards a low-sharpness minimizer. Experimental illustration connecting the result with gradient descent with non-vanishing learning rate is provided in Appendix C.

## 6 Conclusion

This paper studies dynamics of gradient flow for deep linear networks on a regression task. For small-scale initialization, we prove that the learned weight matrices are approximately rank-one and that their singular vectors align. For residual initialization, convergence of the gradient flow for a Gaussian initialization is proven. In both cases, we obtain that the sharpness of the solution found by gradient flow is close to the smallest sharpness among all minimizers. Interesting next steps include studying the dynamics at any initialization scale, for non-vanishing learning rates, as well as extension to non-linear networks. We refer to Appendix C for additional comments and preliminary experimental results regarding possible extensions.

## Acknowledgments

P.M. acknowledges support of Google through a Google PhD Fellowship. Authors thank Matus Telgarsky for insightful discussions, and Eloïse Berthier, Guillaume Dalle, Benjamin Dupuis, as well as anonymous NeurIPS reviewers, for thoughtful proofreading and remarks. An improvement to a proof was also made possible by a tweet of Gabriel Peyré.

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

# Appendix

**Organization of the Appendix.** Appendix A presents some useful preliminary lemmas. The proofs of the results of the main paper are presented in Appendix B, while additional plots, discussion, and experimental details are given in Appendix C. Finally, Appendix D discusses some additional related work.

## A    Technical lemmas

**Lemma 4.** *For $\alpha > 0$ and $x \in [0, 1/2]$,*

$$(1 - x)^\alpha \geqslant 1 - 2\alpha x \,.$$

*For $\alpha > 0$ and $x > 0$ such that $\alpha x \leqslant 1$,*

$$(1 + x)^\alpha \leqslant 1 + 2\alpha x \,.$$

*Proof.* Regarding the first inequality of the Lemma, we have

$$(1 - x)^\alpha = \exp(\alpha \log(1 - x)) \geqslant \exp(\alpha(-2x)) \geqslant 1 - 2\alpha x \,,$$

where the first inequality holds for $x \in [0, 1/2]$. The second inequality of the Lemma is proven by

$$(1 + x)^\alpha = \exp(\alpha \log(1 + x)) \leqslant \exp(\alpha x) \leqslant 1 + 2\alpha x \,,$$

where the second inequality holds when $\alpha x \leqslant 1$. $\qquad\square$

**Lemma 5.** *There exists an absolute constant $C > 0$ such that, for $L \geqslant C$ and $x \geqslant 1$,*

$$L \exp(-\sqrt{L} x) \leqslant 4 \exp(-x) \,.$$

*Proof.* For any $x \in \mathbb{R}$,

$$L \exp(-\sqrt{L} x) \leqslant 4 \exp(-x) \Leftrightarrow \exp((\sqrt{L} - 1)x) \geqslant \frac{L}{4} \,.$$

Then, for $L \geqslant 1$ and $x \geqslant 1$, we have

$$
\begin{aligned}
\exp((\sqrt{L} - 1)x) &\geqslant 1 + (\sqrt{L} - 1)x + \frac{1}{2}(\sqrt{L} - 1)^2 x^2 \\
&= 1 + (\sqrt{L} - 1)x + \frac{1}{2}(L + 1 - 2\sqrt{L})x^2 \\
&\geqslant (\sqrt{L} - 1)x + \frac{L}{4} + \frac{1}{2}(\frac{L}{2} + 1 - 2\sqrt{L})x^2
\end{aligned}
$$

For $L$ large enough, $\frac{L}{2} + 1 - 2\sqrt{L} \geqslant 0$. Thus, since $x \geqslant 1$,

$$
\begin{aligned}
\exp((\sqrt{L} - 1)x) &\geqslant (\sqrt{L} - 1)x + \frac{L}{4} + \frac{1}{2}(\frac{L}{2} + 1 - 2\sqrt{L})x \\
&= \frac{L}{4} + (\frac{L}{4} - \frac{1}{2})x \\
&\geqslant \frac{L}{4} \,,
\end{aligned}
$$

where the last inequality holds for $L$ large enough. This concludes the proof. $\qquad\square$

**Lemma 6.** *Let $h \in \mathbb{R}^d$, $N \in \mathbb{R}^{d \times d}$ with i.i.d. standard Gaussian entries, and*

$$Y_1 = \frac{\|Nh\|_2^2}{\|h\|_2^2} \,, \quad Y_2 = \frac{h^\top N h}{\|h\|_2^2} \,,$$

*Then*

$$Y_1 \sim \chi^2(d) \quad \text{and} \quad Y_2 \sim \mathcal{N}(0, 1) \,.$$

*Proof.* We have, for $i \in \{1, \ldots, d\}$,

$$(Nh)_i = \sum_{j=1}^{d} N_{ij} h_j \,.$$

By independence of the $(N_{ij})_{1 \leqslant j \leqslant d}$, we deduce that $(Nh)_i$ follows a $\mathcal{N}(0, \|h\|^2)$ distribution. Furthermore, by independence of the rows of $N$, the $(Nh)_i$ are independent. Thus

$$Y_1 = \frac{1}{\|h\|^2} \sum_{i=1}^{d} (Nh)_i^2$$

follows a $\chi^2(d)$ distribution. Moving on to $Y_2$, we have

$$Y_2 = \frac{1}{\|h\|^2} \sum_{i,j=1}^{d} N_{ij} h_i h_j \,.$$

Thus $Y_2$ follows a centered Gaussian distribution, and by independence of the $(N_{ij})_{1 \leqslant i,j \leqslant d}$, its variance is equal to

$$\frac{1}{\|h\|^4} \sum_{i,j=1}^{d} h_i^2 h_j^2 = 1 \,,$$

which concludes the proof. $\qquad \square$

The next lemma shows that the PL condition implies exponential convergence of the gradient flow. It is a well-known fact (see, e.g., [Rebjock and Boumal, 2023](), for an overview of similar conditions), proved here for completeness.

**Lemma 7.** *Let $f : \mathbb{R}^D \to \mathbb{R}$ be a differentiable function lower bounded by $f_{\min} \in \mathbb{R}$, and consider the gradient flow dynamics*

$$\frac{dx}{dt} = -\nabla f(x(t)) \,.$$

*If $f$ satisfies the Polyak-Łojasiewicz inequality for $t \geqslant 0$*

$$\|\nabla f(x(t))\|_2^2 \geqslant \mu(f(x(t)) - f_{\min}) \,,$$

*then $x(t)$ converges to a global minimizer $x_\infty$, and, for $t \geqslant 0$,*

$$f(x(t)) - f_{\min} \leqslant (f(x(0)) - f_{\min}) e^{-\mu t} \,.$$

*Proof.* By the chain rule,

$$\frac{d}{dt} f(x(t)) = \left\langle \nabla f(x(t)), \frac{dx}{dt} \right\rangle = -\|\nabla f(x(t))\|_2^2 \,.$$

Plugging in the Polyak-Łojasiewicz inequality,

$$\frac{d}{dt} f(x(t)) \leqslant -\mu(f(x(t)) - f_{\min}) \,.$$

Thus

$$\frac{d}{dt} (f(x(t)) - f_{\min}) \leqslant -\mu(f(x(t)) - f_{\min}) \,.$$

To solve this differential inequality, one can for instance use the comparison theorem ([Michel Petrovitch, 1901]()), which states that $f(x(t)) - f_{\min}$ is smaller that the solution $g$ of the initial value problem

$$g(0) = f(x(0)) - f_{\min} \,, \quad \frac{d}{dt} g(t) = -\mu g(t) \,.$$

This shows that

$$f(x(t)) - f_{\min} \leqslant (f(x(0)) - f_{\min}) e^{-\mu t} \,.$$

To show convergence of the iterates, let $F(t) = \sqrt{\mu}\int_0^t \|\nabla f(x(s))\|ds + 2\sqrt{f(x(t)) - f_{\min}}$. We have by the Polyak-Łojasiewicz inequality

$$F'(t) = \|\nabla f(x(t))\|\left(\sqrt{\mu} - \frac{\|\nabla f(x(t))\|}{f(x(t)) - f_{\min}}\right) \leqslant 0.$$

Then

$$0 \leqslant \int_0^t \left\|\frac{dx}{ds}\right\|ds = \int_0^t \|\nabla f(x(s))\|ds \leqslant \frac{F(t)}{\sqrt{\mu}} \leqslant \frac{F(0)}{\sqrt{\mu}},$$

showing that $x(t)$ converges. $\qquad\square$

**Lemma 8.** *With the notation introduced in Section 2, the following identities holds*

$$R^L(\mathcal{W}) = R_{\min} + \frac{1}{n}\|X(w^\star - w_{\mathrm{prod}})\|_2^2,$$

*and*

$$4\lambda(R^L(\mathcal{W}) - R_{\min}) \leqslant \|\nabla R^1(w_{\mathrm{prod}})\|_2^2 \leqslant 4\Lambda(R^L(\mathcal{W}) - R_{\min}).$$

*Proof.* We have

$$
\begin{aligned}
R^L(\mathcal{W}) &= \frac{1}{n}\|y - Xw_{\mathrm{prod}}\|_2^2 \\
&= \frac{1}{n}\|y - Xw^\star + X(w^\star - w_{\mathrm{prod}})\|_2^2 \\
&= \frac{1}{n}\|y - Xw^\star\|_2 + \frac{1}{n}\|X(w^\star - w_{\mathrm{prod}})\|_2^2 + \frac{2}{n}\langle y - Xw^\star, X(w^\star - w_{\mathrm{prod}})\rangle \\
&= R_{\min} + \frac{1}{n}\|X(w^\star - w_{\mathrm{prod}})\|_2^2 + \frac{2}{n}\langle X^\top(y - Xw^\star), w^\star - w_{\mathrm{prod}}\rangle,
\end{aligned}
$$

where the scalar product is equal to zero because $\nabla R^1(w^\star) = -\frac{2}{n}X^\top(y - Xw^\star) = 0$. This gives the first identity of the Lemma. Next, denoting $\pi(w_{\mathrm{prod}})$ the projection of $w_{\mathrm{prod}}$ on the orthogonal subspace to the kernel of $X$, we have

$$
\begin{aligned}
R^L(\mathcal{W}) - R_{\min} &= \frac{1}{n}\|X(w^\star - \pi(w_{\mathrm{prod}}))\|_2^2 \\
&= \frac{1}{n}(w^\star - \pi(w_{\mathrm{prod}}))^\top X^\top X(w^\star - \pi(w_{\mathrm{prod}})) \\
&\leqslant \frac{1}{n}\|w^\star - \pi(w_{\mathrm{prod}})\|_2\|X^\top X(w^\star - \pi(w_{\mathrm{prod}}))\|_2.
\end{aligned}
$$

By the formula (6) for the gradient of $R^1$,

$$
\begin{aligned}
\|\nabla R^1(w_{\mathrm{prod}})\|_2 &= \frac{2}{n}\|X^\top X(w^\star - w_{\mathrm{prod}})\|_2 \\
&= \frac{2}{n}\|X^\top X(w^\star - \pi(w_{\mathrm{prod}}))\|_2 \geqslant 2\lambda\|w^\star - \pi(w_{\mathrm{prod}})\|_2,
\end{aligned}
$$

where the last lower bound holds because both $w^\star$ and $\pi(w_{\mathrm{prod}})$ are in the orthogonal to the kernel of $X$. Plugging in the formula above, we obtain that

$$R^L(\mathcal{W}) - R_{\min} \leqslant \frac{1}{2\lambda}\|\nabla R^1(w_{\mathrm{prod}})\|_2 \cdot \frac{1}{2}\|\nabla R^1(w_{\mathrm{prod}})\|_2 = \frac{1}{4\lambda}\|\nabla R^1(w_{\mathrm{prod}})\|_2^2.$$

Finally, to obtain the upper bound on the gradient, note that

$$
\begin{aligned}
\|\nabla R^1(w_{\mathrm{prod}})\|_2^2 &= \frac{4}{n^2}\|X^\top X(w^\star - w_{\mathrm{prod}})\|_2^2 \\
&= \frac{4}{n^2}(w^\star - w_{\mathrm{prod}})^\top X^\top X X^\top X(w^\star - w_{\mathrm{prod}}) \\
&\leqslant \frac{4\Lambda}{n}(w^\star - w_{\mathrm{prod}})^\top X^\top X(w^\star - w_{\mathrm{prod}}) \\
&= 4\Lambda(R^L(\mathcal{W}) - R_{\min}).
\end{aligned}
$$

$\qquad\square$

**Lemma 9.** *Take $W_1, \ldots W_L \in \mathbb{R}^{d \times d}$ such that, for all $k \in \{1, \ldots, L\}$,*

$$\|W_k \ldots W_1\|_2 \leqslant M\,,$$

*and*

$$\sigma_{\min}(W_k \ldots W_1) \geqslant m\,,$$

*where $M \geqslant 1$ and $m \in (0,1)$. Then, for all $\theta$ such that $\max_{1 \leqslant k \leqslant L} \|\theta_k\|_2 \leqslant \frac{m^2}{4M^2}$, letting $\tilde{W}_k = W_k + \frac{\theta_k}{L}$, we have*

$$\|\tilde{W}_k \ldots \tilde{W}_1\|_2 \leqslant 2M\,,$$

*and*

$$\sigma_{\min}(\tilde{W}_k \ldots \tilde{W}_1) \geqslant \frac{m}{2}\,.$$

*Proof.* First note that the assumptions imply that, for any $1 \leqslant j < k \leqslant L$,

$$\|W_k \ldots W_{j+1}\|_2 \leqslant \frac{\|W_k \ldots W_{j+1} W_j \ldots W_1\|_2}{\sigma_{\min}(W_j \ldots W_1)} = \frac{\|W_k \ldots W_1\|_2}{\sigma_{\min}(W_j \ldots W_1)} \leqslant \frac{M}{m}\,. \tag{9}$$

It shall come in handy to extend this formula to the case where $j = k$, where we define the empty matrix product to be equal to the identity matrix, which has an operator norm of $1 \leqslant \frac{M}{m}$.

Now, take any $\theta$ as in the Lemma and any $h_0 \in \mathbb{R}^d$. Let, for $k \geqslant 0$,

$$h_k = W_k \ldots W_1 h_0\,,$$

and

$$\tilde{h}_k = \tilde{W}_k \ldots \tilde{W}_1 h_0\,.$$

Then

$$h_k = W_k h_{k-1}$$

and

$$\tilde{h}_k = \left(W_k + \frac{\theta_k}{L}\right)\tilde{h}_{k-1}\,.$$

Thus

$$\tilde{h}_k - h_k = \frac{\theta_k}{L}\tilde{h}_{k-1} + W_k(\tilde{h}_{k-1} - h_{k-1})\,.$$

Since $\tilde{h}_0 - h_0 = 0$, we get by recurrence that, for $k \geqslant 1$,

$$\tilde{h}_k - h_k = \sum_{j=1}^{k} W_k \ldots W_{j+1} \frac{\theta_j}{L}\tilde{h}_{j-1}\,. \tag{10}$$

From there, let us prove by recurrence that

$$\|\tilde{h}_k\|_2 \leqslant 2M\|\tilde{h}_0\|_2\,. \tag{11}$$

This equation holds for $k = 0$ since $M \geqslant 1$. Next, assume that it holds up to a certain rank $k-1$, and let us prove it at rank $k$. From (10), we get that

$$\|\tilde{h}_k - h_k\|_2 \leqslant \frac{1}{L}\sum_{j=1}^{k} \|W_k \ldots W_{j+1}\|_2 \|\theta_j\|_2 \|\tilde{h}_{j-1}\|_2\,. \tag{12}$$

Since $M \geqslant 1$ and $m < 1$, the bound on $\|\theta_j\|_2$ from the assumptions of the Lemma implies in particular that $\|\theta_j\|_2 \leqslant \frac{m}{2M}$. Utilizing this, as well as (9) and the recurrence hypothesis (11) up to rank $k-1$, we get

$$\|\tilde{h}_k - h_k\|_2 \leqslant \frac{1}{L}\sum_{j=1}^{k} \frac{M}{m} \cdot \frac{m}{2M} \cdot 2M\|\tilde{h}_0\|_2 \leqslant M\|\tilde{h}_0\|_2\,.$$

Then

$$\|\tilde{h}_k\|_2 \leqslant \|h_k\|_2 + \|\tilde{h}_k - h_k\|_2 \leqslant \|W_k \ldots W_1\|_2 \|h_0\|_2 + M\|\tilde{h}_0\|_2 \leqslant 2M\|\tilde{h}_0\|_2\,.$$

This concludes the proof of the recurrence hypothesis at rank $k$. We therefore get that
$$\|\tilde{W}_k \ldots \tilde{W}_1\|_2 \leqslant 2M.$$
To prove the lower bound on the smallest singular value of $\tilde{W}_k \ldots \tilde{W}_1$, observe that
$$\|\tilde{h}_k\|_2 \geqslant \|h_k\|_2 - \|\tilde{h}_k - h_k\|_2$$
$$\geqslant \sigma_{\min}(W_k \ldots W_1)\|h_0\|_2 - \frac{1}{L}\sum_{j=1}^{k}\|W_k \ldots W_{j+1}\|_2\|\theta_j\|_2\|\tilde{h}_{j-1}\|_2,$$
by (12). Finally, by (9), the bound on $\|\theta_j\|_2$ from the assumptions, and the upper bound we just proved on $\|\tilde{h}_{j-1}\|_2$, we have
$$\|\tilde{h}_k\|_2 \geqslant m\|h_0\|_2 - \frac{1}{L}\sum_{j=1}^{k}\frac{M}{m} \cdot \frac{m^2}{4M^2} \cdot 2M\|h_0\|_2 \geqslant \frac{m}{2}\|h_0\|_2.$$
This concludes the proof. $\qquad\square$

# B  Proofs

## B.1  Proof of Theorem 2

For a twice continuously differentiable function $f : \mathbb{R}^D \to \mathbb{R}$, the largest eigenvalue $S$ of its Hessian $H(f) : \mathbb{R}^D \to \mathbb{R}^{D \times D}$ at some $x \in \mathbb{R}^D$ admits the variational characterization
$$S(x) = \lim_{\xi \to 0} \sup_{\|x - \tilde{x}\| \leqslant \xi} \frac{\|\nabla f(x) - \nabla f(\tilde{x})\|_2}{\|x - \tilde{x}\|_2}.$$
In our case, the parameters are a set of matrices and the formula above translates into
$$S(\mathcal{W})^2 = \lim_{\xi \to 0} \sup_{\|W_k - \tilde{W}_k\|_F \leqslant \xi} \frac{\sum_{k=1}^{L}\|\nabla_k R^L(\mathcal{W}) - \nabla_k R^L(\tilde{\mathcal{W}})\|_F^2}{\sum_{k=1}^{L}\|W_k - \tilde{W}_k\|_F^2}. \tag{13}$$
We now take $\mathcal{W}$ to be an arbitrary minimizer of $R^L$. To obtain the lower bounds, consider for $\xi \leqslant 1$
$$\tilde{W}_k(\xi) = W_k + \xi M_k,$$
where the $M_k \in \mathbb{R}^{d_k \times d_{k-1}}$ are parameters that will be chosen later (depending on the $W_k$ but not on $\xi$). Then
$$S(\mathcal{W})^2 \geqslant \lim_{\xi \to 0} \frac{\sum_{k=1}^{L}\|\nabla_k R^L(\mathcal{W}) - \nabla_k R^L(\tilde{\mathcal{W}}(\xi))\|_F^2}{\xi^2 \sum_{k=1}^{L}\|M_k\|_F^2}, \tag{14}$$
To alleviate notations, we drop the dependence of $\tilde{W}_k$ on $\xi$. Recall that, for any parameters $\mathcal{W}^0$,
$$\nabla_k R^L(\mathcal{W}^0) = W_{k+1}^{0\top} \ldots W_L^{0\top} p \nabla R^1(w_{\text{prod}}^0)^\top W_1^{0\top} \ldots W_{k-1}^{0\top},$$
with
$$w_{\text{prod}}^0 = W_1^{0\top} \ldots W_L^{0\top} p \quad \text{and} \quad \nabla R^1(w_{\text{prod}}^0) = -\frac{2}{n}X^\top X(w^\star - w_{\text{prod}}^0).$$
For minimizers of the empirical risk, $\nabla_k R^L(\mathcal{W}) = 0$, so
$$\Delta_k := \nabla_k R^L(\mathcal{W}) - \nabla_k R^L(\tilde{\mathcal{W}}) = -\tilde{W}_{k+1}^\top \ldots \tilde{W}_L^\top p \nabla R^1(\tilde{w}_{\text{prod}})^\top \tilde{W}_1^\top \ldots \tilde{W}_{k-1}^\top. \tag{15}$$
Minimizers of the empirical risk also satisfy that $X^\top X w_{\text{prod}} = X^\top X w^\star$. Thus, by adding and subtracting differences,
$$\nabla R^1(\tilde{w}_{\text{prod}}) = \frac{2}{n}X^\top X(\tilde{w}_{\text{prod}} - w_{\text{prod}})$$
$$= \frac{2}{n}X^\top X\Big(\sum_{k=1}^{L}\tilde{W}_1^\top \ldots \tilde{W}_{k-1}^\top(\tilde{W}_k^\top - W_k^\top)W_{k+1}^\top \ldots W_L^\top p\Big)$$
$$= \frac{2}{n}\xi X^\top X\Big(\sum_{k=1}^{L}W_1^\top \ldots W_{k-1}^\top M_k^\top W_{k+1}^\top \ldots W_L^\top p\Big) + \mathcal{O}(\xi^2). \tag{16}$$

Here, and in the remainder of this proof, the notation $\mathcal{O}$ is taken with respect to the limit when $\xi \to 0$, everything else being fixed. In particular, inspecting the expression above, we see that $\nabla R(\tilde{w}_{\text{prod}}) = \mathcal{O}(\xi)$, and therefore, going back to (15), that

$$\Delta_k = -W_{k+1}^\top \ldots W_L^\top p \nabla R^1(\tilde{w}_{\text{prod}})^\top W_1^\top \ldots W_{k-1}^\top + \mathcal{O}(\xi^2) =: \bar{\Delta}_k + \mathcal{O}(\xi^2). \qquad (17)$$

By the inequality of arithmetic and geometric means, and by subadditivity of the operator norm,

$$\sum_{k=1}^{L} \|\Delta_k\|_F^2 = \sum_{k=1}^{L} \|\bar{\Delta}_k\|_F^2 + \mathcal{O}(\xi^3)$$

$$\geqslant \sum_{k=1}^{L} \|\bar{\Delta}_k\|_2^2 + \mathcal{O}(\xi^3)$$

$$\geqslant L \Big( \prod_{k=1}^{L} \|\bar{\Delta}_k\|_2 \Big)^{2/L} + \mathcal{O}(\xi^3)$$

$$\geqslant L \big( \|\bar{\Delta}_L \cdots \bar{\Delta}_1\|_2 \big)^{2/L} + \mathcal{O}(\xi^3). \qquad (18)$$

By definition of $\bar{\Delta}_k$,

$$\bar{\Delta}_L \cdots \bar{\Delta}_1 = (-1)^L p \nabla R^1(\tilde{w}_{\text{prod}})^\top \underbrace{w_{\text{prod}} \nabla R^1(\tilde{w}_{\text{prod}})^\top \cdots w_{\text{prod}} \nabla R^1(\tilde{w}_{\text{prod}})^\top}_{L-1 \text{ times}}$$

$$= (-1)^L p (\nabla R^1(\tilde{w}_{\text{prod}})^\top w_{\text{prod}})^{L-1} \nabla R^1(\tilde{w}_{\text{prod}})^\top$$

$$= (-1)^L p (\nabla R^1(\tilde{w}_{\text{prod}})^\top w^\star)^{L-1} \nabla R^1(\tilde{w}_{\text{prod}})^\top,$$

where the last identity comes from the formulas $\nabla R^1(\tilde{w}_{\text{prod}}) = \frac{2}{n} X^\top X(\tilde{w}_{\text{prod}} - w_{\text{prod}})$ and $X^\top X w_{\text{prod}} = X^\top X w^\star$. Thus

$$\|\bar{\Delta}_L \cdots \bar{\Delta}_1\|_2 = (\nabla R^1(\tilde{w}_{\text{prod}})^\top w^\star)^{L-1} \|p\|_2 \|\nabla R^1(\tilde{w}_{\text{prod}})\|_2$$

$$\geqslant (\nabla R^1(\tilde{w}_{\text{prod}})^\top w^\star)^{L-1} \|p\|_2 \frac{\nabla R^1(\tilde{w}_{\text{prod}})^\top w^\star}{\|w^\star\|_2}$$

$$= \frac{(\nabla R^1(\tilde{w}_{\text{prod}})^\top w^\star)^L \|p\|_2}{\|w^\star\|_2},$$

and, by (18),

$$\sum_{k=1}^{L} \|\Delta_k\|_F^2 \geqslant L \frac{(\nabla R^1(\tilde{w}_{\text{prod}})^\top w^\star)^2 \|p\|_2^{2/L}}{\|w^\star\|_2^{2/L}} + \mathcal{O}(\xi^3). \qquad (19)$$

Coming back to (16), we have

$$w^{\star\top} \nabla R^1(\tilde{w}_{\text{prod}}) = \xi \sum_{k=1}^{L} w^{\star\top} \frac{2}{n} X^\top X W_1^\top \ldots W_{k-1}^\top M_k^\top W_{k+1}^\top \ldots W_L^\top p + \mathcal{O}(\xi^2).$$

At this point, the computations diverge for the two lower bounds we want to prove. For the first inequality, we take $M_k = \beta_k W_k$, where the $\beta_k$ are free parameters to be optimized. We have

$$w^{\star\top} \nabla R^1(\tilde{w}_{\text{prod}}) = \xi \sum_{k=1}^{L} \beta_k w^{\star\top} \frac{2}{n} X^\top X W_1^\top \ldots W_{k-1}^\top W_k^\top W_{k+1}^\top \ldots W_L^\top p + \mathcal{O}(\xi^2)$$

$$= \xi \sum_{k=1}^{L} \beta_k w^{\star\top} \frac{2}{n} X^\top X w_{\text{prod}} + \mathcal{O}(\xi^2)$$

$$= \xi \sum_{k=1}^{L} \beta_k w^{\star\top} \frac{2}{n} X^\top X w^\star + \mathcal{O}(\xi^2)$$

$$= 2\xi a \|w^\star\|^2 \sum_{j=k}^{L} \beta_k + \mathcal{O}(\xi^2).$$

Therefore, by (19),

$$\sum_{k=1}^{L} \|\Delta_k\|_F^2 \geqslant 4L\xi^2 a^2 \|w^\star\|_2^{4-\frac{2}{L}} \|p\|_2^{\frac{2}{L}} \left(\sum_{k=1}^{L} \beta_k\right)^2 + \mathcal{O}(\xi^3) \,.$$

Furthermore,

$$\sum_{k=1}^{L} \|M_k\|_F^2 = \sum_{k=1}^{L} \beta_k^2 \|W_k\|_F^2 \,.$$

By (14), we get

$$S(\mathcal{W})^2 \geqslant 4La^2 \|w^\star\|_2^{4-\frac{2}{L}} \|p\|_2^{\frac{2}{L}} \frac{\left(\sum_{k=1}^{L} \beta_k\right)^2}{\sum_{k=1}^{L} \beta_k^2 \|W_k\|_F^2} \,.$$

The first lower bound unfolds by taking $\beta_k = 1/\|W_k\|_F$.

Moving on to the second lower-bound, we now take

$$M_k = \beta_k \frac{u_k v_k^\top}{\|u_k\|_2 \|v_k\|_2} \,, \quad u_k = W_{k-1}\ldots W_1 \frac{2}{n} X^\top X w^\star \,, \quad v_k = W_{k+1}^\top \ldots W_L^\top p \,,$$

where again the $\beta_k$ are free parameters to be optimized. We therefore have

$$w^{\star\top} \nabla R^1(\tilde{w}_{\text{prod}}) = \sum_{k=1}^{L} \underbrace{w^{\star\top} \frac{2}{n} X^\top X W_1^\top \ldots W_{k-1}^\top}_{=u_k^\top} M_k^\top \underbrace{W_{k+1}^\top \ldots W_L^\top p}_{=v_k} + \mathcal{O}(\xi^2)$$

$$= \xi \sum_{k=1}^{L} \beta_k \|u_k\|_2 \|v_k\|_2 + \mathcal{O}(\xi^2)$$

$$= \xi \sum_{k=1}^{L} \beta_k \|W_{k+1}^\top \ldots W_L^\top p\|_2 \|w^{\star\top} \frac{2}{n} X^\top X W_1^\top \ldots W_{k-1}^\top\|_2 + \mathcal{O}(\xi^2)$$

$$\geqslant \xi \sum_{k=1}^{L} \beta_k \|W_{k+1}^\top \ldots W_L^\top p w^{\star\top} \frac{2}{n} X^\top X W_1^\top \ldots W_{k-1}^\top\|_2 + \mathcal{O}(\xi^2) \,,$$

where the last line unfolds from subadditivity of the operator norm. We let $A_k = W_{k+1}^\top \ldots W_L^\top p w^{\star\top} \frac{2}{n} X^\top X W_1^\top \ldots W_{k-1}^\top$, and choose $\beta_k = \|A_k\|_2$. Then

$$w_{\text{prod}}^\top \nabla R^1(\tilde{w}_{\text{prod}}) \geqslant \xi \sum_{k=1}^{L} \|A_k\|_2^2 + \mathcal{O}(\xi^2) \,.$$

Coming back to (19), we get

$$\sum_{k=1}^{L} \|\Delta_k\|_F^2 \geqslant \frac{L\xi^2 \|p\|_2^{2/L}}{\|w^\star\|_2^{2/L}} \left(\sum_{k=1}^{L} \|A_k\|_2^2\right)^2 + \mathcal{O}(\xi^3) \,.$$

Furthermore,

$$\sum_{k=1}^{L} \|W_k - \tilde{W}_k\|_F^2 = \xi^2 \sum_{k=1}^{L} \beta_k^2 \frac{\|u_k v_k^\top\|_F^2}{\|u_k\|_2^2 \|v_k\|_2^2} = \xi^2 \sum_{k=1}^{L} \beta_k^2 = \xi^2 \sum_{k=1}^{L} \|A_k\|_2^2 \,.$$

Therefore, we obtain, by (14),

$$S(\mathcal{W})^2 \geqslant \frac{L\|p\|_2^{2/L}}{\|w^\star\|_2^{2/L}} \sum_{k=1}^{L} \|A_k\|_2^2 \,.$$

We can lower bound the sum similarly to (18):

$$\sum_{k=1}^{L} \|A_k\|_2^2 \geqslant L\big(\|A_L \cdots A_1\|_2\big)^{2/L} \,,$$

and

$$A_L \cdots A_1 = p w^{\star\top} \frac{2}{n} X^\top X \underbrace{w_{\text{prod}} w^{\star\top} \frac{2}{n} X^\top X \cdots w_{\text{prod}} w^{\star\top} \frac{2}{n} X^\top X}_{L-1 \text{ times}}$$

$$= p \Big( w^{\star\top} \frac{2}{n} X^\top X w^\star \Big)^{L-1} w^{\star\top} \frac{2}{n} X^\top X.$$

Thus, using the Cauchy-Schwarz inequality, we get

$$\|A_L \cdots A_1\|_2 = \Big( w^{\star\top} \frac{2}{n} X^\top X w^\star \Big)^{L-1} \| p w^{\star\top} \frac{2}{n} X^\top X \|_2$$

$$= \Big( w^{\star\top} \frac{2}{n} X^\top X w^\star \Big)^{L-1} \|p\|_2 \| w^{\star\top} \frac{2}{n} X^\top X \|_2$$

$$\geqslant \frac{\|p\|_2}{\|w^\star\|_2} \Big( w^{\star\top} \frac{2}{n} X^\top X w^\star \Big)^{L}$$

$$= 2^L a^L \|p\|_2 \|w^\star\|_2^{2L-1}.$$

Therefore

$$\sum_{k=1}^{L} \|A_k\|_2^2 \geqslant 4 L a^2 \|w^\star\|_2^{4-\frac{2}{L}} \|p\|_2^{\frac{2}{L}}.$$

We finally obtain

$$S(\mathcal{W})^2 \geqslant 4 L^2 a^2 \|w^\star\|_2^{4-\frac{4}{L}} \|p\|_2^{\frac{4}{L}},$$

which gives the second lower bound.

To obtain the upper bound on $S_{\min}$, we construct explicitly a minimizer, and upper bound its sharpness. More precisely, let the $W_k$ be rank-one matrices, such that $\|W_k\|_2 = \|w^\star\|^{1/L} / \|p\|^{1/L}$, successive matrices have aligned first singular vectors, the first right singular vector of $W_1$ is aligned with $w^\star$, and the first left singular vector of $W_L$ is aligned with $p$. We then have

$$p^\top W_L \ldots W_1 = w^\star,$$

meaning that the network minimizes the loss. We now upper bound its sharpness using (13), where we recall that the matrix appearing in the numerator is denoted by $\Delta_k$ and satisfies $\Delta_k = \bar{\Delta}_k + \mathcal{O}(\xi^2)$, where $\bar{\Delta}_k$ is given by (17). Contrarily to the proof of the lower bounds where we exhibited a specific direction $\tilde{W}$, we here seek an upper bound valid for all $\tilde{W}$. Recall that, by (16), we have

$$\nabla R^1(\tilde{w}_{\text{prod}}) = \frac{2}{n} X^\top X \sum_{k=1}^{L} W_1^\top \ldots W_{k-1}^\top (\tilde{W}_k - W_k)^\top W_{k+1}^\top \ldots W_L^\top p + \mathcal{O}(\xi^2). \quad (20)$$

By subadditivity of the operator norm,

$$\|\nabla R^1(\tilde{w}_{\text{prod}})\|_2 \leqslant 2\Lambda \sum_{k=2}^{L} \|W_1\|_2 \ldots \|W_{k-1}\|_2 \|\tilde{W}_k - W_k\|_2 \|W_{k+1}\|_2 \ldots \|W_L\|_2 \|p\|_2 + \mathcal{O}(\xi^2)$$

$$= 2\Lambda \|w^\star\|_2^{(L-1)/L} \|p\|_2^{1/L} \sum_{k=1}^{L} \|\tilde{W}_k - W_k\|_2 + \mathcal{O}(\xi^2). \quad (21)$$

Moving on to bounding the squared Frobenius norm of $\Delta_k$, we observe that $\bar{\Delta}_k$ decomposes as a rank-one matrix. We split cases for $k = 1$ and $k > 1$. First, for $k = 1$, we have, by subadditivity of the operator norm and by (21),

$$\|\Delta_1\|_F = \|W_2^\top \ldots W_L^\top p\|_2 \|\nabla R^1(\tilde{w}_{\text{prod}})\|_2 + \mathcal{O}(\xi^2)$$

$$\leqslant \|W_2\|_2 \ldots \|W_L\|_2 \|p\|_2 \|\nabla R^1(\tilde{w}_{\text{prod}})\|_2 + \mathcal{O}(\xi^2)$$

$$\leqslant 2\Lambda \|w^\star\|_2^{2(L-1)/L} \|p\|_2^{2/L} \sum_{k=1}^{L} \|\tilde{W}_k - W_k\|_2 + \mathcal{O}(\xi^2).$$

Then

$$\|\Delta_1\|_F^2 \leqslant 4\Lambda^2 \|w^\star\|_2^{4(L-1)/L} \|p\|_2^{4/L} \Big( \sum_{k=1}^{L} \|\tilde{W}_k - W_k\|_2 \Big)^2 + \mathcal{O}(\xi^3)$$

$$\leqslant 4L\Lambda^2 \|w^\star\|_2^{4(L-1)/L} \|p\|_2^{4/L} \sum_{k=1}^{L} \|\tilde{W}_k - W_k\|_2^2 + \mathcal{O}(\xi^3)$$

$$\leqslant 4L\Lambda^2 \|w^\star\|_2^{4(L-1)/L} \|p\|_2^{4/L} \sum_{k=1}^{L} \|\tilde{W}_k - W_k\|_F^2 + \mathcal{O}(\xi^3) \,.$$

For $k > 1$, we have

$$\|\Delta_k\|_F = \|W_{k+1}^\top \dots W_L^\top p\|_2 \|\nabla R^1(\tilde{w}_{\mathrm{prod}})^\top W_1^\top \dots W_{k-1}^\top\|_2 + \mathcal{O}(\xi^2)$$

$$\leqslant \|W_{k+1}\|_2 \dots \|W_L\|_2 \|p\|_2 \|\nabla R^1(\tilde{w}_{\mathrm{prod}})^\top W_1^\top\|_2 \|W_2\|_2 \dots \|W_{k-1}\|_2 + \mathcal{O}(\xi^2)$$

$$= \|w^\star\|_2^{(L-2)/L} \|p\|_2^{2/L} \|\nabla R^1(\tilde{w}_{\mathrm{prod}})^\top W_1^\top\|_2 + \mathcal{O}(\xi^2) \,. \tag{22}$$

Let us now bound $\|\nabla R^1(\tilde{w}_{\mathrm{prod}})^\top W_1^\top\|_2$. By (20), separating the first term, we have

$$\|\nabla R^1(\tilde{w}_{\mathrm{prod}})^\top W_1^\top\|_2$$

$$\leqslant \|p^\top W_L \dots W_2 (\tilde{W}_1 - W_1) \frac{2}{n} X^\top X W_1^\top\|_2$$

$$+ \sum_{k=2}^{L} \|p^\top W_L \dots W_{k+1} (\tilde{W}_k - W_k) W_{k-1} \dots W_1 \frac{2}{n} X^\top X W_1^\top\|_2 + \mathcal{O}(\xi^2)$$

$$\leqslant 2\Lambda \|p\|_2 \|W_L\|_2 \dots \|W_2\|_2 \|\tilde{W}_1 - W_1\|_2 \|W_1\|_2$$

$$+ \sum_{k=2}^{L} \|p\|_2 \|W_L\|_2 \dots \|W_{k+1}\|_2 \|\tilde{W}_k - W_k\|_2 \|W_{k-1}\|_2 \dots \|W_2\|_2 \Big\| W_1 \frac{2}{n} X^\top X W_1^\top \Big\|_2 + \mathcal{O}(\xi^2)$$

$$= 2\Lambda \|w^\star\|_2 \|\tilde{W}_1 - W_1\|_2$$

$$+ \sum_{k=2}^{L} \|w^\star\|_2^{(L-2)/L} \|p\|_2^{2/L} \Big\| W_1 \frac{2}{n} X^\top X W_1^\top \Big\|_2 \|\tilde{W}_k - W_k\|_2 + \mathcal{O}(\xi^2) \,.$$

Finally, recall that $W_1$ is rank-one and its first right singular vector is aligned with $w^\star$. A short computation therefore shows that

$$\Big\| W_1 \frac{2}{n} X^\top X W_1^\top \Big\|_2 = 2a\|W_1\|^2 = 2a\|w^\star\|_2^{2/L} \|p\|_2^{-2/L} \,.$$

Thus

$$\|\nabla R^1(\tilde{w}_{\mathrm{prod}})^\top W_1^\top\|_2 \leqslant 2\|w^\star\|_2 \Big( \Lambda \|\tilde{W}_1 - W_1\|_2 + a \sum_{k=2}^{L} \|\tilde{W}_k - W_k\|_2 \Big) + \mathcal{O}(\xi^2) \,.$$

Then, coming back to (22), for $k > 1$,

$$\|\Delta_k\|_F \leqslant 2\|w^\star\|_2^{2(L-1)/L} \|p\|_2^{2/L} \Big( \Lambda \|\tilde{W}_1 - W_1\|_2 + a \sum_{k=2}^{L} \|\tilde{W}_k - W_k\|_2 \Big) + \mathcal{O}(\xi^2) \,.$$

Thus

$$\|\Delta_k\|_F^2 \leqslant 4\|w^\star\|_2^{4(L-1)/L} \|p\|_2^{4/L} \Big( \Lambda \|\tilde{W}_1 - W_1\|_2 + a \sum_{k=2}^{L} \|\tilde{W}_k - W_k\|_2 \Big)^2 + \mathcal{O}(\xi^3)$$

$$\leqslant 4\|w^\star\|_2^{4(L-1)/L} \|p\|_2^{4/L} (\Lambda^2 + (L-1)a^2) \sum_{k=1}^{L} \|\tilde{W}_k - W_k\|_2^2 + \mathcal{O}(\xi^3)$$

$$\leqslant 4\|w^\star\|_2^{4(L-1)/L} \|p\|_2^{4/L} (\Lambda^2 + (L-1)a^2) \sum_{k=1}^{L} \|\tilde{W}_k - W_k\|_F^2 + \mathcal{O}(\xi^3) \,,$$

where the second inequality holds by the Cauchy-Schwarz inequality. Thus, by (13), putting together the bounds on $\|\Delta_k\|_F^2$ for $k = 1$ and $k > 1$,

$$
\begin{aligned}
S(\mathcal{W})^2 &= \lim_{\xi \to 0} \sup_{\|W_k - \tilde{W}_k\|_F \leqslant \xi} \frac{\sum_{k=1}^L \|\Delta_k\|_F^2}{\sum_{k=1}^L \|W_k - \tilde{W}_k\|_F^2} \\
&\leqslant \lim_{\xi \to 0} 4\|w^\star\|_2^{4(L-1)/L} \|p\|_2^{4/L} (L\Lambda^2 + (L-1)(\Lambda^2 + (L-1)a^2)) + \mathcal{O}(\xi) \,.
\end{aligned}
$$

Therefore

$$
S(\mathcal{W}) \leqslant 2\|w^\star\|_2^{2 - \frac{2}{L}} \|p\|_2^{\frac{2}{L}} \sqrt{(2L-1)\Lambda^2 + (L-1)^2 a^2} \,,
$$

which concludes the proof.

## B.2 Proof of Lemma 1

This identity can be shown by noting that the identity is trivially true for $t = 0$, then differentiating on both sides with respect to time, and using (5). We refer, e.g., to Arora et al. (2018) for a detailed proof.

## B.3 Proof of Lemma 2

Before proving the three statements of the lemma in order, we let

$$
\bar{\varepsilon} = \max_{1 \leqslant k \leqslant L} \|W_k(0)\|_F^2 + \sum_{k=1}^{L-1} \|W_{k+1}^\top(0)W_{k+1}(0) - W_k(0)W_k^\top(0)\|_2 \,.
$$

Note that $\bar{\varepsilon} \leqslant \varepsilon$.

**First statement.**  The claim is true for $k = L$ since $W_L$ is a (row) vector. For $k \in \{1, \ldots, L-1\}$, taking the 2-norm in Lemma 1, we have

$$
\begin{aligned}
\|W_{k+1}^\top W_{k+1}\|_2 &= \|W_k W_k^\top + W_{k+1}^\top(0)W_{k+1}(0) - W_k(0)W_k^\top(0)\|_2 \\
&\leqslant \|W_k W_k^\top\|_2 + \|W_{k+1}^\top(0)W_{k+1}(0) - W_k(0)W_k^\top(0)\|_2 \,.
\end{aligned}
$$

Thus, using $\|A^\top A\|_2 = \|AA^\top\|_2 = \|A\|_2^2$,

$$
\|W_{k+1}\|_2^2 - \|W_{k+1}^\top(0)W_{k+1}(0) - W_k(0)W_k^\top(0)\|_2 \leqslant \|W_k\|_2^2 \,. \tag{23}
$$

We now take the trace in Lemma 1 to obtain

$$
\|W_{k+1}\|_F^2 - \|W_{k+1}(0)\|_F^2 = \|W_k\|_F^2 - \|W_k(0)\|_F^2 \,.
$$

Combining with the inequality above, we have that

$$
\begin{aligned}
\|W_k\|_F^2 - \|W_k\|_2^2 &\leqslant \|W_{k+1}\|_F^2 - \|W_{k+1}\|_2^2 + \|W_k(0)\|_F^2 - \|W_{k+1}(0)\|_F^2 \\
&\quad + \|W_{k+1}^\top(0)W_{k+1}(0) - W_k(0)W_k^\top(0)\|_2
\end{aligned}
$$

Summing from $k$ to $L-1$ and telescoping, we have

$$
\begin{aligned}
\|W_k\|_F^2 - \|W_k\|_2^2 &\leqslant \|W_L\|_F^2 - \|W_L\|_2^2 + \|W_k(0)\|_F^2 - \|W_L(0)\|_F^2 \\
&\quad + \sum_{k'=k}^{L-1} \|W_{k'+1}^\top(0)W_{k'+1}(0) - W_{k'}(0)W_{k'}^\top(0)\|_2 \,.
\end{aligned}
$$

The first two terms compensate since $W_L$ is a vector, and the remainder of the terms is less than $\bar{\varepsilon} \leqslant \varepsilon$ by definition. This gives the first statement of the Lemma.

**Second statement.** Assume without loss of generality that $j > k$. By recurrence over (23), we obtain that

$$\sigma_j^2 \leqslant \sigma_k^2 + \sum_{k'=k}^{j-1} \|W_{k'+1}^\top(0)W_{k'+1}(0) - W_{k'}(0)W_{k'}^\top(0)\|_2 \leqslant \sigma_k^2 + \bar\varepsilon\,.$$

The reverse inequality can be shown similarly: by considering again Lemma 1 and taking the 2-norm, we get

$$\|W_k W_k^\top\|_2 \leqslant \|W_{k+1}^\top W_{k+1}\|_2 + \|W_{k+1}^\top(0)W_{k+1}(0) - W_k(0)W_k^\top(0)\|_2$$

Thus

$$\|W_k\|_2^2 \leqslant \|W_{k+1}\|_2^2 + \|W_{k+1}^\top(0)W_{k+1}(0) - W_k(0)W_k^\top(0)\|_2\,.$$

As previously, we get by recurrence that

$$\sigma_k^2 \leqslant \sigma_j^2 + \bar\varepsilon\,.$$

Combined with the reverse bound above, this gives the second statement of the lemma.

**Third statement.** Let us lower and upper bound $u_k^\top W_{k+1}^\top W_{k+1} u_k$. We have on the one hand, by Lemma 1,

$$\begin{aligned}
u_k^\top W_{k+1}^\top W_{k+1} u_k &= u_k^\top W_k W_k^\top u_k - u_k^\top W_k(0)W_k^\top(0)u_k + u_k^\top W_{k+1}^\top(0)W_{k+1}(0)u_k \\
&\geqslant u_k^\top W_k W_k^\top u_k - u_k^\top W_k(0)W_k^\top(0)u_k \\
&\geqslant \sigma_k^2 - \|W_k(0)\|_2^2\,.
\end{aligned}$$

On the other hand,

$$\begin{aligned}
u_k^\top W_{k+1}^\top W_{k+1} u_k &= u_k^\top \big(W_{k+1}^\top W_{k+1} - v_{k+1}\sigma_{k+1}^2 v_{k+1}^\top\big)u_k + u_k^\top v_{k+1}\sigma_{k+1}^2 v_{k+1}^\top u_k \\
&\leqslant \|W_{k+1}^\top W_{k+1} - v_{k+1}\sigma_{k+1}^2 v_{k+1}^\top\|_2 + \langle v_{k+1}, u_k\rangle^2 \sigma_{k+1}^2
\end{aligned}$$

The 2-norm above is equal to the second largest eigenvalue of $W_{k+1}^\top W_{k+1}$, which is the square of the second largest singular value of $W_{k+1}$. In particular, it is lower than $\|W_{k+1}\|_F^2 - \|W_{k+1}\|_2^2$ which is the sum of the squared singular values of $W_{k+1}$ except the largest one. We obtain

$$u_k^\top W_{k+1}^\top W_{k+1} u_k \leqslant \|W_{k+1}\|_F^2 - \|W_{k+1}\|_2^2 + \langle v_{k+1}, u_k\rangle^2 \sigma_{k+1}^2 \leqslant \bar\varepsilon + \langle v_{k+1}, u_k\rangle^2 \sigma_{k+1}^2$$

by the first statement of the Lemma. Combining the lower and upper bound of $u_k^\top W_{k+1}^\top W_{k+1} u_k$, we get

$$\bar\varepsilon + \langle v_{k+1}, u_k\rangle^2 \sigma_{k+1}^2 \geqslant \sigma_k^2 - \|W_k(0)\|_2^2\,.$$

Thus

$$\langle v_{k+1}, u_k\rangle^2 \geqslant \frac{\sigma_k^2 - \|W_k(0)\|_2^2 - \bar\varepsilon}{\sigma_{k+1}^2} \geqslant \frac{\sigma_{k+1}^2 - \bar\varepsilon - \|W_k(0)\|_2^2 - \bar\varepsilon}{\sigma_{k+1}^2}\,,$$

by the second statement of the Lemma. We finally obtain

$$\langle v_{k+1}, u_k\rangle^2 \geqslant 1 - \frac{2\bar\varepsilon + \|W_k(0)\|_2^2}{\sigma_{k+1}^2} \geqslant 1 - \frac{\varepsilon}{\sigma_{k+1}^2}\,,$$

by definition of $\varepsilon$ and $\bar\varepsilon$.

## B.4  Proof of Theorem 3

We lower bound the first term in the sum of the left-hand side. Recall that, by (5), we have

$$\nabla_1 R^L(W(t)) = \underbrace{(W_L(t)\dots W_2(t))^\top}_{d_1 \times 1}\underbrace{\nabla R^1(w_{\mathrm{prod}}(t))^\top}_{1 \times d_0}\,,$$

thus

$$\big\|\nabla_1 R^L(W(t))\big\|_F^2 = \|W_L(t)\dots W_2(t)\|_2^2 \big\|\nabla R^1(w_{\mathrm{prod}}(t))\big\|_2^2\,. \tag{24}$$

We show that $\|W_L(t)\dots W_2(t)\|_2$ is large by distinguishing between two cases depending on the magnitude of $\sigma_1(t) = \|W_1(t)\|_2$. To this aim, let $C > \sqrt{2\varepsilon L}$.

**Large spectral norm.** We first consider the case where

$$\sigma_1 > C > \sqrt{2\varepsilon}L\,. \tag{25}$$

By Lemma 2, for $k \in \{1, \ldots, L-1\}$,

$$\sigma_k^2 \geqslant \sigma_1^2 - \varepsilon > \varepsilon\,,$$

where the second inequality unfolds from (25). Then, again by Lemma 2, for $k \in \{1, \ldots, L-1\}$,

$$\langle v_{k+1}, u_k \rangle^2 \geqslant 1 - \frac{\varepsilon}{\sigma_{k+1}^2} > 0\,.$$

It is always possible (without loss of generality) to choose the orientation of the $u_k$ and $v_k$ such that $\langle v_{k+1}, u_k \rangle \geqslant 0$ for any $k \in \{1, \ldots L-1\}$. Making this choice, the equation above implies that

$$\begin{aligned}
\|v_{k+1} - u_k\|^2 &= 2 - 2\langle v_{k+1}, u_k \rangle \\
&\leqslant 2\left(1 - \sqrt{1 - \frac{\varepsilon}{\sigma_{k+1}^2}}\right)\,.
\end{aligned}$$

For $x \in [0, 1]$, $\sqrt{1-x} \geqslant 1 - x$, and thus

$$\|v_{k+1} - u_k\|_2^2 \leqslant \frac{2\varepsilon}{\sigma_{k+1}^2}\,. \tag{26}$$

Let us show that this implies a lower bound on $\|W_L \ldots W_2\|_2$. To do so, let us denote recursively

$$x_1 = v_2, \quad x_{k+1} = W_{k+1}x_k \quad \text{for} \quad k \in \{1, \ldots, L-1\}\,.$$

We then have $x_L = W_L \ldots W_2 x_1$, thus

$$\|W_L \ldots W_2\|_2 \geqslant \frac{\|x_L\|_2}{\|x_1\|_2} = \|x_L\|_2\,.$$

Our goal is thus to lower bound $\|x_L\|_2$, which entails a lower bound on $\|W_L \ldots W_2\|_2$. To this aim, first note that, for any $k \in \{1, \ldots, L-1\}$,

$$\begin{aligned}
\langle x_{k+1}, u_{k+1} \rangle &= \langle W_{k+1}x_k, u_{k+1} \rangle \\
&= \sigma_{k+1}\langle x_k, v_{k+1} \rangle \\
&= \sigma_{k+1}\langle x_k, u_k + v_{k+1} - u_k \rangle \\
&\geqslant \sigma_{k+1}\langle x_k, u_k \rangle - \sigma_{k+1}\|x_k\|_2\|v_{k+1} - u_k\|_2 \\
&\geqslant \sigma_{k+1}\langle x_k, u_k \rangle - \sqrt{2\varepsilon}\|x_k\|_2\,. \tag{27}
\end{aligned}$$

where the last equation stems from (26). Denote $\alpha_k = \langle \frac{x_k}{\|x_k\|_2}, u_k \rangle$. Then the previous equation shows that

$$\alpha_{k+1} \geqslant \frac{\|x_k\|_2}{\|x_{k+1}\|_2}(\sigma_{k+1}\alpha_k - \sqrt{2\varepsilon}) = \frac{\|x_k\|_2\sigma_{k+1}}{\|x_{k+1}\|_2}\left(\alpha_k - \frac{\sqrt{2\varepsilon}}{\sigma_{k+1}}\right)\,.$$

Further note that $\|x_{k+1}\|_2 \leqslant \sigma_{k+1}\|x_k\|_2$, thus

$$\alpha_{k+1} \geqslant \alpha_k - \frac{\sqrt{2\varepsilon}}{\sigma_{k+1}}\,.$$

By recurrence,

$$\alpha_k \geqslant \alpha_2 - \sqrt{2\varepsilon}\sum_{k'=2}^{k-1}\frac{1}{\sigma_{k+1}} \geqslant \alpha_2 - \sqrt{2\varepsilon}\sum_{k'=2}^{k-1}\frac{1}{\sqrt{\sigma_1^2 - \varepsilon}} = \alpha_2 - \frac{\sqrt{2\varepsilon}(k-2)}{\sqrt{\sigma_1^2 - \varepsilon}}\,.$$

Coming back to (27), we have

$$\|x_{k+1}\|_2 \geqslant \langle x_{k+1}, u_{k+1} \rangle \geqslant \|x_k\|_2(\sigma_{k+1}\alpha_k - \sqrt{2\varepsilon})\,,$$

thus by recurrence,

$$\|x_L\|_2 \geqslant \|x_2\|_2 \prod_{k=2}^{L-1} (\sigma_{k+1}\alpha_k - \sqrt{2\varepsilon}) \geqslant \|x_2\|_2 \prod_{k=2}^{L-1} \left( \sqrt{\sigma_1^2 - \varepsilon} \Big( \alpha_2 - \frac{\sqrt{2\varepsilon}(k-2)}{\sqrt{\sigma_1^2 - \varepsilon}} \Big) - \sqrt{2\varepsilon} \right)$$

$$\geqslant \|x_2\|_2 \prod_{k=2}^{L-1} \left( \sqrt{\sigma_1^2 - \varepsilon}\, \alpha_2 - \sqrt{2\varepsilon}k \right).$$

Finally, by definition, $\alpha_2 = \langle \frac{x_2}{\|x_2\|_2}, u_2 \rangle$. Since $x_2 = W_2 x_1 = W_2 v_2 = \sigma_2 u_2$, we obtain that $\alpha_2 = 1$, and thus

$$\|W_L \dots W_2\|_2 \geqslant \|x_L\|_2$$

$$\geqslant \sigma_2 \prod_{k=2}^{L-1} \left( \sqrt{\sigma_1^2 - \varepsilon} - \sqrt{2\varepsilon}k \right)$$

$$\geqslant \sqrt{\sigma_1^2 - \varepsilon} \Big( \sqrt{\sigma_1^2 - \varepsilon} - \sqrt{2\varepsilon}(L-1) \Big)^{L-2}$$

$$\geqslant \Big( \sqrt{\sigma_1^2} - \sqrt{\varepsilon} - \sqrt{2\varepsilon}(L-1) \Big)^{L-1}$$

$$\geqslant \big( \sigma_1 - \sqrt{2\varepsilon}L \big)^{L-1}.$$

We finally get

$$\|W_L \dots W_2\|_2 \geqslant \big( C - \sqrt{2\varepsilon}L \big)^{L-1}, \tag{28}$$

which is a positive quantity by definition of $C$.

**Small spectral norm.** We now inspect the case where (25) is not satisfied, that is, $\sigma_1(t) \leqslant C$. First note that, by the formula (6) for the gradient of $R^1$,

$$\|\nabla R^1(w_{\mathrm{prod}})\|_2 \leqslant 2\Lambda \|w^\star - w_{\mathrm{prod}}\|_2 \leqslant 2\Lambda(\|w^\star\|_2 + \|w_{\mathrm{prod}}\|_2).$$

Thus, for $\|w_{\mathrm{prod}}\|_2 \leqslant \|w^\star\|_2$, $w_{\mathrm{prod}} \mapsto R^1(w_{\mathrm{prod}})$ is $4\Lambda\|w^\star\|_2$-Lipschitz. Let us use this property to lower bound $\|w_{\mathrm{prod}}(t)\|_2$ by a constant independent of $t$, for $t \geqslant 1$. Either we have $\|w_{\mathrm{prod}}(t)\|_2 \geqslant \|w^\star\|_2$, or $\|w_{\mathrm{prod}}\|_2 \leqslant \|w^\star\|_2$, but then, by the Lipschitzness property,

$$|R^1(w_{\mathrm{prod}}(t)) - R_0| \leqslant 4\Lambda\|w^\star\|_2 \|w_{\mathrm{prod}}(t) - 0\|_2 = 4\Lambda\|w^\star\|_2 \|w_{\mathrm{prod}}(t)\|_2,$$

where we recall that $R_0$ is the risk associated to the null parameters. Furthermore, for $t \geqslant 1$,

$$\begin{aligned} R^1(w_{\mathrm{prod}}(t)) = R^L(\mathcal{W}(t)) & \\ &\leqslant R^L(\mathcal{W}(1)) && \text{(The risk is decreasing along the gradient flow)} \\ &< R^L(\mathcal{W}(0)) && (\nabla R^L(\mathcal{W}(0)) \neq 0 \text{ by Assumption } (A_1)) \\ &\leqslant R_0. && \text{(by Assumption } (A_1)) \end{aligned}$$

Thus, for $t \geqslant 1$,

$$|R^1(w_{\mathrm{prod}}(t)) - R_0| > R_0 - R^L(\mathcal{W}(1)) > 0.$$

To summarize, we proved that, for $t \geqslant 1$,

$$\|w_{\mathrm{prod}}(t)\|_2 \geqslant \min\left( \frac{R_0 - R^L(\mathcal{W}(1))}{4\Lambda\|w^\star\|_2}, \|w^\star\|_2 \right) > 0,$$

where we recall that $\|w^\star\|_2 > 0$ by assumption (see Section 2). Furthermore,

$$\|w_{\mathrm{prod}}(t)\|_2 \leqslant \|W_L(t) \dots W_2(t)\|_2 \|W_1(t)\|_2 = \|W_L(t) \dots W_2(t)\|_2 \sigma_1(t).$$

Then, for $t \geqslant 1$,

$$\|W_L(t) \dots W_2(t)\|_2 \geqslant \frac{1}{\sigma_1(t)} \min\left( \frac{R_0 - R^L(\mathcal{W}(1))}{4\Lambda\|w^\star\|_2}, \|w^\star\|_2 \right)$$

$$\geqslant \frac{1}{C} \min\left( \frac{R_0 - R^L(\mathcal{W}(1))}{4\Lambda\|w^\star\|_2}, \|w^\star\|_2 \right). \tag{29}$$

**Conclusion.** Combining (28) and (29), we obtain that, for $t \geqslant 1$,

$$\|W_L(t)\dots W_2(t)\|_2 \geqslant \min\left(\left(C - \sqrt{2\varepsilon}L\right)^{L-1}, \frac{1}{C}\min\left(\frac{R_0 - R^L(\mathcal{W}(1))}{4\Lambda\|w^\star\|_2}, \|w^\star\|_2\right)\right) =: \sqrt{\mu_1},$$

where $\mu_1 > 0$ by the proof above. Then, for $t \geqslant 1$, by (24),

$$\left\|\nabla_1 R^L(W(t))\right\|_F^2 \geqslant \mu_1 \|\nabla R^1(w_{\mathrm{prod}}(t))\|_2^2.$$

By Lemma 8,

$$\|\nabla R^1(w_{\mathrm{prod}}(t))\|_2^2 \geqslant 4\lambda(R^L(\mathcal{W}(t)) - R_{\min}).$$

Thus, taking $\mu = 4\mu_1\lambda > 0$, for $t \geqslant 1$,

$$\sum_{k=1}^{L}\left\|\nabla_k R^L(\mathcal{W}(t))\right\|_F^2 \geqslant \left\|\nabla_1 R^L(\mathcal{W}(t))\right\|_F^2 \geqslant \mu(R^L(\mathcal{W}(t)) - R_{\min}).$$

## B.5 Proof of Corollary 1

We first show that Assumption $(A_2)$ implies a number of estimates that are useful in the following. Since $\|w^\star\|_2 \geqslant 1$, we have

$$\left(\frac{\|w^\star\|_2}{2}\right)^{1/L} \geqslant \left(\frac{1}{2}\right)^{1/L} \geqslant \frac{1}{2} \tag{30}$$

thus

$$8L\varepsilon \leqslant 8L\sqrt{\varepsilon} \leqslant \frac{1}{4} \leqslant \left(\frac{\|w^\star\|_2}{2}\right)^{1/L} \leqslant \left(2\|w^\star\|_2\right)^{1/L}. \tag{31}$$

Moreover,

$$\left(\frac{\|w^\star\|_2}{2}\right)^{2/L} \geqslant \frac{1}{4}$$

so we also have

$$8L\varepsilon \leqslant \left(\frac{\|w^\star\|_2}{2}\right)^{2/L}. \tag{32}$$

Let us now prove the Corollary. We first note that Theorem 3 implies exponential convergence of the empirical risk to its minimum by Lemma 7. This also implies the (exponential) convergence of $w_{\mathrm{prod}}$ to $w^\star$, since the covariance matrix $X^\top X$ is full rank, so $w^\star$ is the unique minimizer of $R^1$, and, by Lemma 8,

$$R^L(\mathcal{W}) - R_{\min} = \frac{1}{n}\|X(w^\star - w_{\mathrm{prod}})\|_2^2 \geqslant \lambda\|w^\star - w_{\mathrm{prod}}\|_2^2,$$

Furthermore,

$$\|w_{\mathrm{prod}}\|_2 = \|W_1^\top \dots W_L^\top\|_2 \leqslant \|W_1\|_2 \dots \|W_L\|_2 \leqslant \left(\max_{k=1}^{L}\sigma_k\right)^L.$$

Let us show that, for $t$ large enough,

$$\max_{k=1}^{L}\sigma_k \geqslant \left(\frac{\|w^\star\|}{2}\right)^{1/L} + \varepsilon. \tag{33}$$

If it were not the case, then

$$\begin{aligned}
\|w_{\mathrm{prod}}\|_2 &\leqslant \left(\max_{k=1}^{L}\sigma_k\right)^L \\
&\leqslant \left(\left(\frac{\|w^\star\|_2}{2}\right)^{1/L} + \varepsilon\right)^L \\
&= \frac{\|w^\star\|_2}{2}\left(1 + \varepsilon\left(\frac{2}{\|w^\star\|_2}\right)^{1/L}\right)^L \\
&\leqslant \frac{\|w^\star\|_2}{2}\left(1 + 2L\varepsilon\left(\frac{2}{\|w^\star\|_2}\right)^{1/L}\right),
\end{aligned}$$

where the last inequality holds by Lemma 4 since

$$L\varepsilon\Big(\frac{2}{\|w^\star\|_2}\Big)^{1/L} \leqslant 1 \quad \Leftrightarrow \quad L\varepsilon \leqslant \Big(\frac{\|w^\star\|_2}{2}\Big)^{1/L},$$

which holds by (31). Then, we obtain

$$\|w_{\mathrm{prod}}\|_2 \leqslant \frac{3\|w^\star\|_2}{4},$$

since

$$1 + 2L\varepsilon\Big(\frac{2}{\|w^\star\|_2}\Big)^{1/L} \leqslant \frac{3}{2} \quad \Leftrightarrow \quad 4L\varepsilon \leqslant \Big(\frac{\|w^\star\|_2}{2}\Big)^{1/L},$$

which also holds by (31). The inequality $\|w_{\mathrm{prod}}\|_2 \leqslant \frac{3\|w^\star\|_2}{4}$ contradicts the fact that $w_{\mathrm{prod}}$ converges to $w^\star$, thus proving (33). Then, we have

$$\max_{k=1}^{L} \sigma_k^2 \geqslant \Big(\frac{\|w^\star\|_2}{2}\Big)^{2/L} + 2\varepsilon\Big(\frac{\|w^\star\|_2}{2}\Big)^{1/L} \geqslant \Big(\frac{\|w^\star\|_2}{2}\Big)^{2/L} + \varepsilon,$$

where the last inequality holds by (30). Furthermore, by Lemma 2, for all $k \in \{1, \dots, L\}$,

$$\sigma_k^2 \geqslant \max_{j=1}^{L} \sigma_j^2 - \varepsilon.$$

This brings the first inequality of the Corollary. We now show the second inequality of the Corollary. First note that, by Lemma 2,

$$\|W_k - \sigma_k u_k v_k^\top\|_F^2 = \|W_k\|_F^2 - \|W_k\|_2^2 \leqslant \varepsilon,$$

since both quantities are equal to the sum of the squared singular values of $W_k$ except the largest one. Then, adding and substracting,

$$\|W_1^\top \dots W_L^\top - \sigma_1 \dots \sigma_L v_1 u_1^\top \dots v_L u_L^\top\|_2$$
$$\leqslant \sum_{k=1}^{L} \|W_1^\top \dots W_{k-1}^\top (W_k^\top - \sigma_k u_k v_k^\top)\sigma_{k+1} \dots \sigma_L v_{k+1} u_{k+1}^\top \dots v_L u_L^\top\|_2$$
$$\leqslant \sum_{k=1}^{L} \|W_1\|_2 \dots \|W_{k-1}\|_2 \|W_k^\top - \sigma_k u_k v_k^\top\|_2 \sigma_{k+1} \dots \sigma_L$$
$$\leqslant \sum_{k=1}^{L} \sigma_1 \dots \sigma_{k-1} \|W_k^\top - \sigma_k u_k v_k^\top\|_F \sigma_{k+1} \dots \sigma_L$$
$$\leqslant \sqrt{\varepsilon} \sum_{k=1}^{L} \prod_{j \neq k} \sigma_j$$
$$= \sqrt{\varepsilon} \prod_{j=1}^{L} \sigma_j \sum_{k=1}^{L} \frac{1}{\sigma_k}$$
$$\leqslant \sqrt{\varepsilon} \prod_{j=1}^{L} \sigma_j \frac{L}{\min \sigma_k}$$
$$\leqslant \frac{L\sqrt{\varepsilon}}{\Big(\frac{\|w^\star\|}{2}\Big)^{1/L}} \prod_{j=1}^{L} \sigma_j,$$

by the first inequality of the Corollary. Moreover, using again the first inequality of the Corollary and Lemma 2, we have that

$$\langle v_{k+1}, u_k \rangle^2 \geqslant 1 - \frac{\varepsilon}{\sigma_{k+1}^2} \geqslant 1 - \frac{\varepsilon}{\Big(\frac{\|w^\star\|_2}{2}\Big)^{2/L}}.$$

Since $u_L = 1$, we deduce that

$$\|\sigma_1 \ldots \sigma_L v_1 u_1^\top \ldots v_L u_L^\top\|_2 = \prod_{k=1}^L \sigma_k \|v_1\|_2 \prod_{k=1}^{L-1} u_k^\top v_{k+1}$$

$$\geqslant \prod_{k=1}^L \sigma_k \left(1 - \frac{\varepsilon}{\left(\frac{\|w^\star\|_2}{2}\right)^{2/L}}\right)^{\frac{L-1}{2}}$$

$$\geqslant \prod_{k=1}^L \sigma_k \left(1 - \frac{(L-1)\varepsilon}{\left(\frac{\|w^\star\|_2}{2}\right)^{2/L}}\right)$$

where in the last step we used Lemma 4 which is valid since

$$\frac{\varepsilon}{\left(\frac{\|w^\star\|_2}{2}\right)^{2/L}} \leqslant \frac{1}{2} \quad \Leftrightarrow \quad \varepsilon \leqslant \frac{1}{2}\left(\frac{\|w^\star\|_2}{2}\right)^{2/L},$$

which holds by (32). By the triangular inequality, we now have

$$\prod_{k=1}^L \sigma_k \left(1 - \frac{(L-1)\varepsilon}{\left(\frac{\|w^\star\|_2}{2}\right)^{2/L}}\right) \leqslant \|\sigma_1 \ldots \sigma_L v_1 u_1^\top \ldots v_L u_L^\top\|_2$$

$$\leqslant \|W_1^\top \ldots W_L^\top\|_2 + \|W_1^\top \ldots W_L^\top - \sigma_1 \ldots \sigma_L v_1 u_1^\top \ldots v_L u_L^\top\|_2$$

$$\leqslant \|W_1^\top \ldots W_L^\top\|_2 + \frac{L\sqrt{\varepsilon}}{\left(\frac{\|w^\star\|_2}{2}\right)^{1/L}} \prod_{k=1}^L \sigma_k,$$

Thus

$$\prod_{k=1}^L \sigma_k \left(1 - \frac{(L-1)\varepsilon}{\left(\frac{\|w^\star\|_2}{2}\right)^{2/L}} - \frac{L\sqrt{\varepsilon}}{\left(\frac{\|w^\star\|_2}{2}\right)^{1/L}}\right) \leqslant \|W_1^\top \ldots W_L^\top\|_2 .$$

Using (31) and (32),

$$1 - \frac{(L-1)\varepsilon}{\left(\frac{\|w^\star\|_2}{2}\right)^{2/L}} - \frac{L\sqrt{\varepsilon}}{\left(\frac{\|w^\star\|_2}{2}\right)^{1/L}} \geqslant 1 - \frac{1}{8} - \frac{1}{8} = \frac{3}{4} .$$

Moreover, the product of the singular values can be lower-bounded by the smallest one to the power $L$, so

$$\frac{3(\min \sigma_k)^L}{4} \leqslant \|W_1^\top \ldots W_L^\top\|_2 .$$

Let us show that, for $t$ large enough,

$$\min \sigma_k \leqslant (2\|w^\star\|_2)^{1/L} - \varepsilon .$$

If it were not the case, we would have

$$\|W_1^\top \ldots W_L^\top\|_2 \geqslant \frac{3\left((2\|w^\star\|_2)^{1/L} - \varepsilon\right)^L}{4}$$

$$= \frac{3}{2}\|w^\star\|_2 \left(1 - \frac{\varepsilon}{(2\|w^\star\|_2)^{1/L}}\right)^L$$

$$\geqslant \frac{3}{2}\|w^\star\|_2 \left(1 - \frac{2L\varepsilon}{(2\|w^\star\|_2)^{1/L}}\right)$$

$$\geqslant \frac{9}{8}\|w^\star\|_2 , \tag{34}$$

where the second inequality holds by Lemma 4 since

$$\frac{\varepsilon}{(2\|w^\star\|_2)^{1/L}} \leqslant \frac{1}{2}\,,$$

by (31), and the third since

$$\frac{2L\varepsilon}{(2\|w^\star\|_2)^{1/L}} \leqslant \frac{1}{4} \quad \Leftrightarrow \quad L\varepsilon \leqslant \frac{(2\|w^\star\|_2)^{1/L}}{8}\,,$$

also by (31). Since $9/8 > 1$, the inequality (34) is a contradiction with the fact that $\|W_1^\top \ldots W_L^\top\|_2 = \|w_{\mathrm{prod}}\|_2 \to \|w^\star\|_2$, showing that, for $t$ large enough,

$$\min \sigma_k \leqslant (2\|w^\star\|_2)^{1/L} - \varepsilon\,.$$

Thus

$$\min \sigma_k^2 \leqslant (2\|w^\star\|_2)^{2/L} + \varepsilon^2 - 2\varepsilon(2\|w^\star\|_2)^{1/L}\,.$$

By Lemma 2, for all $k \in \{1, \ldots, L\}$,

$$\begin{aligned}
\sigma_k^2 &\leqslant \min \sigma_j^2 + \varepsilon \\
&\leqslant (2\|w^\star\|_2)^{2/L} + \varepsilon^2 - 2\varepsilon(2\|w^\star\|_2)^{1/L} + \varepsilon \\
&\leqslant (2\|w^\star\|_2)^{2/L}\,,
\end{aligned}$$

since

$$\varepsilon^2 - 2\varepsilon(2\|w^\star\|_2)^{1/L} + \varepsilon \leqslant 0 \quad \Leftrightarrow \quad \varepsilon \leqslant 2(2\|w^\star\|_2)^{1/L} - 1\,,$$

which holds true by Assumption $(A_2)$ since

$$\varepsilon \leqslant 1 \leqslant 2(2\|w^\star\|_2)^{1/L} - 1$$

since $\|w^\star\|_2 \geqslant 1$.

## B.6  Proof of Corollary 2

The lower bound unfolds by definition of $S_{\min}$ since $\mathcal{W}^{\mathrm{SI}}$ is a minimizer of the empirical risk. To obtain the upper bound, we proceed similarly to the proof of Theorem 2. For simplicity, we denote $\mathcal{W} = \mathcal{W}^{\mathrm{SI}}$ in the remainder of the proof. We have, as in the proof of Theorem 2,

$$S(\mathcal{W})^2 = \lim_{\xi \to 0} \sup_{\|W_k - \tilde{W}_k\|_F \leqslant \xi} \frac{\sum_{k=1}^L \|\nabla_k R^L(\mathcal{W}) - \nabla_k R^L(\tilde{\mathcal{W}})\|_F^2}{\sum_{k=1}^L \|W_k - \tilde{W}_k\|_F^2}\,, \tag{35}$$

with

$$\begin{aligned}
\Delta_k &:= \nabla_k R^L(\mathcal{W}) - \nabla_k R^L(\tilde{\mathcal{W}}) \\
&= -W_{k+1}^\top \ldots W_L^\top \nabla R^1(\tilde{w}_{\mathrm{prod}})^\top W_1^\top \ldots W_{k-1}^\top + \mathcal{O}(\xi^2)\,,
\end{aligned}$$

and

$$\nabla R^1(\tilde{w}_{\mathrm{prod}}) = \frac{2}{n} X^\top X \Big( \sum_{k=1}^L W_1^\top \ldots W_{k-1}^\top (\tilde{W}_k^\top - W_k^\top) W_{k+1}^\top \ldots W_L^\top \Big) + \mathcal{O}(\xi^2)\,.$$

We recall that, as in the proof of Theorem 2, the notation $\mathcal{O}$ is taken with respect to the limit when $\xi \to 0$, everything else being fixed. By subadditivity of the operator norm and by Corollary 1, we have

$$\begin{aligned}
\|\nabla R^1(\tilde{w}_{\mathrm{prod}})\|_2 &\leqslant 2\Lambda \sum_{k=1}^L \|W_1\|_2 \ldots \|W_{k-1}\|_2 \|\tilde{W}_k - W_k\|_2 \|W_{k+1}\|_2 \ldots \|W_L\|_2 + \mathcal{O}(\xi^2) \\
&\leqslant 2\Lambda(2\|w^\star\|_2)^{(L-1)/L} \sum_{k=1}^L \|W_k - \tilde{W}_k\|_2 + \mathcal{O}(\xi^2)\,. \tag{36}
\end{aligned}$$

Moving on to bounding the Frobenius norm of $\Delta_k$, we observe that the dominating term of this matrix decomposes as a rank-one matrix. Thus, again by subadditivity of the operator norm, equation (36) and Corollary 1,

$$
\begin{aligned}
\|\Delta_k\|_F &= \|W_{k+1}^\top \ldots W_L^\top\|_2 \|\nabla R^1(\tilde{w}_{\text{prod}})^\top W_1^\top \ldots W_{k-1}^\top\|_2 + \mathcal{O}(\xi^2) \\
&\leqslant \|W_{k+1}\|_2 \ldots \|W_L\|_2 \|\nabla R^1(\tilde{w}_{\text{prod}})\|_2 \|W_1\|_2 \ldots \|W_{k-1}\|_2 + \mathcal{O}(\xi^2) \\
&\leqslant 2\Lambda (2\|w^\star\|_2)^{2(L-1)/L} \sum_{k=1}^{L} \|W_k - \tilde{W}_k\|_2 + \mathcal{O}(\xi^2).
\end{aligned}
$$

Then

$$
\begin{aligned}
\|\Delta_k\|_F^2 &\leqslant 4\Lambda^2 (2\|w^\star\|_2)^{4(L-1)/L} \Big( \sum_{k=1}^{L} \|W_k - \tilde{W}_k\|_2 \Big)^2 + \mathcal{O}(\xi^3) \\
&\leqslant 4L\Lambda^2 (2\|w^\star\|_2)^{4(L-1)/L} \sum_{k=1}^{L} \|W_k - \tilde{W}_k\|_2^2 + \mathcal{O}(\xi^3) \\
&\leqslant 4L\Lambda^2 (2\|w^\star\|_2)^{4(L-1)/L} \sum_{k=1}^{L} \|W_k - \tilde{W}_k\|_F^2 + \mathcal{O}(\xi^3).
\end{aligned}
$$

Thus, by (35),

$$
S(\mathcal{W})^2 \leqslant \lim_{\xi \to 0} 4L^2 \Lambda^2 (2\|w^\star\|_2)^{4(L-1)/L} + \mathcal{O}(\xi) \leqslant 2^6 L^2 \Lambda^2 \|w^\star\|_2^{4-\frac{4}{L}}.
$$

Therefore

$$
S(\mathcal{W}) \leqslant 8L\Lambda \|w^\star\|_2^{2-\frac{2}{L}},
$$

which concludes the proof by the second lower bound on $S_{\min}$ of Theorem 2, where here $\|p\| = 1$ and $a \geqslant \lambda > 0$.

### B.7   Proof of Theorem 4

To alleviate notations, we omit in this proof to write the explicit dependence of parameters on time. Starting from (5), since the gradient decomposes as a rank-one matrix, we have

$$
\begin{aligned}
\|\nabla_k R^L(\mathcal{W})\|_F^2 &= \|W_{k+1}^\top \ldots W_L^\top p\|_2^2 \|\nabla R^1(w_{\text{prod}})^\top W_1^\top \ldots W_{k-1}^\top\|_2^2 \\
&\geqslant \sigma_{\min}^2(W_{k+1}^\top \ldots W_L^\top) \|p\|_2^2 \sigma_{\min}^2(W_1^\top \ldots W_{k-1}^\top) \|\nabla R^1(w_{\text{prod}})\|_2^2
\end{aligned}
$$

By Lemma 8,
$$
\|\nabla R^1(w_{\text{prod}})\|_2^2 \geqslant 4\lambda (R^L(\mathcal{W}) - R_{\min}).
$$
Recall the notation introduced in Section 5

$$
\Pi_{L:k} := W_L \ldots W_k \quad \text{and} \quad \Pi_{k:1} := W_k \ldots W_1.
$$

Then
$$
\|\nabla_k R^L(\mathcal{W})\|_F^2 \geqslant 4\lambda \|p\|_2^2 \sigma_{\min}^2(\Pi_{L:k+1}) \sigma_{\min}^2(\Pi_{k-1:1})(R^L(\mathcal{W}) - R_{\min}). \tag{37}
$$
We now let $u > 0$ (whose value will be specified next), and

$$
t^* = \inf \left\{ t \in \mathbb{R}_+, \exists k \in \{1, \ldots, L\}, \|\theta_k\|_F > \frac{1}{64} \exp(-2s^2 - 4u) \right\}.
$$

Let us lower bound the minimum singular values of $\Pi_{k:}$ and $\Pi_{:k}$ uniformly for $t \in [0, t^\star]$ by Lemma 3. By definition of $t^\star$, for $t \leqslant t^\star$ and for any $k \in \{1, \ldots, L\}$, $\|\theta_k\|_2 \leqslant \frac{1}{64} \exp(-2s^2 - 4u)$. Therefore, renaming $\tilde{C}_1$ to $\tilde{C}_4$ the constants of Lemma 3, and taking $u = \tilde{C}_3$, we get that, if

$$
L \geqslant \max \left( \tilde{C}_1, \left( \frac{\tilde{C}_3}{\tilde{C}_4} \right)^4 \right), \quad d \geqslant \tilde{C}_2, \tag{38}
$$

then, with probability at least

$$
1 - 8 \exp \left( -\frac{d\tilde{C}_3^2}{32s^2} \right),
$$

it holds for $t \leqslant t^\star$ and for all $k \in \{1, \ldots, L\}$ that

$$\|\Pi_{k:1}\|_2 \leqslant 4 \exp \left( \frac{s^2}{2} + \tilde{C}_3 \right), \tag{39}$$

and

$$\sigma_{\min}(\Pi_{k:1}) \geqslant \frac{1}{4} \exp \left( -\frac{2s^2}{d} - \tilde{C}_3 \right).$$

By symmetry, the same statement holds for $\Pi_{L:k}$ instead of $\Pi_{k:1}$ with the same probability. By the union bound, the event

$$E_1 := \bigcap_{1 \leqslant k \leqslant L} \left\{ \|\Pi_{k:1}\|_2, \|\Pi_{L:k}\|_2 \leqslant 4 \exp \left( \frac{s^2}{2} + \tilde{C}_3 \right) \right.$$

$$\left. \text{and } \sigma_{\min}(\Pi_{k:1}), \sigma_{\min}(\Pi_{L:k}) \geqslant \frac{1}{4} \exp \left( -\frac{2s^2}{d} - \tilde{C}_3 \right) \right\}$$

holds with probability at least

$$1 - 16 \exp \left( -\frac{d\tilde{C}_3^2}{32 s^2} \right).$$

Let now

$$E_2 := \left\{ R^L(\mathcal{W}(0)) - R_{\min} \leqslant \frac{C_4 \lambda^2 \|p\|_2^2}{\Lambda} \right\}$$

and $E_3$ the event that the gradient flow dynamics converge to a global minimizer $\mathcal{W}^{\mathrm{RI}}$ of the risk satisfying the statement (8). We show next that, if $E_1$ and $E_2$ hold, then $E_3$ must hold, which shall conclude the proof of the Theorem with

$$C_1 = \max \left( \tilde{C}_1, \left( \frac{\tilde{C}_3}{\tilde{C}_4} \right)^4 \right), \quad C_2 = \tilde{C}_2, \quad C_3 = \frac{\tilde{C}_3^2}{32 s^2},$$

$$C_4 = 2^{-36} \exp \left( -4s^2 - \frac{16 s^2}{\tilde{C}_2} - 20\tilde{C}_3 \right), \quad C_5 = \frac{1}{64} \exp(-2s^2 - 4\tilde{C}_3). \tag{40}$$

Under $E_1$, we have

$$\sigma_{\min}^2(\Pi_{L:k+1}) \sigma_{\min}^2(\Pi_{k-1:1}) \geqslant \frac{1}{2^8} \exp \left( -\frac{8s^2}{d} - 4\tilde{C}_3 \right),$$

thus by (37)

$$\|\nabla_k R^L(\mathcal{W})\|_F^2 \geqslant \frac{1}{2^6} \exp \left( -\frac{8s^2}{d} - 4\tilde{C}_3 \right) \lambda \|p\|_2^2 (R^L(\mathcal{W}) - R_{\min}).$$

Therefore we get the PL condition, for $t \leqslant t^\star$,

$$\sum_{k=1}^L \left\| \nabla_k R^L(\mathcal{W}) \right\|_F^2 \geqslant \mu (R^L(\mathcal{W}) - R_{\min}),$$

with

$$\mu := \frac{1}{2^6} \exp \left( -\frac{8s^2}{d} - 4\tilde{C}_3 \right) \lambda \|p\|^2 L.$$

By Lemma 7, this implies that, for $t \leqslant t^\star$,

$$R^L(\mathcal{W}) - R_{\min} \leqslant (R^L(\mathcal{W}(0)) - R_{\min}) e^{-\mu t}. \tag{41}$$

Let us now show that $t^\star = \infty$. We have, since $\theta(0) = 0$ and by definition of the gradient flow,

$$\|\theta_k\|_F = \|\theta_k - \theta_k(0)\|_F \leqslant L \int_0^t \|\nabla_k R^L(\mathcal{W}(\tau))\|_F d\tau. \tag{42}$$

We now upper bound the gradient as follows: starting again from (5),

$$\|\nabla_k R^L(\mathcal{W})\|_F = \|W_{k+1}^\top \ldots W_L^\top p\|_2 \|\nabla R^1(w_{\mathrm{prod}})^\top W_1^\top \ldots W_{k-1}^\top\|_2$$

$$\leqslant \|W_{k+1}^\top \ldots W_L^\top\|_2 \|p\|_2 \|W_1^\top \ldots W_{k-1}^\top\|_2 \|\nabla R^1(w_{\mathrm{prod}})\|_2.$$

By Lemma 8, we get

$$\|\nabla_k R^L(\mathcal{W})\|_F \leqslant 2\sqrt{\Lambda}\|W_{k+1}^\top \ldots W_L^\top\|_2\|p\|_2\|W_1^\top \ldots W_{k-1}^\top\|_2\sqrt{R^L(\mathcal{W}) - R_{\min}}\,. \qquad (43)$$

By (39) and (41), for $t \leqslant t^\star$,

$$\|\nabla_k R^L(\mathcal{W})\|_F \leqslant 16\exp(s^2 + 2\tilde{C}_3)\sqrt{\Lambda}\|p\|_2\sqrt{R^L(\mathcal{W}(0)) - R_{\min}}\, e^{-\frac{\mu}{2}t}\,. \qquad (44)$$

Plugging this into (42), we get, for $t \leqslant t^\star$,

$$
\begin{aligned}
\|\theta_k\|_F &\leqslant 16\exp(s^2 + 2\tilde{C}_3)\sqrt{\Lambda}\|p\|_2\sqrt{R^L(\mathcal{W}(0)) - R_{\min}}\, L \int_0^t e^{-\frac{\mu}{2}\tau}d\tau \\
&\leqslant \frac{32\exp(s^2 + 2\tilde{C}_3)\sqrt{\Lambda}\|p\|_2\sqrt{R^L(\mathcal{W}(0)) - R_{\min}}\, L}{\mu} \\
&= \frac{2^{11}\exp(s^2 + \frac{8s^2}{d} + 6\tilde{C}_3)\sqrt{\Lambda}\sqrt{R^L(\mathcal{W}(0)) - R_{\min}}}{\lambda\|p\|_2}\,,
\end{aligned}
$$

where the last equality comes from the definition of $\mu$. By $E_2$ and by definition (40) of $C_4$, for $t \leqslant t^\star$,

$$\|\theta_k\|_F \leqslant 2^{11}\exp(s^2 + \frac{8s^2}{d} + 6\tilde{C}_3)\sqrt{C_3} \leqslant \frac{1}{128}\exp(-2s^2 - 4\tilde{C}_3)\,.$$

If we had $t^\star < \infty$, we would have, by definition of $t^\star$, $\|\theta_k(t^\star)\|_F \geqslant \frac{1}{64}\exp(-2s^2 - 4\tilde{C}_3)$. This contradicts the equation above, showing that $t^\star = \infty$. By (41), this implies convergence of the risk to its minimum. We also see by (44) that $\nabla_k R^L(\mathcal{W})$ is integrable, so $\mathcal{W}$ has a limit as $t$ goes to infinity. This limit $\mathcal{W}^{\mathrm{RI}}$ is a minimizer of the risk and satisfies the condition (8) by definition of $t^\star = \infty$.

**Extension to multivariate regression.** We emphasize that a very similar proof holds in the case of multivariate regression, where the neural network is defined as the linear map from $\mathbb{R}^d$ to $\mathbb{R}^d$

$$x \mapsto W_L \ldots W_1 x\,,$$

and we now aim at minimizing the mean squared error loss

$$R^L(\mathcal{W}) = \frac{1}{n}\|Y - W_L \ldots W_1 X^\top\|_2^2\,,$$

where $X, Y \in \mathbb{R}^{n \times d}$. In this case, as shown in Zou et al. (2020); Sander et al. (2022), the following bounds on the gradient hold:

$$\|\nabla_k R^L(\mathcal{W})\|_F^2 \geqslant 4\lambda\sigma_{\min}^2(\Pi_{L:k+1})\sigma_{\min}^2(\Pi_{k-1:1})(R(\mathcal{W}) - R_{\min})\,,$$

and

$$\|\nabla_k R^L(\mathcal{W})\|_F^2 \leqslant 4\Lambda\|\Pi_{L:k+1}\|_2^2\|\Pi_{k-1:1}\|_2^2(R(\mathcal{W}) - R_{\min})\,.$$

Comparing with (37) and (43), the only difference is the absence of $\|p\|_2$ here. From there, the same computations as above hold (taking $\|p\|_2 = 1$), and give the following result.

**Theorem 5.** *There exist $C_1, \ldots, C_5 > 0$ depending only on $s$ such that, if $L \geqslant C_1$ and $d \geqslant C_2$, then, with probability at least*

$$1 - 16\exp(-C_3 d)\,,$$

*if*

$$R^L(\mathcal{W}(0)) - R_{\min} \leqslant \frac{C_4\lambda^2\|p\|_2^2}{\Lambda}\,,$$

*the gradient flow dynamics (4) converge to a global minimizer $\mathcal{W}^{\mathrm{RI}}$ of the risk. Furthermore, the minimizer $\mathcal{W}^{\mathrm{RI}}$ satisfies*

$$W_k^{\mathrm{RI}} = I + \frac{s}{\sqrt{Ld}}N_k + \frac{1}{L}\theta_k^{\mathrm{RI}} \quad \text{with} \quad \|\theta_k^{\mathrm{RI}}\|_F \leqslant C_5\,, \quad 1 \leqslant k \leqslant L\,.$$

## B.8 Proof of Lemma 3

We begin by proving the result when $\theta = 0$, then explain how to extend to any $\theta$ in a ball around $0$. Recall that $\Pi_{k:1}$ denotes the product of weight matrices up to the $k$-th. Let

$$M_{s,u} = 2 \exp\left(\frac{s^2}{2} + u\right), \quad m_{s,u} = \frac{1}{2} \exp\left(-\frac{2s^2}{d} - u\right),$$

and denote by $A$ the event that for all $k$, $\|\Pi_{k:1}\|_2 \leqslant M_{s,u}$ and $B$ the event that for all $k$, $\sigma_{\min}(\Pi_{k:1}) \geqslant m_{s,u}$. We begin by bounding the probability of $\bar{A}$, then bound the probability of $\bar{B} \cap A$.

**Useful identities.** To this aim, we first introduce some notations and derive some useful identities. We let $\mathcal{F}_k$ the filtration generated by the random variables $N_1, \ldots, N_k$. For some $h_0 \in \mathbb{R}^d$, we let

$$h_k = \Pi_{k:1} h_0 = \left(I + \frac{s}{\sqrt{Ld}} N_k\right) \ldots \left(I + \frac{s}{\sqrt{Ld}} N_1\right) h_0$$

In particular, $h_{k+1}$ and $h_k$ are related through

$$h_{k+1} = \left(I + \frac{s}{\sqrt{Ld}} N_{k+1}\right) h_k.$$

Taking the squared norm, we get

$$\|h_{k+1}\|_2^2 = \|h_k\|_2^2 + \frac{s^2}{Ld}\|N_{k+1} h_k\|_2^2 + \frac{2s}{\sqrt{Ld}} h_k^\top N_{k+1} h_k.$$

Dividing by $\|h_k\|_2^2$ and taking the logarithm leads to

$$\ln(\|h_{k+1}\|_2^2) = \ln(\|h_k\|_2^2) + \ln\left(1 + \frac{s^2\|N_{k+1}h_k\|_2^2}{Ld\|h_k\|_2^2} + \frac{2s h_k^\top N_{k+1} h_k}{\sqrt{Ld}\|h_k\|_2^2}\right). \tag{45}$$

Let

$$Y_{k,1} = \frac{s^2\|N_{k+1}h_k\|_2^2}{Ld\|h_k\|_2^2}, \quad Y_{k,2} = \frac{2s h_k^\top N_{k+1} h_k}{\sqrt{Ld}\|h_k\|_2^2},$$

and $Y_k = Y_{k,1} + Y_{k,2}$. Then, by Lemma 6,

$$Y_{k,1}|\mathcal{F}_k \sim \frac{s^2}{Ld}\chi^2(d) \quad \text{and} \quad Y_{k,2}|\mathcal{F}_k \sim \mathcal{N}\left(0, \frac{4s^2}{Ld}\right). \tag{46}$$

This shows in particular that $Y_{k,1}$ and $Y_{k,2}$ are independent of $N_1, \ldots, N_k$, and depend only on $N_{k+1}$. Then, letting

$$S_{k,1} = \sum_{j=0}^{k-1} Y_{j,1} \quad \text{and} \quad S_{k,2} = \sum_{j=0}^{k-1} Y_{j,2},$$

both are sums of i.i.d. random variables. In particular,

$$S_{L,1} \sim \frac{s^2}{Ld}\chi^2(Ld) \quad \text{and} \quad S_{L,2} \sim \mathcal{N}\left(0, \frac{4s^2}{d}\right).$$

**Bound on $\mathbb{P}(\bar{A})$.** We first bound the deviation of the norm of $h_k = \Pi_{k:1} h_0$ for any fixed $h_0 \in \mathbb{R}^d$, then conclude on the operator norm of $\Pi_{k:1}$ by an $\varepsilon$-net argument. For any (fixed) $h_0 \in \mathbb{R}^d$ and $u > 0$, we have

$$\mathbb{P}\left(\max_{1 \leqslant k \leqslant L} \frac{\|h_k\|_2^2}{\|h_0\|_2^2} \geqslant \exp(s^2 + 2u)\right)$$

$$= \mathbb{P}\left(\max_{1 \leqslant k \leqslant L} \ln(\|h_k\|_2^2) - \ln(\|h_0\|_2^2) \geqslant s^2 + 2u\right)$$

$$= \mathbb{P}\left(\max_{1 \leqslant k \leqslant L} \sum_{j=0}^{k-1} \ln\left(1 + \frac{s^2\|N_{j+1}h_j\|_2^2}{Ld\|h_j\|_2^2} + \frac{2s h_j^\top N_{j+1} h_j}{\sqrt{Ld}\|h_j\|_2^2}\right) \geqslant s^2 + 2u\right)$$

$$\leqslant \mathbb{P}\left(\max_{1 \leqslant k \leqslant L} \sum_{j=0}^{k-1} \frac{s^2\|N_{j+1}h_j\|_2^2}{Ld\|h_j\|_2^2} + \frac{2s h_j^\top N_{j+1} h_j}{\sqrt{Ld}\|h_j\|_2^2} \geqslant s^2 + 2u\right),$$

by using $\ln(1+x) \leqslant x$. Thus

$$\mathbb{P}\Big( \max_{1\leqslant k\leqslant L} \frac{\|h_k\|_2^2}{\|h_0\|_2^2} \geqslant \exp(s^2+2u) \Big) \leqslant \mathbb{P}\Big( \max_{1\leqslant k\leqslant L} S_{k,1} + S_{k,2} \geqslant s^2+2u \Big)$$

$$\leqslant \mathbb{P}\Big( \max_{1\leqslant k\leqslant L} S_{k,1} \geqslant s^2+u \Big) + \mathbb{P}\Big( \max_{1\leqslant k\leqslant L} S_{k,2} \geqslant u \Big)$$

by the union bound. We now study the two deviation probabilities separately. Beginning by the first one, recall that $\frac{Ld}{s^2} S_{L,1}$ follows a $\chi^2(Ld)$ distribution. Chi-squared random variables are sub-exponential, and more precisely satisfy the following property (Ghosh, 2021): if $X \sim \chi^2(c)$ and $u > 0$, then

$$\mathbb{P}(X \geqslant c+u) \leqslant \exp\Big( -\frac{u^2}{4(c+u)} \Big). \tag{47}$$

Since the $S_{k,1}$ are increasing, we have, for $u > 0$,

$$\mathbb{P}\Big( \max_{1\leqslant k\leqslant L} S_{k,1} \geqslant s^2+u \Big) = \mathbb{P}(S_{L,1} \geqslant s^2+u)$$

$$= \mathbb{P}\Big( \frac{Ld}{s^2} S_{L,1} \geqslant Ld + \frac{Ldu}{s^2} \Big)$$

$$\leqslant \exp\Big( -\frac{L^2 d^2 u^2}{4s^4(Ld + \frac{Ldu}{s^2})} \Big) = \exp\Big( -\frac{Ldu^2}{4s^4 + us^2} \Big).$$

Moving on to $S_{k,2}$, we have, for $u, \lambda > 0$,

$$\mathbb{P}\Big( \max_{1\leqslant k\leqslant L} S_{k,2} \geqslant u \Big) = \mathbb{P}\Big( \max_{1\leqslant k\leqslant L} \exp(\lambda S_{k,2}) \geqslant \exp(\lambda u) \Big).$$

Furthermore, $\exp(\lambda S_{k,2})$ is a sub-martingale, since $S_{k,2}$ is a martingale and by Jensen's inequality:

$$\mathbb{E}(\exp(\lambda S_{k+1,2})|\mathcal{F}_k) \geqslant \exp(\lambda \mathbb{E}(S_{k+1,2}|\mathcal{F}_k)) = \exp(\lambda S_{k,2}).$$

Thus, by Doob's martingale inequality,

$$\mathbb{P}\Big( \max_{1\leqslant k\leqslant L} S_{k,2} \geqslant u \Big) \leqslant \mathbb{E}(\exp(\lambda S_{L,2})) \exp(-\lambda u).$$

Furthermore, since $S_{L,2} \sim \mathcal{N}(0, \frac{4s^2}{d})$,

$$\mathbb{E}(\exp(\lambda S_{L,2})) = \exp\Big( \frac{4s^2\lambda^2}{d} \Big).$$

Therefore,

$$\mathbb{P}\Big( \max_{1\leqslant k\leqslant L} S_{k,2} \geqslant u \Big) \leqslant \exp\Big( -\lambda u + \frac{4s^2\lambda^2}{d} \Big).$$

This quantity is minimal for $\lambda = \frac{du}{8s^2}$. We get, for all $u > 0$,

$$\mathbb{P}\Big( \max_{1\leqslant k\leqslant L} S_{k,2} \geqslant u \Big) \leqslant \exp\Big( -\frac{du^2}{16s^2} \Big). \tag{48}$$

Therefore, for any $u > 0$,

$$\mathbb{P}\Big( \max_{1\leqslant k\leqslant L} \frac{\|h_k\|_2^2}{\|h_0\|_2^2} \geqslant \exp(s^2+2u) \Big) \leqslant \exp\Big( -\frac{du^2}{16s^2} \Big) + \exp\Big( -\frac{Ldu^2}{4s^4 + us^2} \Big). \tag{49}$$

To conclude on the operator norm of $\Pi_{k:1}$, consider $\Sigma$ a $1/2$-net of the unit sphere of $\mathbb{R}^d$. By Vershynin (2018, Corollary 4.2.13), it is possible to take such a net of cardinality $5^d$. Let us show that, for any $u \in \mathbb{R}$,

$$\mathbb{P}\Big( \max_{1\leqslant k\leqslant L} \|\Pi_{k:1}\|_2 \geqslant 2\exp\Big( \frac{s^2}{2}+u \Big) \Big) \leqslant \mathbb{P}\Big( \bigcup_{h_0\in\Sigma} \max_{1\leqslant k\leqslant L} \|\Pi_{k:1}h_0\|_2 \geqslant \exp\Big( \frac{s^2}{2}+u \Big) \Big). \tag{50}$$

Indeed, assume that there exists $k$ such that $\|\Pi_{k:1}\|_2 \geqslant 2\exp(\frac{s^2}{2} + u)$. By definition of the operator norm, there exists $x$ on the unit sphere of $\mathbb{R}^d$ such that

$$\|\Pi_{k:1}x\| = \|\Pi_{k:1}\|_2 \,.$$

But then, by definition of $\Sigma$, there exists $h_0 \in \Sigma$ such that $\|x - h_0\|_2 \leqslant 1/2$. Then

$$\|\Pi_{k:1}(x - h_0)\|_2 \leqslant \frac{1}{2}\|\Pi_{k:1}\|_2 \,.$$

Therefore, by the triangular inequality,

$$\|\Pi_{k:1}h_0\|_2 \geqslant \|\Pi_{k:1}x\|_2 - \|\Pi_{k:1}(x - h_0)\|_2 \geqslant \frac{1}{2}\|\Pi_{k:1}\|_2 \,.$$

Then

$$\|\Pi_{k:1}h_0\|_2 \geqslant \frac{1}{2}\|\Pi_{k:1}\|_2 \geqslant \exp\left(\frac{s^2}{2} + u\right),$$

which proves (50). By the union bound, we conclude that

$$\mathbb{P}(\bar{A}) = \mathbb{P}\Big(\max_{1\leqslant k\leqslant L}\|\Pi_{k:1}\|_2 \geqslant 2\exp\left(\frac{s^2}{2} + u\right)\Big)$$

$$\leqslant \mathbb{P}\Big(\bigcup_{h_0\in\Sigma}\max_{1\leqslant k\leqslant L}\|\Pi_{k:1}h_0\|_2 \geqslant \exp\left(\frac{s^2}{2} + u\right)\Big)$$

$$\leqslant |\Sigma|\mathbb{P}\Big(\max_{1\leqslant k\leqslant L}\|\Pi_{k:1}h_0\|_2 \geqslant \exp\left(\frac{s^2}{2} + u\right)\Big),$$

where now $h_0$ denotes any unit-norm fixed vector in $\mathbb{R}^d$. Thus, by (49),

$$\mathbb{P}(\bar{A}) \leqslant 5^d\mathbb{P}\Big(\max_{1\leqslant k\leqslant L}\|\Pi_{k:1}h_0\|_2^2 \geqslant \exp(s^2 + 2u)\Big)$$

$$\leqslant 5^d\Big(\exp\Big(-\frac{du^2}{16s^2}\Big) + \exp\Big(-\frac{Ldu^2}{4s^4 + us^2}\Big)\Big).$$

This upper bound will be simplified in the conclusion of the proof.

**Bound on $\mathbb{P}(\bar{B} \cap A)$.** We now move on to proving the lower-bound on $\sigma_{\min}(\Pi_{k:1})$. We again use an $\varepsilon$-net argument, as follows. Let $\Sigma$ be an $\varepsilon$-net for $\varepsilon = \frac{m_{s,u}}{M_{s,u}}$. Then

$$\mathbb{P}\Big(A \cap \Big\{\min_{1\leqslant k\leqslant L}\sigma_{\min}(\Pi_{k:1}) \leqslant m_{s,u}\Big\}\Big) \leqslant \mathbb{P}\Big(\bigcup_{h_0\in\Sigma}\min_{1\leqslant k\leqslant L}\|\Pi_{k:1}h_0\|_2 \leqslant 2m_{s,u}\Big).$$

Indeed, assume that $A$ holds, that is $\|\Pi_{k:1}\|_2 \leqslant M_{s,u}$, and that there exists $k$ such that

$$\sigma_{\min}(\Pi_{k:1}) \leqslant m_{s,u} \,.$$

By definition of the singular value, there exists $x$ on the unit sphere of $\mathbb{R}^d$ such that

$$\|\Pi_{k:1}x\|_2 = \sigma_{\min}(\Pi_{k:1}) \,.$$

But then, by definition of $\Sigma$, there exists $h_0 \in \Sigma$ such that $\|x - h_0\|_2 \leqslant \varepsilon$. Then

$$\|\Pi_{k:1}(x - h_0)\|_2 \leqslant \varepsilon\|\Pi_{k:1}\|_2 \,.$$

Under $A$, the right-hand side is smaller than $\varepsilon M_{s,u}$. Therefore, by the triangular inequality,

$$\|\Pi_{k:1}h_0\|_2 \leqslant \|\Pi_{k:1}x\|_2 + \|\Pi_{k:1}(h_0 - x)\|_2 \leqslant m_{s,u} + \varepsilon M_{s,u} = 2m_{s,u} \,.$$

By the union bound, we conclude that

$$\mathbb{P}\Big(A \cap \Big\{\min_{1\leqslant k\leqslant L}\sigma_{\min}(\Pi_{k:1}) \leqslant m_{s,u}\Big\}\Big) \leqslant \mathbb{P}\Big(A \cap \Big\{\bigcup_{h_0\in\Sigma}\min_{1\leqslant k\leqslant L}\|\Pi_{k:1}h_0\|_2 \leqslant 2m_{s,u}\Big\}\Big)$$

$$\leqslant \mathbb{P}\Big(\bigcup_{h_0\in\Sigma}\min_{1\leqslant k\leqslant L}\|\Pi_{k:1}h_0\|_2 \leqslant 2m_{s,u}\Big)$$

$$\leqslant |\Sigma|\mathbb{P}\Big(\min_{1\leqslant k\leqslant L}\|\Pi_{k:1}h_0\|_2 \leqslant 2m_{s,u}\Big)$$

$$= \Big(\frac{2M_{s,u}}{m_{s,u}} + 1\Big)^d\mathbb{P}\Big(\min_{1\leqslant k\leqslant L}\|\Pi_{k:1}h_0\|_2 \leqslant 2m_{s,u}\Big),$$

where the cardinality of the net is given by Vershynin (2018, Corollary 4.2.13). We now take a fixed $h_0 \in \mathbb{R}^d$, and compute

$$\mathbb{P}\Big( \min_{1 \leqslant k \leqslant L} \frac{\|h_k\|_2}{\|h_0\|_2} \leqslant 2m_{s,u} \Big) = \mathbb{P}\Big( \min_{1 \leqslant k \leqslant L} \frac{\|h_k\|_2^2}{\|h_0\|_2^2} \leqslant 4m_{s,u}^2 \Big) .$$

Denote by $E$ the event

$$\bigcap_{1 \leqslant k \leqslant L} \Big\{ Y_{k,2} \geqslant -\frac{1}{2} \Big\} .$$

We have, by (45),

$$\mathbb{P}\Big( \min_{1 \leqslant k \leqslant L} \frac{\|h_k\|_2^2}{\|h_0\|_2^2} \leqslant 4m_{s,u}^2 \Big) = \mathbb{P}\Big( \min_{1 \leqslant k \leqslant L} \ln(\|h_k\|_2^2) - \ln(\|h_0\|_2^2) \leqslant \ln(4m_{s,u}^2) \Big)$$

$$\leqslant \mathbb{P}\Big( \min_{1 \leqslant k \leqslant L} \sum_{j=0}^{k-1} \ln(1 + Y_{j,2}) \leqslant \ln(4m_{s,u}^2) \Big)$$

$$\leqslant \mathbb{P}\Big( \Big\{ \min_{1 \leqslant k \leqslant L} \sum_{j=0}^{k-1} \ln(1 + Y_{j,2}) \leqslant \ln(4m_{s,u}^2) \Big\} \cap E \Big) + \mathbb{P}(\bar{E}) .$$

Using the inequality $\ln(1 + x) \geqslant x - x^2$ for $x \geqslant -1/2$, we obtain

$$\mathbb{P}\Big( \frac{\|h_L\|_2^2}{\|h_0\|_2^2} \leqslant 4m_{s,u}^2 \Big) \leqslant \mathbb{P}\Big( \Big\{ \min_{1 \leqslant k \leqslant L} \sum_{j=0}^{k-1} (Y_{j,2} - Y_{j,2}^2) \leqslant \ln(4m_{s,u}^2) \Big\} \cap E \Big) + \mathbb{P}(\bar{E}) .$$

Thus, by the union bound,

$$\mathbb{P}\Big( \min_{1 \leqslant k \leqslant L} \frac{\|h_L\|_2^2}{\|h_0\|_2^2} \leqslant 4m_{s,u}^2 \Big) \leqslant \mathbb{P}\Big( \min_{1 \leqslant k \leqslant L} \sum_{j=0}^{k-1} (Y_{j,2} - Y_{j,2}^2) \leqslant \ln(4m_{s,u}^2) \Big) + \sum_{k=0}^{L-1} \mathbb{P}\Big( Y_{k,2} < -\frac{1}{2} \Big)$$

$$=: P_1 + P_2 .$$

We handle both terms separately. Beginning by the second term, we have, for $k \in \{1, \ldots L\}$,

$$\mathbb{P}\Big( Y_{k,2} < -\frac{1}{2} \Big) \leqslant \exp\Big( -\frac{Ld}{32s^2} \Big) ,$$

where we used (46) and the tail bound $\mathbb{P}(N \geqslant u) \leqslant e^{-u^2/2}$ if $N \sim \mathcal{N}(0,1)$. Moving on to the first term, we have

$$P_1 = \mathbb{P}\Big( \min_{1 \leqslant k \leqslant L} S_{k,2} - S_{k,3} \leqslant \ln(4m_{s,u}^2) \Big) ,$$

where we let $S_{k,3} = \sum_{j=0}^{k-1} Y_{j,2}^2$. By definition of $m_{s,u}$, we have

$$\ln(4m_{s,u}^2) = -\frac{4s^2}{d} - 2u .$$

Thus we can split the probability into two parts by the union bound:

$$P_1 \leqslant \mathbb{P}\Big( \min_{1 \leqslant k \leqslant L} S_{k,2} \leqslant -u \Big) + \mathbb{P}\Big( \max_{1 \leqslant k \leqslant L} S_{k,3} \geqslant \frac{4s^2}{d} + u \Big) .$$

Let us bound each probability separately. We have, for $u > 0$,

$$\mathbb{P}\Big( \min_{1 \leqslant k \leqslant L} S_{k,2} \leqslant -u \Big) = \mathbb{P}\Big( \max_{1 \leqslant k \leqslant L} -S_{k,2} \geqslant u \Big) = \mathbb{P}\Big( \max_{1 \leqslant k \leqslant L} S_{k,2} \geqslant u \Big) \leqslant \exp\Big( -\frac{du^2}{16s^2} \Big) ,$$

by symmetry of $S_{k,2}$ and by (48). Moving on to $S_{k,3}$, we have that

$$Y_{j,2}^2 | \mathcal{F}_j \sim \frac{4s^2}{Ld} \mathcal{N}(0,1)^2 = \frac{4s^2}{Ld} \chi^2(1) ,$$

thus
$$\frac{Ld}{4s^2}S_{k,3} \sim \chi^2(L).$$

By monotonicity of $S_{k,3}$ and by (47),

$$\mathbb{P}\Big(\max_{1\leqslant k\leqslant L}S_{k,3} \geqslant \frac{4s^2}{d}+u\Big) = \mathbb{P}\Big(S_{L,3} \geqslant \frac{4s^2}{d}+u\Big)$$

$$= \mathbb{P}\Big(\frac{Ld}{4s^2}S_{L,3} \geqslant L + \frac{Ldu}{4s^2}\Big)$$

$$\leqslant \exp\Big(-\frac{L^2d^2u^2}{64s^4(L+\frac{Ldu}{4s^2})}\Big) = \exp\Big(-\frac{Ld^2u^2}{16(4s^2+du)}\Big).$$

Putting everything together, we proved that

$$\mathbb{P}(\bar{B}\cap A) \leqslant \Big(\frac{2M_{s,u}}{m_{s,u}}+1\Big)^d(P_1+P_2) = \exp\Big(d\ln\Big(\frac{2M_{s,u}}{m_{s,u}}+1\Big)\Big)(P_1+P_2),$$

with

$$P_1 \leqslant \exp\Big(-\frac{du^2}{16s^2}\Big) + \exp\Big(-\frac{Ld^2u^2}{16(4s^2+du)}\Big)$$

and

$$P_2 \leqslant L\exp\Big(-\frac{Ld}{32s^2}\Big).$$

**Bounding the probability of failure.**  Putting together the two main bounds we showed, we have
$$\mathbb{P}(\bar{A}\cup\bar{B}) = \mathbb{P}(\bar{A}) + \mathbb{P}(\bar{B}\cup A)$$

$$\leqslant 5^d\Big(\exp\Big(-\frac{du^2}{16s^2}\Big)+\exp\Big(-\frac{Ldu^2}{4s^4+us^2}\Big)\Big)$$

$$+ \exp\Big(d\ln\Big(\frac{2M_{s,u}}{m_{s,u}}+1\Big)\Big)\Big(\exp\Big(-\frac{du^2}{16s^2}\Big)+\exp\Big(-\frac{Ld^2u^2}{16(4s^2+du)}\Big)+L\exp\Big(-\frac{Ld}{32s^2}\Big)\Big).$$

This expression is valid for all $u,s > 0$ and $L,d \geqslant 1$. We now simplify the expression of our upper bound by algebraic computations using the assumptions of the Theorem. To somewhat alleviate the technicality of the computations, we stop at this point tracking some of the explicit constants and let $C$ denote a positive absolute constant that might vary from equality to equality. We show next that the conditions

$$L \geqslant C, \quad d \geqslant C, \quad u \geqslant C\max(s^2, s^{-2/3}), \quad u \leqslant CL^{1/4} \tag{51}$$

imply that

$$\mathbb{P}(\bar{A}\cup\bar{B}) \leqslant 8\exp\Big(-\frac{du^2}{32s^2}\Big).$$

This shall conclude the proof of the Lemma with

$$C_1 = C, \quad C_2 = C, \quad C_3 = C\max(s^2, s^{-2/3}), \quad C_4 = C.$$

First note that the conditions (51) imply that

$$u \geqslant Cs \quad \text{and} \quad u \geqslant C. \tag{52}$$

We study the terms of the bound on $\mathbb{P}(\bar{A}\cup\bar{B})$ one by one. First, we have by (52) that

$$5^d\exp\Big(-\frac{du^2}{16s^2}\Big) = \exp\Big(d\ln(5)-\frac{du^2}{16s^2}\Big) \leqslant \exp\Big(-\frac{du^2}{32s^2}\Big).$$

Next,

$$5^d\exp\Big(-\frac{Ldu^2}{4s^4+us^2}\Big) \leqslant 5^d\exp\Big(-\frac{Ldu^2}{8us^2}\Big)$$

$$= 5^d\exp\Big(-\frac{Ldu}{8s^2}\Big)$$

$$= \exp\Big(d\ln(5)-\frac{Ldu}{8s^2}\Big)$$

$$\leqslant \exp\Big(-\frac{Ldu}{16s^2}\Big),$$

where the first inequality uses $u \geqslant s^2$ by (51), and the last one uses that $\frac{Lu}{s^2} \geqslant C$. This is true since $L \geqslant C$ and $u \geqslant Cs^2$ by (51). Then we can bound this term by

$$\exp\left(-\frac{du^2}{32s^2}\right),$$

since $u \leqslant CL$ which is implied by the assumption $u \leqslant CL^{1/4}$ in (51). Next,

$$\ln\left(\frac{2M_{s,u}}{m_{s,u}} + 1\right) = \ln\left(8\exp\left(\frac{s^2}{2} + \frac{2s^2}{d} + 2u\right) + 1\right)$$
$$\leqslant \ln\left(9\exp\left(\frac{s^2}{2} + \frac{2s^2}{d} + 2u\right)\right),$$

where we used that the exponential of a positive term is greater than 1. Then

$$\ln\left(\frac{2M_{s,u}}{m_{s,u}} + 1\right) \leqslant \ln(9) + \frac{s^2}{2} + \frac{2s^2}{d} + 2u \leqslant 3 + 3u, \tag{53}$$

since $u \geqslant s^2$, $d \geqslant 4$ by (51). Thus we can bound the three remaining terms appearing in the bound of $\mathbb{P}(\bar{A} \cup \bar{B})$, as follows:

$$\exp\left(d\ln\left(\frac{2M_{s,u}}{m_{s,u}} + 1\right)\right)\exp\left(-\frac{du^2}{16s^2}\right) \leqslant \exp\left(3d + 3du - \frac{du^2}{16s^2}\right) \leqslant \exp\left(-\frac{du^2}{32s^2}\right),$$

since

$$u^2 \geqslant Cs^2(1 + u).$$

This is the case by (52), which implies that

$$u^2 = \frac{1}{2}u^2 + \frac{1}{2}uu \geqslant \frac{C^2}{2}s^2 + \frac{C}{2}s^2u = \frac{C^2 + C}{2}s^2(1 + u).$$

Next, by (53),

$$\exp\left(d\ln\left(\frac{2M_{s,u}}{m_{s,u}} + 1\right)\right)\exp\left(-\frac{Ld^2u^2}{16(4s^2 + du)}\right) \leqslant \exp\left(3d + 3du - \frac{Ld^2u^2}{16(4s^2 + du)}\right)$$
$$\leqslant \exp\left(3d + 3du - \frac{Ld^2u^2}{32du}\right),$$

since $du \geqslant u \geqslant Cs^2$ by (51). Thus

$$\exp\left(d\ln\left(\frac{2M_{s,u}}{m_{s,u}} + 1\right)\right)\exp\left(-\frac{Ld^2u^2}{16(4s^2 + du)}\right) \leqslant \exp\left(3d + 3du - \frac{Ldu}{32}\right) \leqslant \exp\left(-\frac{Ldu}{64}\right),$$

where the second inequality uses that $L \geqslant C$ by (51), and $Lu \geqslant C$ since $L \geqslant C$ and $u \geqslant C$ by (51) and (52). We can also bound this term by

$$\exp\left(-\frac{du^2}{32s^2}\right),$$

since $L \geqslant Cu/s^2$, using first that $L \geqslant Cu^4$ then that $u \geqslant Cs^{-2/3}$, by (51). Regarding the last term of the bound of $\mathbb{P}(\bar{A} \cup \bar{B})$, we have by (53) that

$$L\exp\left(d\ln\left(\frac{2M_{s,u}}{m_{s,u}} + 1\right)\right)\exp\left(-\frac{Ld}{32s^2}\right) \leqslant L\exp\left(3d + 3du - \frac{Ld}{32s^2}\right) \tag{54}$$

Let us show that this implies that

$$L\exp\left(d\ln\left(\frac{2M_{s,u}}{m_{s,u}} + 1\right)\right)\exp\left(-\frac{Ld}{32s^2}\right) \leqslant L\exp\left(-\frac{\sqrt{L}du^2}{32s^2}\right). \tag{55}$$

This statement is true because the three terms appearing inside the exponential of the right-hand side of (54) can be bounded by $\frac{C\sqrt{L}du^2}{s^2}$. More precisely, we have

$$\frac{Ld}{32s^2} \geqslant \frac{C\sqrt{L}du^2}{s^2} \Leftrightarrow \sqrt{L} \geqslant Cu^2,$$

which holds by (51);

$$3du \leqslant \frac{C\sqrt{L}du^2}{s^2} \Leftrightarrow Cs^2 \leqslant \sqrt{L}u \,,$$

which is implied by $L \geqslant C$ and $u \geqslant Cs^2$ by (51); and

$$3d \leqslant \frac{C\sqrt{L}du^2}{s^2} \Leftrightarrow Cs^2 \leqslant \sqrt{L}u^2 \,,$$

which is implied by $L \geqslant C$ and $u \geqslant Cs$ by (51) and (52). Finally, we use Lemma 5 to bound the right-hand side of (55) by

$$4\exp\Big(-\frac{du^2}{32s^2}\Big)\,.$$

This is possible for $L \geqslant C$ and $\frac{du^2}{32s^2} \geqslant 1$, which is implied by $d \geqslant C$ and $u \geqslant Cs$, by (51) and (52). Collecting everything, we bounded $\mathbb{P}(\bar{A} \cup \bar{B})$ by

$$8\exp\Big(-\frac{du^2}{32s^2}\Big)\,,$$

which concludes the proof when $\theta = 0$.

**Summary when $\theta = 0$.** In summary, we proved so far the following result: there exist $C_1, \dots, C_4 > 0$ depending only on $s$ such that, if

$$L \geqslant C_1\,, \quad d \geqslant C_2\,, \quad u \in [C_3, C_4 L^{1/4}]\,,$$

then, with probability at least

$$1 - 8\exp\Big(-\frac{du^2}{32s^2}\Big)\,,$$

it holds for all $k \in \{1, \dots, L\}$ that

$$\Big\|\Big(I + \frac{s}{\sqrt{Ld}}N_k\Big)\dots\Big(I + \frac{s}{\sqrt{Ld}}N_1\Big)\Big\|_2 \leqslant m_{s,u}\,,$$

and

$$\sigma_{\min}\Big(\Big(I + \frac{s}{\sqrt{Ld}}N_k\Big)\dots\Big(I + \frac{s}{\sqrt{Ld}}N_1\Big)\Big) \geqslant M_{s,u}\,.$$

**Conclusion for arbitrary $\theta$.** The conclusion is a direct application of Lemma 9, with $W_k = I + \frac{s}{\sqrt{Ld}}N_k$. The size of the admissible ball for $\theta$ is

$$\frac{m_{s,u}^2}{4M_{s,u}^2} = \frac{1}{64}\exp\Big(-s^2 - \frac{4s^2}{d} - 4u\Big) \geqslant \frac{1}{64}\exp(-2s^2 - 4u)$$

for $d \geqslant 4$, which concludes the proof.

**Comparison with the bounds of Zhang et al. (2022).** Theorem 1 of Zhang et al. (2022) gives an upper bound on the singular values of residual networks at initialization, and their Theorem 2 gives a lower bound on the norm of the activations. Comparing with our results, we note two important differences. First, their probability of failure grows quadratically with the depth, whereas ours is independent of depth. This is achieved by a more precise martingale argument making use of Doob's martingale inequality. Second, their lower bound incorrectly assumes that $\chi^2$ random variables are sub-Gaussian (see equation (21) of their paper), while in fact they are only sub-exponential (Ghosh, 2021). Finally, their upper bound holds for the product

$$\Big(I + \frac{s}{\sqrt{Ld}}N_k + \frac{1}{L}\theta_k\Big)\dots\Big(I + \frac{s}{\sqrt{Ld}}N_j + \frac{1}{L}\theta_j\Big)$$

for any $1 \leqslant j \leqslant k \leqslant L$, which could seem stronger than our result stated for $j = 1$. In fact, both statements are equivalent, because it is possible to deduce the statement for any $j$ by combining the upper bound and the lower bound for $j = 1$. The precise argument is given in the beginning of the proof of our Lemma 9.

## B.9 Proof of Corollary 3

The beginning of the proof is very similar to the one of Corollary 2. Denoting $\mathcal{W} := \mathcal{W}^{\mathrm{RI}}$, we have

$$S(\mathcal{W})^2 = \lim_{\xi \to 0} \sup_{\|W_k - \tilde{W}_k\|_F \leqslant \xi} \frac{\sum_{k=1}^L \|\nabla_k R^L(\mathcal{W}) - \nabla_k R^L(\tilde{\mathcal{W}})\|_F^2}{\sum_{k=1}^L \|W_k - \tilde{W}_k\|_F^2}, \tag{56}$$

with

$$\begin{aligned}
\Delta_k &:= \nabla_k R^L(\mathcal{W}) - \nabla_k R^L(\tilde{\mathcal{W}}) \\
&= -W_{k+1}^\top \ldots W_L^\top p \nabla R^1(\tilde{w}_{\mathrm{prod}})^\top W_1^\top \ldots W_{k-1}^\top + \mathcal{O}(\xi^2),
\end{aligned}$$

and

$$\nabla R^1(\tilde{w}_{\mathrm{prod}}) = \frac{2}{n} X^\top X \Big( \sum_{k=1}^L W_1^\top \ldots W_{k-1}^\top (\tilde{W}_k^\top - W_k^\top) W_{k+1}^\top \ldots W_L^\top p \Big) + \mathcal{O}(\xi^2).$$

At this point, the proofs diverge. We have, by subadditivity of the operator norm,

$$\|\nabla R^1(\tilde{w}_{\mathrm{prod}})\|_2 \leqslant 2\Lambda \sum_{k=1}^L \|W_{k-1} \ldots W_1\|_2 \|\tilde{W}_k - W_k\|_2 \|W_L \ldots W_{k+1}\|_2 \|p\|_2 + \mathcal{O}(\xi^2).$$

Let us now briefly recall the outline of the proof of Theorem 4, which will be useful in bounding the quantity above. The proof shows the existence of $\tilde{C}_3$ depending only on $s$ such that, with high probability (which is exactly the probability in the statement of the Theorem), we have for all $t \geqslant 0$ and $k \in \{1, \ldots, L\}$ that

$$\|W_{k-1}(t) \ldots W_1(t)\|_2 \leqslant 4 \exp \Big( \frac{s^2}{2} + \tilde{C}_3 \Big) \quad \text{and} \quad \|W_L(t) \ldots W_{k+1}(t)\|_2 \leqslant 4 \exp \Big( \frac{s^2}{2} + \tilde{C}_3 \Big),$$

as well as

$$\sigma_{\min}(W_k(t) \ldots W_1(t)) \geqslant \frac{1}{4} \exp \Big( -\frac{2s^2}{d} - \tilde{C}_3 \Big). \tag{57}$$

Under this high-probability event, the proof of Theorem 4 shows convergence of the gradient flow to $\mathcal{W} = \mathcal{W}^{\mathrm{RI}}$. In particular, this means that $\mathcal{W}$ also verifies the bounds on the operator norm of the matrix products. We therefore obtain

$$\begin{aligned}
\|\nabla R^1(\tilde{w}_{\mathrm{prod}})\|_2 &\leqslant 2\Lambda \sum_{k=1}^L 4 \exp \Big( \frac{s^2}{2} + \tilde{C}_3 \Big) \|\tilde{W}_k - W_k\|_2 4 \exp \Big( \frac{s^2}{2} + \tilde{C}_3 \Big) \|p\|_2 + \mathcal{O}(\xi^2) \\
&= 32 \exp(s^2 + 2\tilde{C}_3) \Lambda \|p\|_2 \sum_{k=1}^L \|\tilde{W}_k - W_k\|_2 + \mathcal{O}(\xi^2).
\end{aligned}$$

Moving on to bounding the Frobenius norm of $\Delta_k$, we have

$$\begin{aligned}
\|\Delta_k\|_F &= \|W_{k+1}^\top \ldots W_L^\top p\|_2 \|\nabla R^1(\tilde{w}_{\mathrm{prod}})^\top W_1^\top \ldots W_{k-1}^\top\|_2 + \mathcal{O}(\xi^2) \\
&\leqslant \|W_L \ldots W_{k+1}\|_2 \|p\|_2 \|\nabla R^1(\tilde{w}_{\mathrm{prod}})\|_2 \|W_{k-1} \ldots W_1\|_2 + \mathcal{O}(\xi^2) \\
&\leqslant 2^9 \exp(2s^2 + 4\tilde{C}_3) \Lambda \|p\|_2^2 \sum_{k=1}^L \|W_k - \tilde{W}_k\|_2 + \mathcal{O}(\xi^2),
\end{aligned}$$

by bounding the three norms by the expressions given above. Then

$$\begin{aligned}
\|\Delta_k\|_F^2 &\leqslant 2^{18} \exp(4s^2 + 8\tilde{C}_3) \Lambda^2 \|p\|_2^4 \Big( \sum_{k=1}^L \|W_k - \tilde{W}_k\|_2 \Big)^2 + \mathcal{O}(\xi^3) \\
&\leqslant 2^{18} \exp(4s^2 + 8\tilde{C}_3) L\Lambda^2 \|p\|_2^4 \sum_{k=1}^L \|W_k - \tilde{W}_k\|_2^2 + \mathcal{O}(\xi^3). \\
&\leqslant 2^{18} \exp(4s^2 + 8\tilde{C}_3) L\Lambda^2 \|p\|_2^4 \sum_{k=1}^L \|W_k - \tilde{W}_k\|_F^2 + \mathcal{O}(\xi^3).
\end{aligned}$$

Thus, by (56),

$$S(\mathcal{W})^2 \leqslant \lim_{\xi \to 0} 2^{18} \exp(4s^2 + 8\tilde{C}_3) L^2 \Lambda^2 \|p\|_2^4 + \mathcal{O}(\xi).$$

Therefore

$$S(\mathcal{W}) \leqslant 2^9 \exp(2s^2 + 4\tilde{C}_3) L\Lambda \|p\|_2^2. \tag{58}$$

To conclude, we need to upper bound $\|p\|_2$ by a constant times $\|w^\star\|_2$. To this aim, we leverage the bound from the assumptions of Theorem 4 on the risk at initialization, to show that $\|p\|_2$ cannot be too far away from $\|w^\star\|_2$. More precisely, by Lemma 8, since the covariance matrix $X^\top X$ is full rank,

$$R^L(\mathcal{W}(0)) - R_{\min} = \frac{1}{n} \|X(w_{\text{prod}}(0) - w^\star)\|_2^2$$

$$= \frac{1}{n} (w_{\text{prod}}(0) - w^\star)^\top X^\top X (w_{\text{prod}}(0) - w^\star)$$

$$\geqslant \lambda \|w_{\text{prod}}(0) - w^\star\|_2^2.$$

Thus

$$\sqrt{R^L(\mathcal{W}(0)) - R_{\min}} \geqslant \sqrt{\lambda} \|w_{\text{prod}}(0) - w^\star\|_2.$$

Then, by the triangular inequality,

$$\|w^\star\|_2 \geqslant \|w_{\text{prod}}(0)\|_2 - \|w_{\text{prod}}(0) - w^\star\|_2$$

$$\geqslant \|W_1^\top(0) \dots W_L^\top(0) p\|_2 - \sqrt{\frac{R^L(\mathcal{W}(0)) - R_{\min}}{\lambda}}$$

$$\geqslant \sigma_{\min}(W_L(0) \dots W_1(0)) \|p\|_2 - \sqrt{\frac{R^L(\mathcal{W}(0)) - R_{\min}}{\lambda}}$$

$$\geqslant \frac{1}{4} \exp\left(-\frac{2s^2}{d} - \tilde{C}_3\right) \|p\|_2 - \sqrt{C_3} \sqrt{\frac{\lambda}{\Lambda}} \|p\|_2,$$

by (57) and by the assumption of Theorem 4 on the risk at initialization. We now note that the value of $C_3$ is given by (40) as

$$C_3 = 2^{-36} \exp\left(-4s^2 - \frac{16s^2}{\tilde{C}_2} - 20\tilde{C}_3\right),$$

where $\tilde{C}_2 \leqslant d$ by (38). Thus

$$\sqrt{C_3}\sqrt{\frac{\lambda}{\Lambda}} \leqslant \sqrt{C_3} = 2^{-18} \exp(-2s^2 - \frac{8s^2}{\tilde{C}_2} - 10\tilde{C}_3) < \frac{1}{4} \exp\left(-\frac{2s^2}{d} - \tilde{C}_3\right).$$

Denoting

$$C' = \frac{1}{4} \exp\left(-\frac{2s^2}{d} - \tilde{C}_3\right) - \sqrt{C_3} \in (0, 1),$$

we therefore obtain that $\|w^\star\|_2 \geqslant C'\|p\|_2$. Therefore, by (58),

$$S(\mathcal{W}) \leqslant 2^9 \exp(2s^2 + 4\tilde{C}_3)(C')^{-2+\frac{2}{L}} L\Lambda \|w^\star\|_2^{2-\frac{2}{L}} \|p\|_2^{\frac{2}{L}}.$$

Thus, by the second lower bound on $S_{\min}$ from Theorem 2, and since $a \geqslant \lambda > 0$,

$$\frac{S(\mathcal{W})}{S_{\min}} \leqslant 2^8 \exp(2s^2 + 4\tilde{C}_3)(C')^{-2+\frac{2}{L}} \frac{\Lambda}{\lambda},$$

which concludes the proof by setting

$$C := 2^8 \exp(2s^2 + 4\tilde{C}_3)(C')^{-2} \geqslant 2^8 \exp(2s^2 + 4\tilde{C}_3)(C')^{-2+\frac{2}{L}}.$$

## C  Experimental details, additional plots, and additional comments

Our code is available at https://github.com/PierreMarion23/implicit-reg-sharpness. Our framework for experiments is JAX (Bradbury et al., 2018). The experiments take around 3 hours to run on a laptop CPU.

**Setup.** We take $n = 50$, $d = 5$, $L = 10$. The design matrix $X$ is sampled from an isotropic Gaussian distribution. The target $y$ is computed in two steps. First, we compute $y_0 = X w_{\text{true}} + \zeta$, where $w_{\text{true}}$ and $\zeta$ are standard Gaussian vectors. Then, we compute $w_0^\star$ as the optimal regressor of $y_0$ on $X$. Finally, we let $y = y_0 / \|w_0^\star\|$ and $w^\star = w_0^\star / \|w_0^\star\|$. This simplifies the expressions of our bounds by having $w^\star$ of unit norm. All Gaussian random variables are independent.

**Details of Figure 1.** We consider a Gaussian initialization of the weight matrices, where the scale of the initialization (x-axis of some the graphs) is the standard deviation of the entries. All weight matrices are $d \times d$, except the last one which is $1 \times d$. The square distance to the optimal regressor corresponds to $\|w_{\text{prod}} - w^\star\|_2^2$. The largest eigenvalue of the Hessian is computed by a power iteration method, stopped after 20 iterations. In Figures 1a and 1b, the $95\%$ confidence intervals are plotted. The number of gradient steps and number of independent repetitions depend on the learning rate, and are given below.

| Learning rate | Number of steps | Number of repetitions |
|---|---|---|
| 0.005 | 40,000 | 20 |
| 0.02 | 10,000 | 20 |
| 0.07 | 4,000 | 20 |
| 0.1 | 4,000 | 20 |
| 0.2 | 2,000 | 80 |

For large values of the initialization scale, it may happen that the gradient descent diverges. Figure 3 shows the probability of divergence depending on the initialization scale and the learning rate.

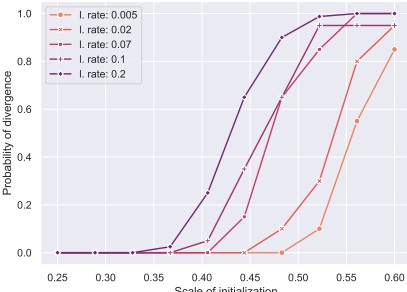

Figure 3: Probability of divergence of gradient descent for a Gaussian initialization of the weight matrices, depending on the initialization scale and the learning rate.

When the probability of divergence is equal to one, no point is reported in Figure 1. When it is strictly between 0 and 1, the confidence intervals are computed over non-diverging runs.

Figures 1c and 1d show one randomly-chosen run each. The plots are subsampled 5 times for readability, due to the oscillations in Figure 1d.

**Residual initialization.** We now consider the case of a residual initialization as in Section 5. Results are given in Figure 4. The scale of the initialization now corresponds to the hyperparameter $s$ in (2). The projection vector $p \in \mathbb{R}^d$ is a random isotropic vector of unit norm, which does not change during training. For each learning rate, we use $4,000$ steps of gradient descent, and perform 20 independent repetitions. The plots are similar to the case of Gaussian initialization, apart from the fact that the sharpness at initialization is better conditioned.

As previously, for large values of the initialization scale, it may happen that the gradient descent diverges. Figure 5 shows the probability of divergence depending on the initialization scale and the learning rate.

**Details of Figure 2.** The setup is the same as for the residual initialization. For each learning rate, we use $1,000$ steps of gradient descent, and perform 50 independent repetitions. We take $s = 0.25$.

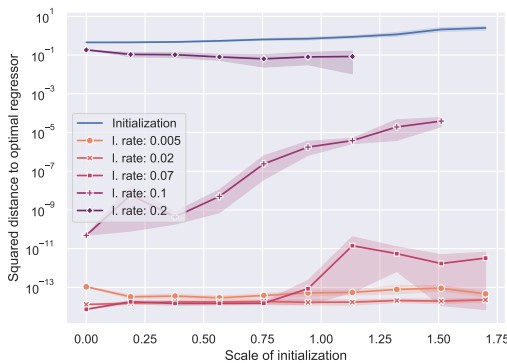
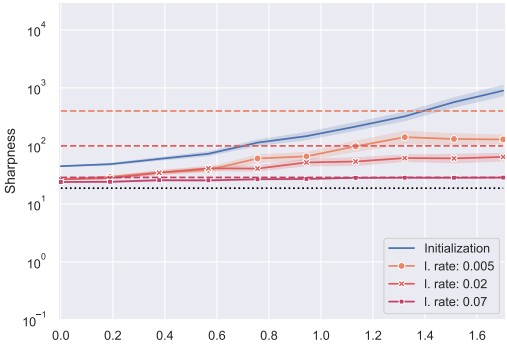

(a) Squared distance of the trained network to the empirical risk minimizer, for various learning rates and initialization scales. Training succeeds when the learning rate is lower than a critical value independent of the initialization scale.

(b) Sharpness at initialization and after training, for various learning rates and initialization scales. For a given learning rate $\eta$, the dashed lines represent the $2/\eta$ threshold. The dotted black line represents the lower bound given in Theorem 1.

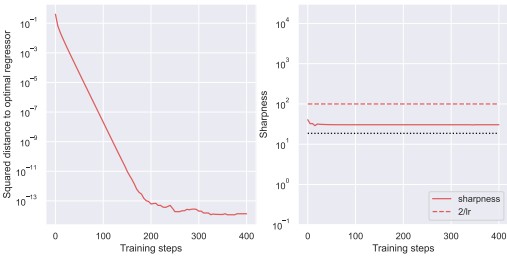
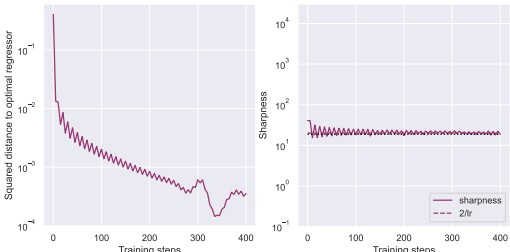

(c) Evolution during training of the squared distance to the empirical risk minimizer and of sharpness, for $\eta = 0.02$ and an initialization scale of $0.5$. The network does not enter edge of stability.

(d) Evolution during training of the squared distance to the empirical risk minimizer and of sharpness, for $\eta = 0.1$ and an initialization scale of $0.5$. The network enters edge of stability.

Figure 4: Training a deep linear network on a univariate regression task with quadratic loss. The initialization is a residual initialization as in Section 5.

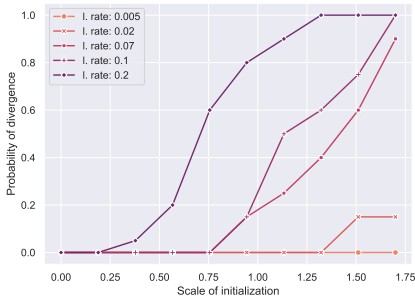

Figure 5: Probability of divergence of gradient descent for a residual initialization of the weight matrices, depending on the initialization scale and the learning rate.

**Underdetermined regression and link to generalization.** Although this is not our original motivation, we note that a simple change to our setting allows to make appear the connection between sharpness and generalization. To this aim, we consider the underdetermined case, where the number of data is lower than the dimension (while keeping the rest of the setup identical). Figure 6 shows in this case a correlation between generalization and sharpness. This suggests that the tools developed in the paper could be used in this case to understand the generalization performance of deep (linear) networks, and we leave this analysis for future work. We also qualitatively observe in Figure 6 a similar connection between learning rate, initialization scale and sharpness as in the case of full-rank data (Figure 1b). We take here $n = 15, d = 20, L = 5$. The number of gradient steps and number of

independent repetitions depend on the learning rate, and are given below. Other technical details are as Figure 1.

| Learning rate | Number of steps | Number of repetitions |
|---|---|---|
| 0.001 | 40,000 | 100 |
| 0.004 | 10,000 | 20 |
| 0.01 | 4,000 | 20 |

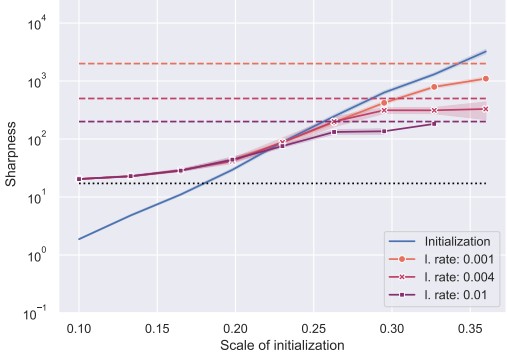 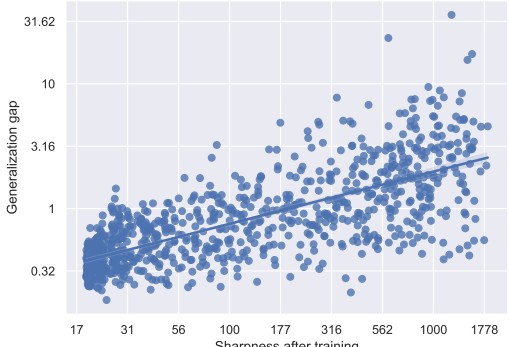

(a) Sharpness at initialization and after training, for various learning rates and initialization scales. We qualitatively observe a similar connection between learning rate, initialization scale and sharpness as in the case of a full rank data matrix (Figure 1b of the paper).

(b) Link between generalization and sharpness. Each dot corresponds to one realization of the experiment (where the randomness comes from the random initialization of the neural network). The plot is shown in log-log scale. The line corresponds to the linear regression of the $\log_{10}$ of the generalization gap on the $\log_{10}$ of the sharpness after training (slope=$0.42 \pm 0.01$, intercept = $-0.96 \pm 0.03$).

Figure 6: Experiment with a deep linear network and a degenerate data covariance matrix, where the number of data $n$ is less than the dimension $d$.

**Non-linear MLPs.** As a first attempt to extend our results to non-linear networks, we consider the case of non-linear MLPs. The non-linearity is GELU (Hendrycks and Gimpel, 2016), a smooth version of ReLU, which we chose because smoothness is necessary in order to compute the sharpness. We qualitatively observe in Figure 7 a similar connection between learning rate, initialization scale and sharpness as for deep linear neural networks (Figure 1b). For large initialization, the sharpness after training plateaus at $2/\eta$, as in the linear case. For small initialization, the sharpness after training is less that $2/\eta$, and is close to the bounds in the linear case (dotted black lines). For this experiment, we consider noiseless data, meaning that $y_0 = X w_{\text{true}}$ (see paragraph "Setup" above for notations). We perform 20 independent repetitions of each experiment. The number of gradient steps depends on the learning rate, and is given below. Other details are as for Figure 1. Finally, we also performed the same experiment in the case of noisy data $y_0 = X w_{\text{true}} + \zeta$ (no plot reported). We observed that the sharpness of the network reaches $2/\eta$, for every learning rate and initialization scale reported in Figure 7. We suspect that this is because the network (over)fits the noise in the data, resulting in a high sharpness.

| Learning rate | Number of steps |
|---|---|
| 0.005 | 160,000 |
| 0.02 | 40,000 |
| 0.07 | 16,000 |

**Comments on the connection between gradient flow and gradient descent.** In this paper, we show that gradient flow from a small-scale initialization is driven towards low-sharpness regions. This should imply that gradient flow and gradient descent up to a reasonably large learning rate should follow the same trajectory when starting from small-scale initialization, because they do not

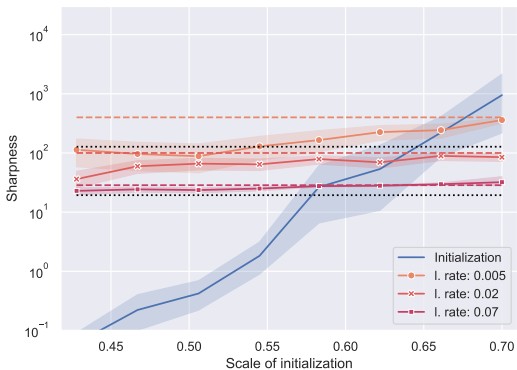

Figure 7: For a non-linear MLP, sharpness at initialization and after training, for various learning rates and initialization scales. For a given learning rate $\eta$, the dashed lines represent the $2/\eta$ threshold. The dotted black lines represent the lower and upper bounds given in Theorem 1 and Corollary 2 of the paper.

go in regions of high sharpness where the difference between gradient flow and gradient descent would become significant. This intuition is supported by Figure 1b where we see that, for small initializations, the sharpness after training is independent of the learning rate. We leave further investigation of these questions for future work.

## D   Additional related work

**Progressive sharpening.**   Cohen et al. (2021) show that the edge of stability phase is typically preceded by a phase of progressive sharpening, where the sharpness steadily increases until reaching the value of $2/\eta$. Our setting of small-scale initialization presents an example of such a progressive sharpening (although we make no statement on the monotonicity of the increase in sharpness). Other works have proposed analyses of progressive sharpening. Wang et al. (2022) suggest that progressive sharpening is driven by the increase in norm of the output layer. MacDonald et al. (2023) assume from empirical evidence a link between sharpness and the magnitude of the input-output Jacobian, and show that the latter has to be large for the loss to decrease. Finally, Agarwala et al. (2023) propose and analyze a simplified model with quadratic dependence in its parameters, which exhibits a progressive sharpening phenomenon.

**Connection with deep matrix factorization.**   Regression with deep linear networks can be seen as an instance of a matrix factorization problem. There is a well-established literature studying the implicit regularization of gradient descent for this class of problem (see, e.g., Gunasekar et al., 2017; Arora et al., 2019b; Li et al., 2021; Yun et al., 2021). However, this line of work study under-determined settings where there are an infinite number of factorizations reaching zero empirical risk, and study the implicit regularization in function space. On the contrary, we consider an over-determined setting where there is a single optimal regressor, and study the regularization in parameter space.

