# OpenReview forum: "Deep linear networks for regression are implicitly regularized towards flat minima"
_NeurIPS.cc/2024/Conference — NeurIPS 2024 poster_

### Official Review · Reviewer_PPoT · 2024-07-01

**Soundness:** 4
**Presentation:** 4
**Contribution:** 3
**Rating:** 6
**Confidence:** 2

**Summary:**

The paper explores the behavior of deep linear neural networks in the context of overdetermined univariate regression, which is the setting of a univariate response $y$, more samples than input dimensions, and nonsingular data covariance matrix.

The first result concerns the empirical risk minimizer (this is not equal to the gradient descent or gradient flow estimator)  and shows a lower bound on the sharpness of the ERM.

The next set of results show that gradient flow, a limit of gradient descent with a vanishing learning rate, implicitly regularizes the network towards flat minima, with sharpness close to a constant times the lower bound. This is proven for both small-scale and residual initializations.

**Strengths:**

* The exposition is excellent. There are three main results, each given in its own section corresponding to Section 3, 4, and 5.
* The setting described in Section 2 is mathematically clear
* The first main result, Theorem 1, reaches a similar conclusion to previous work in Mulayoff and Michaeli (2020) but relaxes identity data covariance assumption and uses a supposedly simpler proof.
* The next set of results characterize the gradient flow minimizers. This is indeed of independent interest, although the connection to sharpness is also interesting.
* The sharpness papers I have come across seem to focus on sharpness induced by SGD. It's interesting that here the authors were able to demonstrate a preference for sharpness for deterministic gradient flow, albeit with specific initialization schemes.

**Weaknesses:**

* I missed a rigorous theoretical connection between the three main results and the behavior of the gradient descent minimizer. If the connection is only empirical in nature, it might be good to highlight this
* The work employs, to my eyes at least, a rather limited definition of sharpness. Are the results really very particular to sharpness being the largest eigenvalue of the Hessian?
* I'm missing some motivation for the two initialization schemes studied. Are they studied simply because they've been studied before and so the results here can borrow from existing work?
* I find the title misleading. An architecture cannot have a preference for flat minima(?). It must be the architecture in conjunction with how it is learned. In this case, the main results concern very specific configurations of gradient flow.
* The implication for learning rate design from this theoretical analysis seems rather weak to me. I mean that I cannot envision a practitioner implementing a learning rate design based on the results presented here.

**Questions:**

* Apologies for this very naive question, but what does $S(\mathcal W)$, the largest eigenvalue of $\mathcal W=(W_1,\ldots,W_L)$ mean exactly? This $\mathcal W$ is not itself a matrix.
* Each of your results in Sections 3, 4, and 5 are based on existing proof techniques. Specifically, Section 3 proof is based on Mulayoff and Michaeli 2020, Section 4 on Ji and Telgarsky (2020), and Section 5 on Zou et al 2020, Sander et al 2022. I am not familiar with these papers. Do they attempt to prove similar things to this paper? Do you reach the same conclusions but with different proof techniques?

**Limitations:**

* There are some obvious limitations that are par for the course in papers attempting to prove theoretical results in deep learning. Namely, the architecture here is extremely oversimplified. The data covariance being full rank is also unrealistic. Finally, the main results are about gradient flow and discrete-time stochastic gradient descent is surely very different in nature.

---

> ### Author Rebuttal · Authors · 2024-08-06
>
> * Rigorous connection between theoretical results and GD:
>
> We agree that the connection between our results on gradient flow and the case of GD is not rigorously proven, and we will further emphasize this in the next version.
>
> * Other definitions of sharpness:
>
> We agree that there are several definitions of sharpness, but believe that the largest eigenvalue of the Hessian is the most relevant quantity to study optimization dynamics. For example, in quadratic optimization, the maximal learning rate is given by 2 over this quantity. This quantity also appears to show convergence of GD in convex optimization. Our results are specific to this quantity, and we will mention this caveat.
>
> * Motivation for the initialization schemes:
>
> The first scheme (small-scale initialization) is a classical Gaussian initialization, but with small variance. The scale of the initialization is known to play a key role in training of neural networks: the small-scale initialization corresponds to the “feature learning” regime where the weights change significantly during training, by opposition to the “lazy” regime (see e.g. [1]). This motivates the study of this regime.
>
> [1] On Lazy Training in Differentiable Programming, Chizat, Oyallon, Bach, NeurIPS 2019.
>
> The second scheme (residual initialization) corresponds to a simplification of non-linear residual networks. More precisely, a simple non-linear residual architecture is given by
>
> $$h_{k+1} = h_k + \sigma(N_k h_k),$$
>
> where the matrices $N_k$ are Gaussian at initialization. Removing the non-linearity, we get
>
> $$h_{k+1} = h_k + N_k h_k = (I + N_k) h_k = W_k h_k.$$
>
> We recover (up to scaling factors) our residual initialization (see eq. (2) of the paper). We further note that this specific initialization scheme for linear networks has not been previously studied to our knowledge. The simpler but less realistic scheme which has been studied is the identity initialization $W_k = I$.
>
> We will add these remarks in the next version.
>
> * Title:
>
> We agree that the choice of optimization algorithm plays a key role. We had preferred brevity in the current title, but will consider a more precise title.
>
> * Implication for learning rate design:
>
> Our paper provides a step towards understanding training dynamics of neural networks. In particular, we explain the largest learning rate for training deep linear networks for regression tasks (see in particular Fig. 1 in the additional PDF). However, the goal of this work is not to make claims beyond this setting, and we leave extensions to more realistic settings for future work.
>
> We nevertheless note that our dependence of learning rate on depth (namely, constant over depth) matches other papers that study scaling of neural networks [2, 3]. These analysis concern more complex cases (non-linear residual networks), but are limited to the beginning of training. Our analysis regards a simpler architecture but goes beyond, in describing the evolution of the weights throughout training, and showing convergence to a global minimizer.
>
> [2] Tensor Programs VI. Yang, Yu, Zhu, Hayou, ICLR 2024.
>
> [3] The Feature Speed Formula. Chizat, Netrapalli, 2023.
>
> * Definition of $S(\mathcal{W})$:
>
> Thanks for this question. Denoting the empirical risk as $R(\mathcal{W})$ which depends on parameters $\mathcal{W} \in \mathbb{R}^p$ (where we flatten all parameters in a single vector), we can consider the Hessian of $R$, which associates $\mathcal{W}$ to a matrix $H(\mathcal{W}) \in \mathbb{R}^{p \times p}$. Then $S(\mathcal{W})$ is the largest eigenvalue of $H(\mathcal{W})$. This is explained in the paper in Section 3. Nevertheless, we agree that the notation might be misleading when first introduced, and we will make this clearer.
>
> * Link with the literature:
>
> While the paper emphasizes connections with the literature, there are very substantial differences between our paper and those mentioned by the reviewer, both in the proof techniques and in the goal of the papers.
>
> Closest to our approach is Mulayoff and Michaeli, which also study the sharpness of deep linear networks. As noted by the reviewer, we relax their assumption on the identity covariance matrix. Our proof technique is also different, and does not involve tools from tensor algebra. Furthermore, their study is concerned with describing the sharpness of the set of minimizers (the equivalent of our Theorem 1), while we go significantly beyond, in describing the training dynamics and the sharpness of the network after training, as well as providing experiments which connect the initialization scale, the learning rate and the sharpness.
>
> The other papers mentioned by the reviewer study convergence of deep linear networks. The goal of the present paper is therefore very different, since it is centrally concerned with sharpness properties of the network.
>
> More precisely, Ji and Telgarsky show convergence of deep linear networks for classification tasks. In the present paper, we consider a regression task, which introduces additional technicalities, because we cannot leverage the fact that the minimizer is at infinity, as is done by Ji and Telgarsky to simplify the analysis. Furthermore, many ideas in our paper are not present in theirs, such as the connection with sharpness, the experiments with large learning rates, and the study of residual networks.
>
> Zou et al and Sander et al study convergence of deep linear residual networks starting from an identity initialization ($W_k = I$). We consider the more realistic residual initialization (see the answer on initialization schemes above), bringing significant additional technicalities in controlling the initialization randomness. Again, many ideas in our paper are not present in theirs, such as the connection with sharpness, the experiments with large learning rates, and the study of non-residual networks.
>
> * Limitations:
>
> The question of extension to more complex settings is shared with other reviewers and is addressed in the common rebuttal.

---

> > ### Comment · Reviewer_PPoT · 2024-08-13
> > **Thanks!**
> >
> > Thank you to the authors for their detailed response. I'd like to maintain my original rating that the paper is a "technically solid, moderate-to-high impact paper, with no major concerns with respect to evaluation, resources, reproducibility, ethical considerations."

---

> > > ### Author Response · Authors · 2024-08-13
> > >
> > > Thank you again for your time and very interesting review.

---

### Official Review · Reviewer_hhnc · 2024-07-07

**Soundness:** 3
**Presentation:** 3
**Contribution:** 2
**Rating:** 6
**Confidence:** 4

**Summary:**

This paper studies the implicit bias of gradient flow on deep linear networks for overdetermined univariate regression problems. A lower bound on the sharpness of any minimizer is first derived for the general data covariance matrix, then it is shown that gradient flow with small initialization finds a minimizer with a sharpness no larger than a constant times the lower bound, where the constant depends on the condition number of the data covariance matrix. A similar result is also derived for GF on linear residual networks.

**Strengths:**

1. Well-motivated problem. Good introduction. Clear writings. Theoretical results are carefully explained in the main paper.
2. New convergence results for deep linear networks and linear residual networks, together with the upper bound on the sharpness.

**Weaknesses:**

1. **Tightness of Theorem 2**: In Mulayoff and Michaeli (2020), the lower bound on the sharpness is tight for whitened data: there exists a minimizer that achieves the lower bound. However, Theorem 2 in this paper only provides a lower bound. Is this lower bound improvable? For example, if the true lower bound has $\Lambda$ instead of $\lambda$, then the following results show that the sharpness minimizes found by GF is no larger than a constant times the lower bound, where the constant now does not depend on condition number.
2. **Relevance of the problem setting**:  The sharpness is often considered to be affecting the generalization error of the trained network. However, this paper studies overdetermined linear regression, where every minimizer corresponds to the same input-output map $w^*$, thus having the same generalization error. Why does one care about studying the sharpness of the minimizer in this case, if sharpness does not affect generalization at all?

**Questions:**

See Weaknesses

**Limitations:**

See Weaknesses

---

> ### Author Rebuttal · Authors · 2024-08-06
>
> > Tightness of Theorem 2: In Mulayoff and Michaeli (2020), the lower bound on the sharpness is tight for whitened data: there exists a minimizer that achieves the lower bound. However, Theorem 2 in this paper only provides a lower bound. Is this lower bound improvable? (...)
>
> We agree with the reviewer that there is a gap between our lower bound $2L \lambda \||w^\star\||^{2-2/L}$ in Theorem 2 and the upper bound on the sharpness of the GF solution $8L \Lambda \||w^\star\||^{2-2/L}$ in Corollary 3, and that understanding this gap is a very interesting question. We give below a few comments, which we will add to the paper.
>
> First, an inspection of the proof for the lower bound shows that it can be improved by replacing $\lambda$ by $a := (w^\star / \||w^\star\||) X^\top X (w^\star /  \||w^\star\||)$. Note that $\lambda \leq a \leq \Lambda$, and the value of $a$ depends on the alignment between the optimal regressor $w^\star$ and the eigenvectors of the data covariance matrix $X^\top X$. For example, if $w^\star$ is aligned with the eigenvector associated with the largest eigenvalue $\Lambda$, then $a=\Lambda$, and we get the improvement suggested by the reviewer. However, if $w^\star$ is aligned with the eigenvector associated with the smallest eigenvalue $\lambda$, then $a=\lambda$, showing no improvement with respect to our current bound.
>
> Second, moving on to the upper bound, it is possible to construct a minimizer with sharpness
> $2 \||w^\star\||^{2-2/L} \sqrt{L (\Lambda^2 + (L-1) a^2)}$. While this quantity is not exactly matching the lower bound, it is close to the lower bound when $L$ is large. In particular, the ratio between both quantities goes to 1 when $L$ goes to infinity. This minimizer is constructed by taking all the weight matrices to be rank-one, with norm $\||w^\star\||^{1/L}$, aligned first singular vectors, and the first right singular vector of $W_1$ is aligned with $w^\star$. The proof of this fact is too long to be included in the present rebuttal, but will of course be added to the next version of the paper. It is quite similar to the proof of Corollary 3 of the paper.
>
> A question which remains is to rewrite the sharpness of the minimizer found by GF as a function of the quantity $a$. We believe that it is possible to do so, and to obtain a bound on this sharpness close to the upper bound given above. The main difficulty is to quantify the distance between the first right singular vector of $W_1$ and $w^\star$. As explained below Corollary 1 in the paper, our proof shows that these two vectors have to be close, but quantifying their distance involves significant additional computations. We leave these for future work.
>
> Finally, we note that Figure 1b shows that the sharpness after training is less than the upper bound of Corollary 3, showing indeed some room for improvement.
>
> > Relevance of the problem setting: The sharpness is often considered to be affecting the generalization error of the trained network. However, this paper studies overdetermined linear regression (...). Why does one care about studying the sharpness of the minimizer in this case, if sharpness does not affect generalization at all?
>
> Your question on the link with generalization is shared with other reviewers and is addressed in the common rebuttal. In particular, our results are mostly concerned with optimization dynamics, and studying the sharpness in this setting allows for example to understand the maximal learning rate leading to stable training.
>
> Let us mention two other applications of sharpness to the understanding of the training dynamics.
>
> First, we show that GF from a small-scale initialization is driven towards low-sharpness regions. This should imply that GF and GD up to a reasonably large learning rate should follow the same trajectory when starting from small-scale initialization (because they do not go in regions of high sharpness where the difference between GF and GD would become significant). This is suggested by Figure 1b where we see that, for small initializations, the sharpness after training is independent of the learning rate. The fact that GF and GD follow the same trajectory is itself interesting because GF is often easier to analyze than GD, so it is nice to describe settings where the GF analysis provably matches the GD case.  We leave this study for future work, as well as the investigation of more complex cases where the minimizers implement different input-output functions.
>
> Second, phenomena that have been of interest lately in understanding training dynamics of deep networks are the “progressive sharpening” and “edge of stability” [1-5]. As explained in the introduction, these phenomena are quite crucial to the training dynamics since they suggest an implicit regularization by large learning rates (see equation (1) in the paper). One of our initial motivations for this paper was to understand progressive sharpening for deep linear networks. Our results partially answer this question, in showing that a small-scale initialization leads to an increase of the sharpness during training (see lines 114-116). However, there are still many open questions on this topic (in particular the link with the phenomena observed for non-linear neural networks), which are left for future work.
>
> We will add a more detailed discussion of these topics in the next version of the paper.
>
> [1] Agarwala, Pedregosa, Pennington. Second-order regression models exhibit progressive sharpening to the edge of stability. ICML 2023.
>
> [2] Cohen, Kaur, Li, Kolter, Talwalkar. Gradient descent on neural networks typically occurs at the edge of stability, ICLR 2021.
>
> [3] Damian, Nichani, Lee. Self-stabilization: The implicit bias of gradient descent at the edge of stability. ICLR 2023.
>
> [4] MacDonald, Valmadre, Lucey. On progressive sharpening, flat minima and generalisation. 2023.
>
> [5] Wang, Li, Li. Analyzing sharpness along GD trajectory: Progressive sharpening and edge of stability. NeurIPS 2022.

---

> > ### Comment · Reviewer_hhnc · 2024-08-09
> >
> > Thank the authors for addressing my concerns. I increase the score from 4 to 6. I suggest the authors adding these discussions to the appendix.

---

> > > ### Author Response · Authors · 2024-08-11
> > >
> > > Thank you for your very thoughtful review and for raising the score. We will add the discussion to the next version of the paper.

---

### Official Review · Reviewer_WC3N · 2024-07-13

**Soundness:** 3
**Presentation:** 4
**Contribution:** 3
**Rating:** 7
**Confidence:** 2

**Summary:**

The paper considers a toy non-convex optimization problem, namely overdetermined univariate regression with a deep linear network. Their main contributions are:

1) A lower bound on the sharpness of the minimizers of the empirical risk. In particular, if the step size is chosen too big, then gradient descent will fail to converge.

2) They show that the sharpness of the minimizer found by gradient flow is at most a constant multiple of the above lower bound, for both small-scale and residual initialization. This constant does not depend on the width or depth, but only on the conditioning number of the data. This shows an implicit regularization towards flat minima (the empirical minimizers of the risk can have arbitrarily large sharpness).

On the technical side, they prove convergence of the GF and characterize the solution at convergence in both initialization regimes. Numerical simulations are provided to illustrate and substantiate their claims.

**Strengths:**

- The theoretical analysis is substantial. The results are novel and not straightforward. At the same time, they are easy to understand  and offer clear insights.

- The qualitative picture obtained in this paper, with the interplay between step size, scale at initialization, and sharpness of the gradient descent solution, is convincing and surprisingly clear, given the complex non-convex problem.  In particular, I think it is interesting to have in a single model the behavior of sharpness, edge of stability and GD minimizer (even if only qualitatively).

- The discussions are reasonably clear. The plots help a lot to understand the general message of the paper.

**Weaknesses:**

The paper considers a simple setting: deep linear network and underdetermined regime. This allows the authors to characterize interesting and non-trivial behavior. However, it is unclear how much these results can extend beyond this simple setting.

**Questions:**

Here, it is assumed that the data matrix is full rank. Is it not sufficient to assume that there exists a unique minimizer?

**Limitations:**

Yes

---

> ### Author Rebuttal · Authors · 2024-08-06
>
> > The paper considers a simple setting: deep linear network and underdetermined regime. This allows the authors to characterize interesting and non-trivial behavior. However, it is unclear how much these results can extend beyond this simple setting.
>
> The question on the extension to more complex settings is shared with other reviewers and is addressed in the common rebuttal.
>
> > Here, it is assumed that the data matrix is full rank. Is it not sufficient to assume that there exists a unique minimizer?
>
> We note that our assumption that the data covariance matrix $X^\top X$ is full rank is in fact _equivalent_ to the uniqueness of the minimizer of the linear regression of $y$ on $X$.
>
> Nevertheless, it is a very interesting question to study the case where this assumption is relaxed, that is, where $X^\top X$ is not full rank. A part of our results which adapts reasonably easily to this relaxation is Theorem 1 (and 2). In this case, there exists an infinity of minimizers of the linear regression of $y$ on $X$, and $w^\star$ should now be the minimizer with the smallest norm. Furthermore, the lowest eigenvalue $\lambda$ of $X^\top X$ is now equal to $0$ and should be replaced by  $a := (w^\star / \||w^\star\||) X^\top X (w^\star /  \||w^\star\||)$. If $w^\star$ is nonzero, then $a$ is also nonzero even though $X^\top X$ is not full rank.
>
> We also refer the reviewer to Figure 3a in the additional PDF, which considers the case $n < d$, implying in particular that the data covariance matrix $X^\top X$ is not full rank. We qualitatively observe a similar connection between learning rate, initialization scale and sharpness as in the case of full-rank data (Figure 1b).
>
> We will add these comments (and corresponding proof) in the next version of the paper. We leave the study of the extension of the other results in the paper to future work.

---

> > ### Comment · Reviewer_WC3N · 2024-08-10
> >
> > I thank the authors for their detailed response. I have no further questions at the moment.

---

> > > ### Author Response · Authors · 2024-08-11
> > >
> > > Thank you again for your time and very interesting review.

---

### Official Review · Reviewer_hdrG · 2024-07-19

**Soundness:** 4
**Presentation:** 4
**Contribution:** 2
**Rating:** 6
**Confidence:** 3

**Summary:**

The authors show three new results for deep linear networks (DLN).  First, they show that any DLN that implements the optimal linear regressor must have a certain "sharpness".  This amounts to a lower bound on the largest eigenvalue of the Hessian matrix at that set of weights.  They then argue that the weights found by using gradient flow with two different types of initialization find minima that have a sharpness that is within a constant of this lower bound.  They interpret these results as an implicit regularization of gradient flow towards flat minima.

**Strengths:**

Results are novel.  They are also of at least abstract interest to the community of researches working on the theory of deep learning.  I also agree with their assertion that some of the intermediate results presented on the structure of weight matrices in DLNs post-training (e.g. approx rank 1 and aligned in small scale initialization case) might be of interest.

**Weaknesses:**

The paper kind of has a mixed message, and doesn't really make the connection to generalization power in a way that is intepretable.  Theorem 1 shows that the sharpness has to grow at a rate that is essentially linear in the number of layers.  But then they mention that prior work indicates that flat minima generalize better, which suggests that making the DLN deeper is going to hurt you in this regard.  The subjective interpretation of gradient flow going to flat minima is hard to digest when the authors just proved that there are no flat minima.

**Questions:**

Do these results add to our understanding of the generalization power of deep linear networks trained with gradient flow?

---

> ### Author Rebuttal · Authors · 2024-08-06
>
> > The paper kind of has a mixed message, and doesn't really make the connection to generalization power in a way that is intepretable. Theorem 1 shows that the sharpness has to grow at a rate that is essentially linear in the number of layers. But then they mention that prior work indicates that flat minima generalize better, which suggests that making the DLN deeper is going to hurt you in this regard.
>
> The link between sharpness and generalization is very delicate, for example because reparameterizing the neural network changes the sharpness but not the generalization power. Your question on the link with generalization is shared with other reviewers and is addressed in the common rebuttal. Our results are mostly concerned with optimization dynamics and not so much with the link with generalization. In particular, we do not claim that our results indicate that increasing the depth of the neural networks leads to a worse generalization. Nevertheless, we agree that the current formulation of the introduction might be somewhat misleading in this regard, and we will reformulate to remove any confusion.
>
> > The subjective interpretation of gradient flow going to flat minima is hard to digest when the authors just proved that there are no flat minima.
>
> For a given (fixed) architecture, our results show that gradient flow drives the network towards flat minima, since, among all minima, the one selected by gradient flow is “close” to the one with lowest sharpness. Here, “close” has an objective and rigourous meaning, which is that the ratio between the sharpness of the gradient flow solution and the lowest sharpness is architecture-independent and is given by precise formulas in Corollaries 3 and 4. This being said, as the reviewer rightfully indicates, the lowest sharpness itself depends on the architecture (through the depth). However, this should not be interpreted as the fact that there are no flat minima, but rather that the characteristics of the loss landscape (in this case, the sharpness of the flattest minima) are architecture-dependent, which is reasonable. We finally note that the fact that the sharpness of the flattest minima depends linearly on the depth was already shown in previous works, for instance in Mulayoff and Michaeli (2020).
>
> > Do these results add to our understanding of the generalization power of deep linear networks trained with gradient flow?
>
> This question is shared with other reviewers and is addressed in the common rebuttal.

---

> > ### Comment · Reviewer_hdrG · 2024-08-13
> >
> > Thank you for your careful response both here an in the general rebuttal.

---

> > > ### Author Response · Authors · 2024-08-14
> > >
> > > Thank you again for your time and very interesting review.

---

### Author Rebuttal · Authors · 2024-08-06

Dear reviewers,

We warmly thank you for your time and relevant comments, which will help us improve our work. If accepted, we will take into account your suggestions, making use of the additional page.

Since several reviewers raised the relevant questions of the link with generalization and of the extension to more complex settings, we address these two questions below, and leave the answers to other questions in individual responses.

Sincerely,

The authors

----

**Link with generalization:**

Sharpness analysis is quite intricate because **sharpness is linked both to generalization and optimization**. Generalization depends on the input-output function implemented by the trained neural network, while optimization depends on the loss landscape, and thus on the parameterization of the neural network.

**In this paper, we focus on the link with optimization dynamics** by choosing a setting (linear network, overdetermined linear regression) where we are able to disentangle both effects since all minimizers of the empirical risk implement the same function and thus have the same generalization error.

In this context, sharpness allows to **understand the optimization dynamics**. For example, leveraging our Theorem 1, we can **predict the largest learning rate for successful training**. This is mentioned in the paper (in the comments after Theorem 1). To better illustrate this fact, we refer to Figure 1 in the additional PDF, which shows the distance to the optimal regressor as a function of the learning rate, for various depths. We see a transition occurring when the learning rate crosses a threshold which depends on depth. For a given depth, the value for the threshold (dashed line) is equal to $2/S_{\min}$ where $S_{\min}$ is given in Theorem 1. We note that this gives a quantitative answer to the observations of [1], which shows the existence of a maximal architecture-dependent learning rate beyond which training fails.

Although this is not our original motivation, we also note that a simple change to our setting allows to make appear the connection between sharpness and generalization. To this aim, we consider the underdetermined case, where the number of data $n$ is lower than the dimension $d$ (while keeping the rest of the setup identical to the paper). We refer to Figure 3b in the additional PDF, which shows in this case a **correlation between generalization and sharpness**. This suggests that the tools developed in the paper could be used in this case to understand the generalization performance of deep (linear) networks, and we leave this analysis for future work.

All in all, the connection with generalization is an important question, which we will clarify by adding this discussion to the paper.

[1] The large learning rate phase of deep learning: the catapult mechanism, Lewkowycz, Bahri, Dyer, Sohl-Dickstein, Gur-Ari, 2020.

-----------

**Extension to more complex settings:**

Extending our results to more complex settings is an important question raised by the reviewers, and we performed **additional experiments** that we will add to the paper. We consider two additional settings: **non-linear MLP, and overdetermined regime for deep linear network** (when the number of data $n$ is lower than the dimension $d$). In both cases (see Figures 2 and 3a of the additional PDF), **we qualitatively observe a similar connection between learning rate, initialization scale and sharpness** as in Figure 1b of the paper. More precisely, we observe in all cases that the sharpness after training grows when increasing the scale of the initialization or when decreasing the learning rate. In the case of non-linear MLPs (Figure 2 of the additional PDF), we also see that for large initialization, the sharpness after training plateaus at 2/lr, as in the linear case. For small initialization, the sharpness after training is less that 2/lr, and is close to the bounds described in the paper in the linear case (dotted black lines).

---

### Decision · Program_Chairs · 2024-09-25

**Decision:**

Accept (poster)

**Comment:**

The paper proposes an implicit bias of gradient flow for deep linear networks and overdetermined univariate regression that the sharpness of the minimizer is bounded.

The reviewers discussed the paper in detail, most of the concerns (regarding link with generalization, extension to general settings, and tightness of the bounds, etc.) were resolved during the discussion period and thus the reviewers agreed upon positive ratings. Overall I recommend accept.